# 4+3 Phases of Compute-Optimal Neural Scaling Laws

**Elliot Paquette**[*]
McGill University
`elliot.paquette@mcgill.ca`

**Courtney Paquette**
Google DeepMind & McGill University
`courtney.paquette@mcgill.ca`

**Lechao Xiao**[†]
Google DeepMind

**Jeffrey Pennington**[†]
Google DeepMind

## Abstract

We consider the solvable neural scaling model with three parameters: data complexity, target complexity, and model-parameter-count. We use this neural scaling model to derive new predictions about the compute-limited, infinite-data scaling law regime. To train the neural scaling model, we run one-pass stochastic gradient descent on a mean-squared loss. We derive a representation of the loss curves which holds over all iteration counts and improves in accuracy as the model parameter count grows. We then analyze the compute-optimal model-parameter-count, and identify 4 phases (+3 subphases) in the data-complexity/target-complexity phase-plane. The phase boundaries are determined by the relative importance of model capacity, optimizer noise, and embedding of the features. We furthermore derive, with mathematical proof and extensive numerical evidence, the scaling-law exponents in all of these phases, in particular computing the optimal model-parameter-count as a function of floating point operation budget. We include a colab notebook nanoChinchilla[3] that reproduces some key results of the paper.

## 1 Introduction

The advent of large language models (LLMs) has changed our perceptions of the landscape of optimization and is resulting in the emergence of new interesting questions related to scaling. Prior to LLMs and other large models, we often viewed the large-scale optimization problems as being limited by the amount of data. In training language models, in contrast, data can be effectively infinite. Thus, *compute budgets* can be the limitation. This leads to the following natural question: given an architecture, given a fixed compute budget, and having unlimited data, *how should one select the model size to minimize loss?*

To formally address this question, let us consider the general learning problem,

$$\min_{\theta \in \mathbb{R}^d} \left\{ \mathscr{P}(\theta) = \mathbb{E}_x[\mathscr{R}(\theta; x)] \right\}, \quad \text{where } \mathscr{R} : \mathbb{R}^d \to \mathbb{R}, \tag{1}$$

the number of parameters $d$ is large, and the data vector $x$ is drawn from an unknown distribution. We solve (1) using stochastic algorithms, such as stochastic gradient descent (SGD) with batch size $B$, under various parameter sizes $d$, that produce a sequence of iterates $\{\theta_r\}$. A standard formula used in practice to measure compute is the "6ND" formula [26], that is,

$$\text{Compute (flops}^4 \mathfrak{f}) = (\text{iterations of alg. } (r) \times \text{batch size } (B)) \times \text{parameters } (d). \tag{2}$$

---

[*]Corresponding author; website: `https://elliotpaquette.github.io/`.

[†]The authors contributed equally to the paper.

[3]Available at: `https://tinyurl.com/2saj6bkj`

Therefore, we can plot the loss curve $\mathscr{P}(\theta_r; d) = \mathscr{P}(r; d) = \mathscr{P}(\mathfrak{f}/(d \cdot B); d)$ as a function of flops (see Fig. 1). The question now is: given a fixed number of flops $\mathfrak{f}$ and given batch size $B$, how should we choose the parameters $d$ so that we get the best loss, i.e. find $d^\star$ solving the constrained problem

$$d^\star(\mathfrak{f}) \in \arg\min_d \mathscr{P}\left(\tfrac{\mathfrak{f}}{d \cdot B}; d\right) = \arg\min_d\left\{\mathscr{P}(\theta_r; d) \text{ subj. to } \mathfrak{f} = (r \times B) \times d\right\}. \tag{3}$$

**Main contributions.** In this work, we analyze a three parameter simple model, which we call *power-law random features* (PLRF) [30, 5]. The three parameters in the PLRF are the data complexity ($\alpha$), target complexity ($\beta$) and model-parameter count $d$. Using this model, we derive a deterministic equivalent for the expected loss, as a function of $\alpha$, $\beta$, and $d$, that captures the training dynamics of one-pass SGD. This can be used to derive numerical predictions for the scaling laws. We also extract exact expressions for the compute-optimal scaling laws and the optimal parameter $d^\star(\mathfrak{f}) \in \text{argmin}_d \mathscr{P}(\frac{\mathfrak{f}}{d \cdot B}; d)$ for large[5] $d$, and give some estimates on the order of $d$ necessary for these scaling laws to take hold.

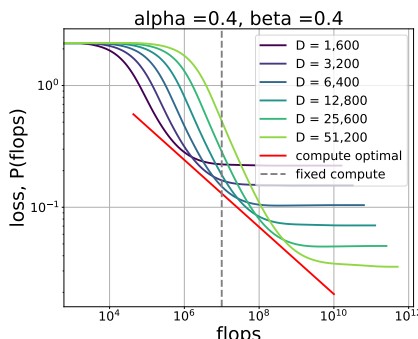

Figure 1: **Toy scaling problem**. We plot the loss function, $\mathscr{P}(\theta_r; d)$ as a function of flops $\mathfrak{f}$ using (2). Consider a fixed number of flops $\mathfrak{f} = 10^7$ (dashed line). If we had chosen, e.g., $d = 1600$, we can run for a long time, but our model does not have a lot of capacity and thus the value of the loss function remains high. On the hand, we can increase capacity by choosing a large number of parameters (e.g., $d = 51,200$), but because our compute is fixed we can not run our algorithm for very long. Thus the loss value is still large. The optimal choice is $d \approx 6,400$. When done for every choice of $\mathfrak{f}$ gives the compute-optimal curve (red line). This choice of $(\alpha, \beta)$ (Phase I) is an example of where *model capacity* controls the compute-optimal curve, but it is not the only behavior we show. In other phases the compute-optimal is controlled by *poor model embedding* (Phase II, III) and *SGD noise* (Phase III, IV).

We also observe for a large portion of the $(\alpha, \beta)$-phase plane, the optimal parameter is $d^\star(\mathfrak{f}) = \mathfrak{f}^{1/2}$, suggesting a regime of *universal scaling behavior* (see Fig. 4a and Table 2). This verifies theoretically the Chinchilla scaling [24].

The PLRF is not only analyzable, but also exhibits a rich behavior of compute-optimal curves/loss curves, which are qualitatively and quantitatively different depending on the strengths of the data ($\alpha$) vs. target ($\beta$) complexity. Particularly, we show that there are *4 distinct (+3 sub phases)* compute-optimal curve/loss curve behaviors.

*Model constrained compute-optimal curves.* In two of the phases (Phase Ia,b,c and Phase II), it is the underlying model that dictates the curves. The algorithm has little/no impact. This appears in two forms. The first behavior are compute-optimal curves controlled by the capacity of the model (Phase Ia,b,c). Here once the algorithm reaches the limiting risk value

possible (capacity), it is better to increase the model-parameter $d$. Another type of loss dynamics is due to poor model feature embedding (Phase II). Here the features are embedded in a way which is difficult to train. After an initial large decrease in the loss value, this feature embedding distortion frustrates the algorithm and training slows, but it continues to solve. However, solving to capacity wastes compute, in that it is compute-favored to increase the model parameter count $d$.

*Algorithm constrained compute-optimal curves.* For some choices of $(\alpha, \beta)$ (Phase III and IV), it is the noise produced by the SGD algorithm that ultimately controls the tradeoff. Here the algorithm matters. Indeed, another algorithm could change the compute-optimal curves for these phases.

**Related work.** The key source of inspiration for this work are [24, 26], which identified compute optimality as a fundamental concept in scaling large language models and made a substantial empirical exploration of it. The problem setup was formulated by [30], where additionally data-limited scalings were considered, but compute optimality was not (nor indeed any algorithmic considerations); see also [8] where gradient flow is considered in the same setting.

---

[4]Here and throughout we use flops to mean "floating point operations" and not as the rate floating point operations per second. We also drop the pre-factor 6 in "6ND" formula for simplicity.

[5]We discuss *how* large is large, but the truth is somewhat complicated and also quite dependent on the desired precision. If $\pm 0.05$ on the achieved scaling laws is tolerable, a flat $d > 1000$ seems to suffice across all phases.

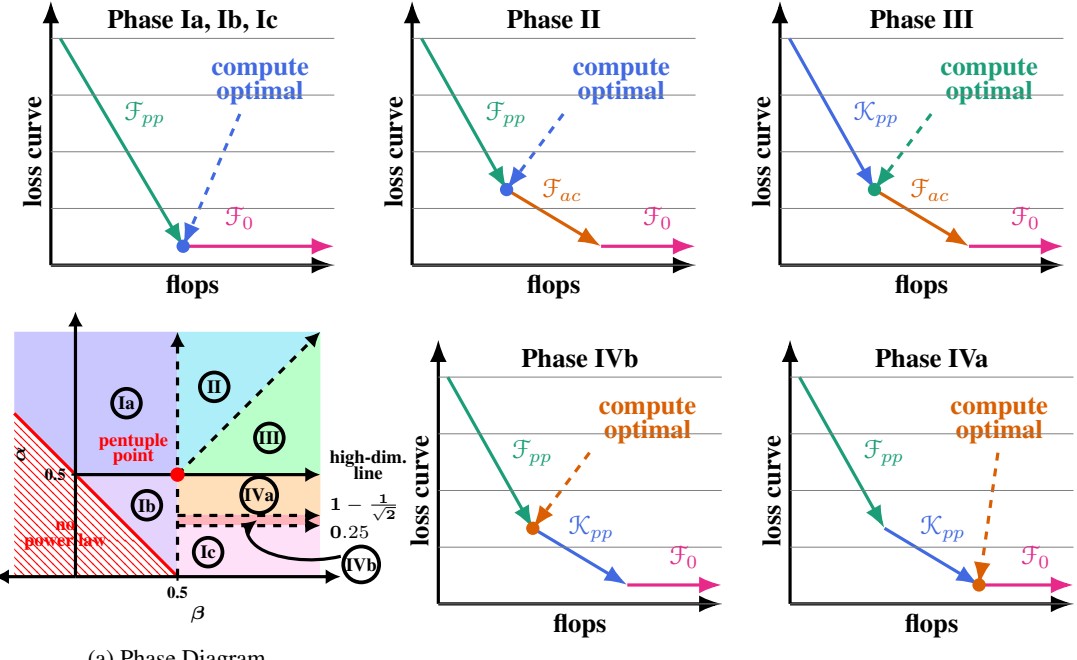

(a) Phase Diagram

Figure 2: **Phase Diagram and Cartoon Plots of Loss Curves in Different Phases. (a) Phase Diagram.** Colored regions represent where the training of the risk/compute-optimal curves look qualitatively and quantitatively different depending on $\alpha$ and $\beta$. This, in term, yields different scaling law ($\eta$) and parameter count ($\xi$) exponents for each of the phases. Critical point at $\alpha = \beta = 1/2$ where all behaviors are observed. The other plots illustrate the components of $\mathcal{F}$ (via $\mathcal{F}_0, \mathcal{F}_{pp}, \mathcal{F}_{ac}$) and $\mathcal{K}_{pp}$ which dominate the loss curve for each phase (see Sec. C.4.1 & Sec. C.4.1 for proofs); tradeoff between the functions where the compute-optimal point occurs is also indicated (see Sec. 2.1 for definitions and Sec. 3.1 & Sec. D for proofs).

There is a substantial body of work considering scaling laws of losses (trained to minimum-loss) of dataset size vs parameter count, in a variety of settings (linear, random features, deep networks). See especially: [5, 39, 41], wherein a "hidden-manifold" model is considered for the data. We note that as we consider one-pass SGD, some dataset/parameter-count scaling laws are implicit from the results here; however, the training method (one-pass SGD) is, in some regimes, suboptimal given unlimited compute.

For additional related work on random features models (and sample complexity), random matrix theory in machine learning, and other deterministic equivalents for SGD, see Section A. We note that while this paper is fundamentally about computation, but the novel mathematical contributions could also be recast in terms of generalization bounds of one-pass SGD, some of which are new. For a detailed comparison of the convergence rates and sample complexity, see Table 4.

## 1.1 Problem Setup: SGD on Power-law Random Features

In this work, we analyze the three parameter *power-law random features* (PLRF) model, that is,

$$\min_{\theta \in \mathbb{R}^d} \left\{ \mathscr{P}(\theta) \overset{\text{def}}{=} \mathbb{E}_x[(\langle W^T x, \theta \rangle - \langle x, b \rangle)^2] \right\}. \tag{4}$$

We embed the data vector $x \in \mathbb{R}^v$ in $\mathbb{R}^d$ through the matrix $W \in \mathbb{R}^{v \times d}$ and construct noiseless targets[6] by dotting a fixed $b \in \mathbb{R}^v$ with the sample $x$. The use of the matrix $W$ allows the model to have variable capacity ($d$) independent of the data set size. The samples $x \in \mathbb{R}^v$ and labels $b \in \mathbb{R}^v$ have power law dependence, whereas the matrix $W$ has entries distributed as $N(0, 1/d)$.

---

[6]With label noise, the scaling laws are the same as we report here, up to a scale at which the label noise is the limiting factor in the optimization and further increase of compute-budget or $d$ does not yield any benefits.

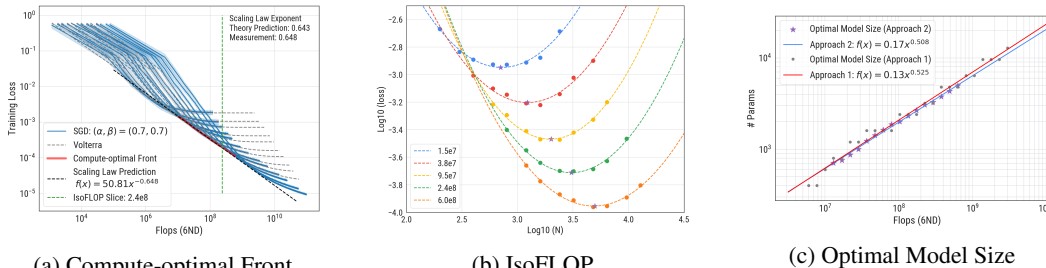

| (a) Compute-optimal Front | (b) IsoFLOP | (c) Optimal Model Size |

Figure 3: **Compute-Optimal Front in Phase II-III boundary.** (a) The Volterra equations perfectly captures the training dynamics of SGD when model-parameter count ranges from $d = 200 \to 12800$. (b) We apply IsoFLOP approach [24] to our toy model to extract the optimal-compute front: (compute-optimal loss) (highlighted in red in (a)) and the optimal model size: (compute-optimal model size) (scattered in purple in (c)). Power-law fitting compute-optimal front gives a measurement of the scaling law exponent 0.648 (vs. theoretical prediction 0.643 in Table 2). In (c), we power-law fit the relation between compute and (empirical) optimal model size via Approach 1 and 2 used in [24]: $d^\star \asymp \mathfrak{f}^{0.508}$ and $d^\star \asymp \mathfrak{f}^{0.525}$, resp. (vs. theory, $d^\star \asymp \mathfrak{f}^{0.5}$). See Sec. J for details.

**Assumption 1** (Data and labels, $\alpha$ and $\beta$). *The samples $x \in \mathbb{R}^v$ are distributed according to $(x_j) \sim j^{-\alpha} z_j$ for all $1 \le j \le v$ and $\{z_j\}_{j=1}^v \sim N(0,1)$. The labels are scalars constructed by dotting the sample $x$ with a signal $b \in \mathbb{R}^v$ whose entries $(b_j) = j^{-\beta}$.*

Without the random matrix $W$, the $\alpha, \beta$ are related to what is known in the literature as source and capacity conditions [10, 11, 20, 36]. For a detailed comparison of the parameters and related work, see Section A and Table 3.

The dimensions we consider throughout are always such that $v \ge Cd$ for $C > 1$. Throughout both $v$ and $d$ need to be large, but for some choices of $\alpha$ and $\beta$, the $v$ will need to be comparable to $d$.

**Definition 1.1** (Admissible $v$ and $d$). *We assume that $v \ge Cd$ with $C > 1$ and $v, d \to \infty$. Above the high-dimensional line, which is when $2\alpha > 1$, we suppose $v/d \to r \in (1, \infty) \cup \{\infty\}$.[7] On the other hand, below the high-dimensional line $(2\alpha < 1)$ we limit $v$ to be $v/d \to r \in (1, \infty)$.[8]*

One can rewrite the expression in (4) using the convenient form:

$$\min_{\theta \in \mathbb{R}^d} \left\{ \mathscr{P}(\theta) = \langle D(W\theta - b), (W\theta - b) \rangle \right\}, \quad \text{where } D = \text{diag}(j^{-2\alpha}) \in \mathbb{R}^{v \times v}. \tag{5}$$

**Algorithmic set-up.** To solve the minimization problem in (5), we use one-pass SGD with mini-batches of size $B$ (independent of $d$)[9] and constant learning rate $\gamma > 0$: letting $\theta_0 = 0$, we iterate

$$\text{drawing } \{x_r^i\}_{i=1}^B \text{ fresh iid samples and } \theta_{r+1} = \theta_r - \gamma \sum_{i=1}^B W^T x_r^i \left[ \langle W^T x_r^i, \theta_r \rangle - \langle x_r^i, b \rangle \right]. \tag{6}$$

The learning rate and batch size will need to satisfy a condition to ensure convergence (Prop. 2.1).

**Main goal.** Under this setup, our main goal is to characterize the compute-optimal frontier. Precisely, we want to find the parameter count exponent $\xi$ and scaling law exponent $\eta$, such that,

$$d^\star(\mathfrak{f}) \asymp \mathfrak{f}^\xi \quad \text{and} \quad \mathscr{P}\left( \tfrac{\mathfrak{f}}{d^\star B}; d^\star \right) \asymp \mathfrak{f}^{-\eta}.$$

**Notation.** We use $\mathscr{P}(\theta_r) = \mathscr{P}(r)$ when we want to emphasize the iteration counter $r$. We say $\mathscr{A}(r, v, d) \sim A(r, v, d)$ for functions $\mathscr{A}(r, v, d), A(r, v, d) > 0$ if for every $\varepsilon > 0$ and for all admissible $v$ and $d$, there exists an $r_0, d_0$ such that for all $d > d_0$ and $r \ge r_0$

$$(1 - \varepsilon) A(r, v, d) \le \mathscr{A}(r, v, d) \le (1 + \varepsilon) A(r, v, d).$$

---

[7]In fact, we may take $v = \infty$ for $2\alpha > 1$.

[8]Indeed one can, in the former case, take $d \le v \le d^{1/(1-2\alpha)}$, but for simplicity of presentation we focus on the proportional regime when $2\alpha < 1$.

[9]One can study batch size $B$ growing with $d$, but for simplicity we let $B = 1$. Thus we only consider $B$ independent of $d$ setting.

Table 1: **Large $d$ behavior of the forcing function and kernel function.** See Sec. H for proofs.

| **Function** | $^*\Gamma(x)$ is the Gamma function |
|---|---|
| $\mathscr{F}_0(r) \asymp d^{-2\alpha+\max\{0,1-2\beta\}}$ | |
| $\mathscr{F}_{pp}(r) \sim (2\alpha)^{-1} \times \Gamma\left(\frac{\beta}{\alpha} - \frac{1}{2\alpha} + 1\right) \times (2\gamma B \times r)^{-(1+\beta/\alpha)+1/(2\alpha)}$ | |
| $\mathscr{F}_{ac}(r) \leq \begin{cases} C \times \mathscr{F}_0(r), & \text{if } 2\beta > 1, 2\alpha < 1 \\ 0, & \text{if } 2\beta < 1 \end{cases}$ $\quad$ for $C > 0$, independent of $d$ | |
| If $2\beta > 1, 2\alpha > 1$, $\mathscr{F}_{ac}(r) \sim \left(\sum_{j=1}^{v} j^{-2\beta}\right)(2\alpha)^{-1}\Gamma\left(1 - \frac{1}{2\alpha}\right) \times (2\gamma B \times r)^{-1+1/(2\alpha)} \times d^{-1}$ | |
| $\mathscr{K}_{pp}(r) \sim (2\alpha)^{-1} \times \Gamma\left(2 - \frac{1}{2\alpha}\right) \times (2\gamma B \times r)^{-2+1/(2\alpha)}$ | |

We write $\asymp$ if the upper and lower bounds hold with some constants $c, C$ in place of $1 \mp \varepsilon$ respectively and $\lesssim, \gtrsim$ if only one inequality holds.

## 2 Learning Dynamics of SGD

Compute-optimal curves (3) for the random features model (4) rely on accurate predictions for the learning trajectory of SGD. Similar to the works of [33, 35], we show that the expected loss under SGD satisfies a convolution-type Volterra equation (for background on Volterra equations, see Section C.3)

$$\mathbb{E}[\mathscr{P}(\theta_r) \,|\, W] = \underbrace{\mathscr{F}(r)}_{\text{grad. descent}}^{\text{forcing func.}} + \underbrace{\mathscr{K} * \mathbb{E}\left[\mathscr{P}(\theta_r) \,|\, W\right]}_{\text{SGD noise}}, \text{ where } (\mathscr{K} * f)(r) = \sum_{s=0}^{r-1} \mathscr{K}(r-1-s)f(s). \quad (7)$$

The forcing function $\mathscr{F}(r)$ and kernel function $\mathscr{K}(r)$ are explicit functions of the matrix $\hat{K} = D^{1/2}WW^T D^{1/2}$, where $D = \mathrm{Diag}(j^{-2\alpha}, 1 \leq j \leq v)$, and $\Gamma \subset \mathbb{C}$ a contour enclosing the spectrum of $\hat{K} \in [0, 1]$,

$$\mathscr{F}(r) \overset{\text{def}}{=} \frac{-1}{2\pi i} \oint_\Gamma \langle (\hat{K} - z)^{-1}(D^{1/2}b), (D^{1/2}b)\rangle (1 - 2\gamma Bz + \gamma^2 B(B+1)z^2)^r \, \mathrm{d}z$$

$$\text{and} \quad \mathscr{K}(r) \overset{\text{def}}{=} \frac{-1}{2\pi i}\mathrm{Tr}\left(\oint_\Gamma (\hat{K} - z)^{-1}z^2(1 - 2\gamma Bz + \gamma^2 B(B+1)z^2)^r \, \mathrm{d}z\right). \quad (8)$$

Intuitively, the forcing function is gradient descent on the random features model and the kernel function is the excess risk due to 1 unit of SGD noise.

**Deterministic equivalent.** The forcing function $\mathscr{F}(r)$ and kernel function $\mathscr{K}(r)$ are random functions depending on the random matrix $\hat{K}$. Indeed, it is the *resolvent of $\hat{K}$*, $(\hat{K} - z)^{-1}$, which plays a significant role in $\mathscr{F}$ and $\mathscr{K}$. We remove this randomness from the expression by using a deterministic equivalent – a technique from random matrix theory.

Formally, we define the deterministic equivalent for the resolvent of $\hat{K}$, denoted by $\mathcal{R}(z)$, implicitly via a fixed point equation

$$m(z) \overset{\text{def}}{=} \frac{1}{1 + \frac{1}{d}\sum_{j=1}^{v} \frac{j^{-2\alpha}}{j^{-2\alpha}m(z)-z}} \quad \text{where} \quad \mathcal{R}(z) = \mathrm{Diag}\left(\frac{1}{j^{-2\alpha}m(z)-z} : 1 \leq j \leq v\right). \quad (9)$$

This deterministic equivalent $\mathcal{R}(z)$ is viewed, roughly, as $\mathbb{E}_W[(\hat{K} - z)^{-1}] \approx \mathcal{R}(z)$; though it is not formally the expectation over $W$. By replacing the resolvent of $\hat{K}$ with $\mathcal{R}(z)$, there exists a

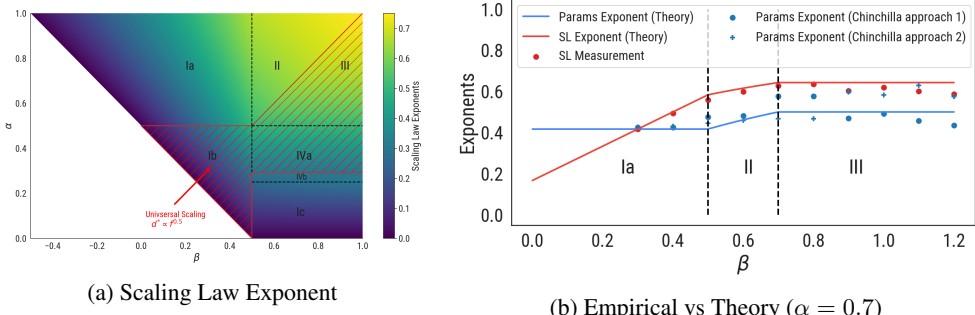

(a) Scaling Law Exponent

(b) Empirical vs Theory ($\alpha = 0.7$)

Figure 4: **(a) Scaling Law Exponents.** The heatmap displays scaling law exponents ($\eta$) in the $(\alpha, \beta)$-plane. Hatched lines represent region with universal scaling behavior, $d^\star \asymp \mathfrak{f}^{0.5}$, independent of $(\alpha, \beta)$. **(b) Exponent Measurements.** Compare empirical exponents (following [24]; see Sec. J for details) to theoretical predictions, traversing the phase diagram horizontally at $\alpha = 0.7$ from Phases Ia $\to$ II $\to$ III as $\beta \uparrow$.

deterministic function $\mathcal{P}(r)$ which solves a convolution Volterra equation, matching (7):

$$\mathcal{P}(r) = \underbrace{\mathcal{F}(r)}_{\text{grad. descent}}^{\text{forcing func.}} + \underbrace{(\mathcal{K} * \mathcal{P})(r)}_{\text{SGD noise}} \tag{10}$$

where $\quad \mathcal{F}(r) \stackrel{\text{def}}{=} \frac{-1}{2\pi i} \oint_\Gamma \langle (\mathcal{R}(z)(D^{1/2}b), (D^{1/2}b) \rangle (1 - 2\gamma Bz + \gamma^2 B(B+1)z^2)^r \, \mathrm{d}z \tag{11}$

and $\quad \mathcal{K}(r) \stackrel{\text{def}}{=} \gamma^2 B \cdot \mathrm{Tr}\left( \frac{-1}{2\pi i} \oint_\Gamma \mathcal{R}(z)z^2 (1 - 2\gamma Bz + \gamma^2 B(B+1)z^2)^r \, \mathrm{d}z \right). \tag{12}$

The solution to the Volterra equation with deterministic equivalent (10) numerically exactly matches the training dynamics of SGD, see Fig. 3. A discussion of the deterministic equivalent for $(\hat{K} - z)^{-1}$ can be found in Sec. E. All our mathematical analysis will be for the deterministic equivalents, going forward.[10] The derivation of the Volterra equation for the expected loss can be found in Sec. B.

An immediate consequence of (10) is that for convolution Volterra equations bounded solutions occur if and only if the forcing function is bounded and the *kernel norm* $\|\mathcal{K}\| \stackrel{\text{def}}{=} \sum_{s=0}^{\infty} \mathcal{K}(s) < 1$. This directly translates into a sufficient condition on the batch size and learning rate of SGD.

**Proposition 2.1** (Sufficient conditions on learning rate and batch). *Suppose learning rate $\gamma$ and batch $B$ satisfy $\|\mathcal{K}\| < 1$ and $\gamma(B+1) < 2$. Then $\mathcal{P}(r)$ is bounded.*

**Remark 2.1.** *Below the line $2\alpha = 1$, the kernel norm diverges with $v$ for fixed constant $\gamma$, and so we must take $\gamma \to 0$ to ensure bounded solutions. Thus, provided $\gamma \sim v^{2\alpha - 1}$, then*

$$\|\mathcal{K}\| \sim \frac{\gamma}{2} \sum_{j=1}^{v} j^{-2\alpha} \sim \frac{\gamma}{2(1 - 2\alpha)} v^{1 - 2\alpha} \quad \text{is order 1}.$$

*Thus, the kernel norm, $\|\mathcal{K}\|$, is always constant order for all $\alpha$.*

The batch $B$ and $\gamma$ can depend on $d$. For simplicity, we only consider $B$ order 1 in this work. For a proof of the necessary and sufficient conditions on $\gamma$ and $B$, see Prop. C.2, and see Cor. G.1 for the asymptotic on $\|\mathcal{K}\|$.

The Volterra equation in (10) can be analyzed to give a more explicit formula for $\mathcal{P}$ (see Section C.3.2 for proof).

---

[10]There is good numerical evidence that the deterministic equivalent captures all interesting features of the PLRF. There is a vast random matrix theory literature on making precise comparisons between resolvents and their deterministic equivalents. It seems a custom analysis will be needed for this problem, given the relatively high precision required, and we do not attempt to resolve this mathematically here.

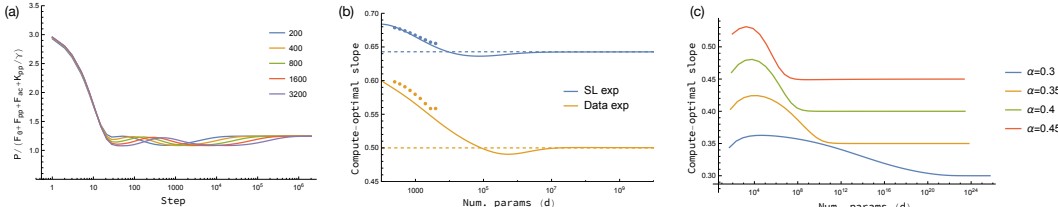

Figure 5: **Finite-size effects.** **(a)** The ratio of the exact solution of eq. (10) to the estimate in eq. (17) is bounded by constants for all $r$, confirming the validity of eq. (17); shown here is $(\alpha, \beta) = (0.7, 1.2)$. **(b)** For non-asymptotic $d$, the estimate in eq. (17) (solid curves) predicts both the magnitudes and trends of the measured exponents of the empirical compute-optimal frontier (points), shown here for $(\alpha, \beta) = (0.7, 1.2)$ computed using Approach 0 (see Appendix J) to capture the instantaneous slope; the dashed lines show the asymptotic exponents from Table 2. **(c)** The finite-size behavior relaxes to the asymptotic predictions over horizons whose length can grow exceedingly large, especially in the vicinity of the phase transition, shown here for $\beta = 0.7$ approaching the Phase 4a→4b boundary.

**Theorem 2.1** (Approximation solution for $\mathcal{P}$). *Suppose $\gamma$ and $B$ are at most half the convergence threshold and $2\alpha + 2\beta > 1$, $\alpha > \frac{1}{4}$.[11] There exists an $M > 0$ large and a constant $C = C(\alpha, \beta, M)$, independent of $d$, so that for all admissible $v$ and $d$, for all $\gamma B r > M$,*

$$\mathcal{F}(r) + (\mathcal{K} * \mathcal{F})(r) \leq \mathcal{P}(r) \leq \mathcal{F}(r) + C \times (\mathcal{K} * \mathcal{F})(r). \tag{13}$$

*The convolution $\mathcal{K} * \mathcal{F}$ further simplifies*

$$\tilde{c} \times \left( \mathcal{F}(r) + \frac{1}{\gamma B} \cdot \mathcal{K}(r) \right) \leq (\mathcal{K} * \mathcal{F})(r) \leq \tilde{C} \times \left( \overbrace{\mathcal{F}(r)}^{\text{forcing func.}} + \frac{1}{\gamma B} \cdot \overbrace{\mathcal{K}(r)}^{\text{kernel func.}} \right). \tag{14}$$

*for some constants $\tilde{c} = \tilde{c}(\alpha, \beta, M)$ and $\tilde{C} = \tilde{C}(\alpha, \beta, M) > 0$ independent of $d$.*

**Remark 2.2.** *If we were to run gradient descent instead of SGD (i.e., $\gamma$ small), then we would only have the forcing term, that is, $\mathcal{P}(r) = \mathcal{F}(r)$. The measurable effect of SGD comes from the second term that contains the kernel function. For this reason, we refer to SGD noise as $\frac{1}{\gamma B} \cdot \mathcal{K}(r)$.*

In light of (13) and (14), we have trapped the training loss between the sum of $\mathcal{F}$ and $\mathcal{K}$, so it suffices now to understand the forcing and kernel functions.

### 2.1 Forcing function and kernel function

We decompose the forcing function (11), $\mathcal{F}$, and the kernel function, (12), $\mathcal{K}$, into

$$\mathcal{F}(r) = \mathcal{F}_0(r) + \mathcal{F}_{ac}(r) + \mathcal{F}_{pp}(r) + \text{errors}_{\mathcal{F}}(r) \quad \text{and} \quad \mathcal{K}(r) = \mathcal{K}_{pp}(r) + \text{errors}_{\mathcal{K}}(r). \tag{15}$$

Each term is explicit and has an asymptotic equivalence (when $1 \lesssim \gamma B r \lesssim d^{2\alpha}$) given by

$$\mathcal{F}_i(r, d), \mathcal{K}_{pp}(r, d) \sim c \times r^{-\tau} \times d^{-\sigma} \quad \text{for some constants } c, \tau, \sigma > 0 \text{ (see Table 1).} \tag{16}$$

The two error terms are such that for large $d$ with $1 \lesssim \gamma B r \lesssim d^{2\alpha}$,

$$|\text{errors}_{\mathcal{F}}(r)| \leq C \times (\mathcal{F}_0(r) + \mathcal{F}_{ac}(r) + \mathcal{F}_{pp}(r)) \quad \text{and} \quad |\text{errors}_{\mathcal{K}}(r)| \leq C \times \mathcal{K}_{pp}(r),$$

for some constant $C > 0$. For $\gamma B r \gtrsim d^{2\alpha}$, the forcing function $\mathcal{F}(r) \asymp \mathcal{F}_0(r)$, the limiting risk value. The terms arise from different parts of the spectrum of the deterministic equivalent for $\hat{K}$ (see Fig. 6).

1. *Point mass at 0:* $\mathcal{F}_0(0) = \mathcal{F}_0(r)$ is the limiting value of $\mathcal{P}(r) \asymp d^{-2\alpha + \max\{0, 1-2\beta\}}$ as $r \to \infty$. It occurs because the loss is irreducible ($d < v$), that is a component of the target is not in the image of the RF model (or equivalently that $\hat{K}$ has a kernel).

---

[11] In spite of Theorem 2.1 holding only for $\alpha > \frac{1}{4}$, we expect this to hold for all $2\alpha + 2\beta > 1$ as supported numerically. When $\alpha < \frac{1}{4}$, the kernel function stops being power law and, as a result, requires a different set of tools to prove the result.

2. *Aligned features:* The function $\mathcal{F}_{pp}(r)$ represents gradient descent on the components of features which are aligned to the underlying population features. Indeed, if we ran gradient descent on the population loss *without* a random features map (or a diagonal $W$), this would be the loss curve.

3. *Distorted features:* The function $\mathcal{F}_{ac}(r)$ is the result of feature distortion, where the matrix $W$ leads to an embedding where a small component of the leading features is distributed across many different eigenmodes. These are still solvable, and given enough compute these will eventually be used, but they are much slower to solve.

4. *Aligned kernel:* $\mathcal{K}_{pp}(r)$ is the excess risk due to 1 unit of SGD noise, which is then solved according to population gradient descent.

Out of brevity, we relegate the exact definitions of $\mathcal{F}_0$, $\mathcal{F}_{pp}$, $\mathcal{F}_{ac}$, and $\mathcal{K}_{pp}$ and all proofs of the asymptotics in Table 1 and analyses of the functions to Section F, G, and H.

## 3 The 4 Phases

We now put together a coherent picture of the effect of different choices of $\alpha$ (data complexity) and $\beta$ (target complexity) and their impact on the compute-optimal frontier. By Theorem 2.1, we estimate

$$\mathcal{P}(r,d) \asymp \mathcal{F}_{pp}(r) + \mathcal{F}_{ac}(r) + \mathcal{F}_0(r) + \tfrac{1}{\gamma B}\mathcal{K}_{pp}(r). \tag{17}$$

Explicitly, we show that the functional form, or *scaling law* for the PLRF model is

$$\mathcal{P}(r,d) \asymp \underbrace{r^{-\sigma_1}}_{\mathcal{F}_{pp}(r)} + \underbrace{d^{-\tau_1}}_{\mathcal{F}_0(r)} + \underbrace{d^{-\tau_2}r^{-\sigma_2}}_{\mathcal{F}_{ac}(r)} + \underbrace{r^{-\sigma_3}}_{\tfrac{1}{\gamma B}\mathcal{K}_{pp}(r)} , \text{ where } \sigma_i, \tau_i > 0 \text{ and explicit, see Table 1.} \tag{18}$$

Fig. 5a. shows empirically that this equivalence of $\mathcal{P}(r)$ is quite good. The first two terms $\mathcal{F}_{pp}(r)$ (i.e., $r^{-\sigma_1}$) and $\mathcal{F}_0(r)$ (i.e., $d^{-\tau_1}$) are often called in the literature as time and model bottlenecks respectively. The functional form using *only* these two components, i.e., $\mathcal{P}(r,d) \asymp r^{-\sigma_1} + d^{-\tau_1}$ were used to find compute-optimal exponents in [24, 9] and in concurrent work [28]. Because the functional form considered in [9, 28] are missing the two other terms (cross-term $\mathcal{F}_{ac}$ and SGD noise $\mathcal{K}_{pp}$), the compute-optimal curves in [9, 28] agree only in Phase Ia with our results. Importantly, we show that the cross-term, i.e., $\mathcal{F}_{ac}(r)$, and SGD noise, $\mathcal{K}_{pp}(r)$, can indeed affect the compute-optimal exponents. (The cross-term also appeared in concurrent work on ridge regression [18].)

The **4 distinct phases** (see Fig. 2a) decompose the $(\alpha, \beta)$-plane based on the shape of the loss curve $\mathcal{P}(r)$, that is, which of the distinct components of the forcing function (i.e., $\mathcal{F}_0, \mathcal{F}_{pp}, \mathcal{F}_{ac},$) and/or kernel function (i.e., $\mathcal{K}_{pp}$) dominate the loss curve at a given iteration $r$. See Table 2 for loss description in each phase. Cartoon pictures of the different features of the loss curves are shown in Fig. 2. For each phase, we derive a compute-optimal curve in Section 3.1.

The high-dimensional line, which occurs where $2\alpha = 1$, distinguishes the phases where the $v$-dimension can be big and independent of $d$ (Phase Ia, II, III, $2\alpha > 1$) and the phases where $d$ and $v$ must be related to each other (Phase Ib, Ic, IVa, IVb, $2\alpha < 1$). When $2\alpha + 2\beta < 1$, the loss does not exhibit any power-law decay as the limit level stops going to 0 as $d \to \infty$ (purely as a consequence of having selected the regime $v > d$). Moreover, there exists an interesting critical point $\alpha = \beta = \frac{1}{2}$ where all the parts of the forcing function and kernel mix and interact with each other. The behavior of the loss at the pentuple point (see Fig 2a) we leave for future research. Across each of the phase boundaries the compute-optimal curves are continuous, but not necessarily differentiable; in contrast, $d^\star$ is discontinuous across some phase boundaries.

### 3.1 Compute-optimal Curves

To simplify the computations for compute-optimal curves, we introduce the following curve

$$\tilde{\mathcal{P}}(r) \overset{\text{def}}{=} \max\left\{\mathcal{F}_{pp}(r), \mathcal{F}_{ac}(r), \mathcal{F}_0(r), \tfrac{1}{\gamma B}\mathcal{K}_{pp}(r)\right\}. \tag{19}$$

The function $\tilde{\mathcal{P}}(r,d)$ achieves the same power law behavior as the original compute-optimal curve $\mathcal{P}(r,d)$ (i.e., the slope of the compute-optimal curve is correct) and deviates from the true curve by an absolute constant (independent of $d$ and $\mathfrak{f}$). Note that some of the terms in the max function

Table 2: **Loss description for $\mathcal{P}(r)$ and compute-optimal curves for $\tilde{\mathcal{P}}(\frac{\mathfrak{f}}{d\cdot B}, d)$ across the 4 phases**.

| | Loss $\mathcal{P}(r)$ | Trade off | Compute-optimal Curves |
|---|---|---|---|
| **Phase I** | $\mathcal{F}_{pp}(r) + \mathcal{F}_0(r)$ | $\mathcal{F}_{pp} = \mathcal{F}_0$ | **Ia** $\;\tilde{\mathcal{P}}^\star_{\text{Phase Ia}}(\mathfrak{f}) \asymp \mathfrak{f}^{\left(\frac{1}{2\alpha+1}-1\right)(1+\beta/\alpha-1/(2\alpha))}$ 
 $d^\star_{\text{Phase Ia}} \asymp \mathfrak{f}^{1/(2\alpha+1)}$ 

 **Ib** $\;\tilde{\mathcal{P}}^\star_{\text{Phase Ib}}(\mathfrak{f}) \asymp \mathfrak{f}^{\frac{1}{2}-\alpha-\beta}$ 
 $d^\star_{\text{Phase Ib}} \asymp \mathfrak{f}^{\frac{1}{2}}$ 

 **Ic** $\;\tilde{\mathcal{P}}^\star_{\text{Phase Ic}}(\mathfrak{f}) \asymp \mathfrak{f}^{\frac{\alpha(2\alpha+2\beta-1)}{\alpha(2\beta-3)-2\beta+1}}$ 
 $d^\star_{\text{Phase Ic}} \asymp \mathfrak{f}^{\frac{1-2(\alpha+\beta)}{2(\alpha(2\beta-3)-2\beta+1)}}$ |
| **Phase II** | $\mathcal{F}_{pp}(r) + \mathcal{F}_{ac}(r)$ 
 $+\mathcal{F}_0(r)$ | $\mathcal{F}_{pp} = \mathcal{F}_{ac}$ | $\tilde{\mathcal{P}}^\star_{\text{Phase II}}(\mathfrak{f}) \asymp \mathfrak{f}^{-\frac{2\alpha+2\beta-1}{2(\alpha+\beta)}}$ 
 $d^\star_{\text{Phase II}} \asymp \mathfrak{f}^{(\beta/\alpha)/(1+\beta/\alpha)}$ |
| **Phase III** | $\mathcal{F}_{ac}(r) + \mathcal{F}_0(r)$ 
 $+\frac{1}{\gamma B}\mathcal{K}_{pp}(r)$ | $\frac{1}{\gamma B}\mathcal{K}_{pp} = \mathcal{F}_{ac}$ | $\tilde{\mathcal{P}}^\star_{\text{Phase III}}(\mathfrak{f}) \asymp \mathfrak{f}^{(1-4\alpha)/(4\alpha)}$ 
 $d^\star_{\text{Phase III}} \asymp \mathfrak{f}^{1/2}$ |
| **Phase IV** | $\mathcal{F}_{pp}(r) + \mathcal{F}_0(r)$ 
 $+\frac{1}{\gamma B}\mathcal{K}_{pp}(r)$ | **IVa** $\;\frac{1}{\gamma B}\mathcal{K}_{pp} = \mathcal{F}_0$ 

 **IVb** $\;\frac{1}{\gamma B}\mathcal{K}_{pp} = \mathcal{F}_{pp}$ | $\tilde{\mathcal{P}}^\star_{\text{Phase IVa}}(\mathfrak{f}) \asymp \mathfrak{f}^{-\alpha}$ 
 $d^\star_{\text{Phase IVa}} \asymp \mathfrak{f}^{1/2}$ 

 $\tilde{\mathcal{P}}^\star_{\text{Phase IVb}}(\mathfrak{f}) \asymp \mathfrak{f}^{\frac{(1-2\alpha)(2\alpha+2\beta-1)}{(2(2\alpha\beta+\alpha-2\beta))}}$ 
 $d^\star_{\text{Phase IVb}} \asymp \mathfrak{f}^{(\alpha-\beta)/(2\alpha\beta+\alpha-2\beta)}$ |

(19) should be taken to be 0 when not defined for the different phases. Therefore, we derive the compute-optimal curves by solving the problem

$$\min_d \tilde{\mathcal{P}}\left(\tfrac{\mathfrak{f}}{d\cdot B}, d\right), \quad \text{and if } d^\star(\mathfrak{f}) \overset{\text{def}}{=} \arg\min_d \tilde{\mathcal{P}}\left(\tfrac{\mathfrak{f}}{d\cdot B}, d\right),$$

$$\text{then the compute-optimal curve is} \quad \tilde{\mathcal{P}}^\star(\mathfrak{f}) \overset{\text{def}}{=} \tilde{\mathcal{P}}\left(\tfrac{\mathfrak{f}}{d^\star(\mathfrak{f})\cdot B}, d^\star(\mathfrak{f})\right). \tag{20}$$

See Table 2 for the exact expressions for $d^\star(\mathfrak{f})$ and the compute-optimal curve $\tilde{\mathcal{P}}^\star(\mathfrak{f})$ for each phase. A more detailed description with proofs can be found in Section C.4 and Section D.

Now to derive $d^\star$ and $\tilde{\mathcal{P}}^\star$, we recall that the functions $\mathcal{F}_0, \mathcal{F}_{pp}, \mathcal{F}_{ac}, \mathcal{K}_{pp}$ take the form $c \times d^{-\sigma_i} \times (\frac{\mathfrak{f}}{d\cdot B})^{-\tau_i}$ (16). Therefore, $\tilde{\mathcal{P}}^\star(\mathfrak{f}/(d^\star \cdot B), d^\star)$ must occur at corner point where two functions meet. These tradeoffs between the two functions for which the compute-optimal point occurs are shown in Fig. 2 and Table 2.

**Details for each phase.** We describe the qualitative and quantitative properties of compute-optimal curves for each phase. These are broken down into *model constrained* (Phase I, II) vs. *algorithm constrained* (Phase III, IV), i.e., whether the PLRF model or SGD is the constraining feature.

**Phase Ia, Ib, Ic. Capacity constrained.** Phase Ia ($2\alpha > 1, 2\beta < 1$), Ib ($2\alpha < 1, 2\beta < 1, 2(\alpha + \beta) > 1$), Ic are characterized by having the simplest loss description, $\mathcal{P}(r) \asymp \mathcal{F}_{pp}(r) + \mathcal{F}_0(r)$. Here the SGD noise is irrelevant and one would have the same loss (and thus compute-optimal curve) as gradient descent on the population loss. Compute optimality is characterized by training the model completely (to its limit loss) and choosing the model parameter count large enough so that at the end of training, the smallest loss is attained. The main distinctions between Phase Ia, Ib, Ic are the model capacities (i.e., $\mathcal{F}_0(r, d) = d^{-2\alpha+1-2\beta}$ in Ia, Ib, and $\mathcal{F}_0(r, d) = d^{-2\alpha}$ in Ic) and the dependence of dimension in the learning rate due to Ib,Ic being below the high-dimensional line. Consequently, while the qualitative features of the loss curve are the same for Ia, Ib, and Ic, the actual values of the compute-optimal curve vary across the different regions. Notably, in Phase Ib, the compute-optimal parameter is $d^\star = \mathfrak{f}^{1/2}$ and it is independent of $\alpha$ and $\beta$.

**Phase II. Distortion constrained.** Phase II ($2\alpha > 1, 2\beta > 1, \beta < \alpha$) has a loss curve where the $\mathcal{F}_{ac}$ is important, that is, $\mathcal{P}(r) \asymp \mathcal{F}_{pp}(r) + \mathcal{F}_{ac}(r) + \mathcal{F}_0(r)$. The $\mathcal{F}_{ac}$ term becomes the dominant term after running for some intermediate amount of time $d^c$; in fact it is compute-optimal to stop at this point, and then select the number of model parameters so to minimize the loss with this early stopping criterion. It transpires that across *all* phases, it *never* pays to solve through the $\mathcal{F}_{ac}$ part of the loss curve – it is always better to just increase the number of model parameters.

**Phase III. SGD frustrated, distortion constrained.** In this phase ($2\alpha > 1, 2\beta > 1, \beta > \alpha$), SGD noise is important. The loss curve is $\mathcal{P}(r) \asymp \mathcal{F}_{ac}(r) + \mathcal{F}_0(r) + \frac{1}{\gamma B}\mathcal{K}_{pp}(r)$. Notably, in this phase, the compute-optimal parameter is $d^\star(\mathfrak{f}) = \mathfrak{f}^{1/2}$, which is independent of $\alpha$ and $\beta$. PLRF that fall within this phase have the same scaling law regardless of data complexity and target complexity. Moreover, the tradeoff occurs, like in Phase II, once the optimizer reaches the $\mathcal{F}_{ac}$-dominated part of the loss curve. Unlike in Phase II, the optimization is slowed by SGD noise ($\mathcal{K}_{pp}$) leading up to that point. We note that there is a dimension-independent burn-in period required for SGD noise to dominate, and for small numerical simulations, one may actually observe an ($\mathcal{F}_{pp}, \mathcal{F}_{ac}$) tradeoff.

**Phase IV. SGD frustrated, capacity constrained.** Like Phase III, SGD noise is important. The SGD algorithm in Phase IV will be distinguished from gradient descent. As one approaches the high-dimensional line ($2\alpha = 1$) in Phase III, the $\mathcal{F}_{ac}(r)$ disappears. It becomes too small relative to $\mathcal{F}_{pp}$ and $\mathcal{K}_{pp}$. Moreover at the high-dimensional line, $\mathcal{F}_{pp}$ becomes important again. Thus, the loss curve in Phase IV (a and b) look like $\mathcal{P}(r, d) \asymp \mathcal{F}_{pp}(r, d) + \mathcal{F}_0(r, d) + \frac{1}{\gamma B}\mathcal{K}_{pp}(r, d)$. The distinction between Phase IVa ($1 - \frac{1}{\sqrt{2}} < \alpha < 0.5, 2\beta > 1$) and Phase IVb ($\frac{1}{4} < \alpha < 1 - \frac{1}{\sqrt{2}}, 2\beta > 1$) is where the compute-optimal tradeoff occurs. It changes from $\mathcal{K}_{pp} = \mathcal{F}_0$ (Phase IVa) to $\mathcal{F}_{pp} = \mathcal{K}_{pp}$ (Phase IVb). In particular it can be (Phase IVb) the SGD noise is so large that increasing the model parameter count is compute-optimal. We note that in this phase $d$ must be taken very large (in particular larger than we could numerically attain) to get quantitative agreement between the exponents and theory.

**Other observations.** In Phase III, Ib, and IVa, the optimal parameter $d^\star = \mathfrak{f}^{1/2}$ (see dashed lines in Fig. 4a). These phases, taken together, encompass a large section of the $(\alpha, \beta)$-phase plane. This suggests that there is a potential universal scaling law. Moreover using 1 A100-GPU-day of compute, one reaches scales of $d$ where the observed exponents in the scaling laws – SGD, the theoretically-derived Volterra equation eq. (10), and the equivalence of $\mathcal{P}(r)$ eq. (17) – are still changing (see Fig. 5b and c). This serves as a potential warning for empirically derived scaling laws. Additionally, although we have identified the lower-left of the phase diagram ($\alpha + \beta < 1/2$) as "no power-law", this designation relies on the assumption $v > d$, which could be relaxed to interesting effect in more realistic (e.g. non-linear) models.

**Compute-optimal learning rate and batch.** Previously, we have used $B = 1$ and the maximal learning rate allowed. One can also consider finding the compute-optimal curves with respect to batch size and learning rate, i.e., find $d^\star, \gamma^\star, B^\star$ such that

$$(d^\star, \gamma^\star, B^\star) \in \arg\min_{d, \gamma, B} \in \arg\min \mathcal{P}(\tfrac{\mathfrak{f}}{dB}, \gamma, d) \quad \text{s.t. } \gamma B < 1 \text{ and } \|\mathcal{K}_{pp}\| < 1. \quad (21)$$

In Section I, we show that $\mathcal{P}(\frac{\mathfrak{f}}{dB}, \gamma, d)$ is monotonic in $B$ and therefore $B = 1$ is optimal. Similarly for $\gamma$, in Phases I, II, III, the loss $\mathcal{P}(\frac{\mathfrak{f}}{dB}, \gamma, d)$ is monotonic and thus the maximally stable learning rate is optimal. For Phase IV, this is not true. There does exist an optimal $\gamma^\star$ (with $B = 1$),

$$\gamma^\star \asymp \mathfrak{f}^{\frac{4\alpha(\alpha-\beta)}{4\alpha\beta + 2\alpha + 2\beta - 1}}, \quad d^\star(\mathfrak{f}) \asymp \mathfrak{f}^{\frac{2\alpha + 2\beta - 1}{4\alpha\beta + 2\alpha + 2\beta - 1}}, \quad \text{and} \quad \mathcal{P}^\star(\mathfrak{f}) \asymp \mathfrak{f}^{\frac{-2\alpha(2\alpha + 2\beta - 1)}{4\alpha\beta + 2\alpha + 2\beta - 1}},$$

where the tradeoff occurs between $\frac{1}{\gamma}\mathcal{K}_{pp}(r) = \mathcal{F}_0(r)$ and Phase IVa and IVb collapse to a single phase. This is proven in Proposition I.1.

**Conclusion.** We analyze a simple three parameter model, PLRF, and derive deterministic expressions for the training dynamics (see Volterra equation (10)). We then extract compute-optimal scaling laws for large $d$. We identify 4 phases (+3 subphases) in the $(\alpha, \beta)$-phase plane, corresponding to different compute-optimal curve/loss behaviors. These phase boundaries are determined by the relative importance of model capacity (Phase I, IV), poor embedding of the features (Phase II, III), and the noise produced by the SGD algorithm (Phase III, IV). The latter suggesting that another stochastic algorithm might change the compute-optimal curve; we leave this interesting direction to future research. We also show evidence of a universal scaling law which we also leave for future research to explore.

## Acknowledgments and Disclosure of Funding

C. Paquette is a Canadian Institute for Advanced Research (CIFAR) AI chair, Quebec AI Institute (MILA) and a Sloan Research Fellow in Computer Science (2024). C. Paquette was supported by a Discovery Grant from the Natural Science and Engineering Research Council (NSERC) of Canada, NSERC CREATE grant Interdisciplinary Math and Artificial Intelligence Program (INTER-MATH-AI), Google research grant, and Fonds de recherche du Québec – Nature et technologies (FRQNT) New University Researcher's Start-Up Program. Research by E. Paquette was supported by a Discovery Grant from the Natural Science and Engineering Council (NSERC). Additional revenues related to this work: C. Paquette has 20% part-time employment at Google DeepMind.

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

# 4+3 Phases of Compute-Optimal Neural Scaling Laws

## Supplementary material

**Broader Impact Statement.** The work presented in this paper is foundational research and it is not tied to any particular application. The set-up is on a simple well-studied random features model with synthetic data and solved using a commonly deployed algorithm – stochastic gradient descent. We present (theoretical) compute-optimal curves for this model. The results are theoretical and we do not anticipate any direct ethical and societal issues. We believe the results will be used by machine learning practitioners and we encourage them to use it to build a more just, prosperous world.

**Outline of the paper.** The remainder of the article is structured as follows: in Section A, we provide additional related work. In Section B, we derive the convolution-type Volterra equation for the expected risk under SGD, (7). In Section C.1, we analyze the Volterra equation under the deterministic equivalent. A discussion on the convergence threshold for $\mathcal{P}(r)$ including a necessary and sufficient condition for bounded solutions of (10) (Proposition C.2) and a proof of Proposition 2.1 are provided in Section C.2. Some background on Volterra equations and their solutions are provided in Section C.3.1 followed by the proof of Theorem 2.1 in Section C.3.2. We finish this section with a detailed description and proofs for the risk curves in all phases, Section C.4. Section D is devoted to deriving and proving the compute-optimal curves in Table 2. We follow this by Section E which analyzes the deterministic equivalent for the resolvent of $\hat{K}$. Here we examine the spectrum of $\hat{K}$ from a random matrix point of view. In particular, in this section, we prove estimates on the fixed point equation, $m$, see eq. (9). We then give explicit descriptions of the components of the forcing function, $\mathcal{F}_0, \mathcal{F}_{pp}, \mathcal{F}_{ac}$, as contour integrals and show that the error terms error$_\mathcal{F}$ are small, see Section F. We do the same with the kernel function $\mathcal{K}$ and kernel norm in Section G. In Section H, we derive the asymptotic formulas for the components of the forcing and kernel functions (see Table 1) used in the compute-optimal curve derivations. Finally, we end with some additional numerical experiments (and their experimental setups) as well as detailed descriptions of the different approaches to estimating the exponents in the scaling law and optimal model-parameter, Section J.

## Contents

# A   Additional Related Work.

Table 3: **Comparison of the source/capacity parameters across various related work.** We note this table is taken from Table 1 in [DLM24][1] with the addition of [Lin+24][5]. We note that both [DLM24] and [Lin+24] appeared concurrently with this article.

|  | **This work** | [DLM24][1] | [Bahri+21][2] | [MRS22][3] | [BAP24][4] | [Lin+24][5] |
|---|---|---|---|---|---|---|
| **Input dimension** | $d$ | $d$ | $d$ | $M$ | $M$ | $M$ |
| **# of features** | $v$ | $p$ | $P$ | $N$ | $N$ | - |
| **Iterations/samples** | $r$ | $n$ | $D$ | $T$ | $P$ | $N$ |
| **Capacity** | $2\alpha$ | $\alpha$ | $1+\alpha$ | $1+\alpha$ | $b$ | $a$ |
| **Source** | $\frac{2\alpha+2\beta-1}{4\alpha}$ | $r$ | $\frac{1}{2}(1-\frac{1}{\alpha})$ | $\frac{1}{2}(1-\frac{1}{\alpha})$ | $\frac{a-1}{2b}$ | $\frac{b-1}{2a}, b>1$ |
| **Target decay (in $L_2$)** | $\alpha+\beta$ | $\alpha r+\frac{1}{2}$ | $0$ | $0$ | $\frac{a}{2}$ | $\frac{b}{2}$ |

[1] [DLM24] L. Defilippis, B. Loureiro, T. Misiakiewicz. *Dimension-free deterministic equivalents for random feature regression.* 2024

[2] [Bahri21] Y. Bahri, D. Dyer, J. Kaplan, J. Lee, and U. Sharma. *Explaining neural scaling laws.* 2024.

[3] [MRS22] A. Maloney, D.A. Roberts, J. Sully. *A solvable model of neural scaling laws*

[4] [BAP24] B. Bordelon, A. Atanasov, C. Pehlevan. *A dynamical model of neural scaling laws.* 2024.

[5] [Lin24] L. Lin, J. Wu, S. Kakade, P. Barlett, J.D. Lee. *Scaling Laws in Linear Regression: Compute, Parameters, and Data.* 2024

**Random features and random matrices.**   This paper uses random matrix theory to analyze a random features problem, which in statistical language would be the generalization error of the one-pass SGD estimator. Random matrix theory has played an increasingly large role in machine learning (see for [14] for a modern introduction).

The input we need for our random matrix analysis is for sample covariance matrices with power-law population covariance (i.e. linear random features). The analysis of sample covariance matrices precedes their usage in machine learning (see e.g. [6]), but to our knowledge, a detailed study of all parts of the spectrum of sample covariance matrices with power-law population covariances has not appeared before. The narrower study of ridge regression has been extensively investigated (see for e.g. [4, 12]), and the concurrent work [18] provides a complete analysis of the ridge regression problem when $2\alpha > 1$. However, while (roughly speaking) ridge regression requires analysis of resolvent $(A - z\text{Id})^{-1}$ statistics for negative spectral parameter $z$ (which might be very close to 0), the analysis in this work requires resolvent statistics for essentially all $z$.

There is a larger theory of nonlinear random features regression, mostly in the case of isotropic random features. Including nonlinearities in this model is a natural future direction; for isotropic random features with *proportional dimension asymptotics* this has been explored in works such as [31] and for some classes of anisotropic random features in [32, 17, 29] (we mention that lots of the complexity of the analysis of power-law random features arises from the analysis of the self-consistent equations – indeed the self-consistent equations we use date to [40], but the analysis of these equations may still be novel). This strongly motivates non-proportional scalings (which would be inevitable in power-law random features with nonlinearities); in the isotropic case, the state of the art is [25].

**Random features regression, 'source/capacity' conditions, and SGD.**   A large body of kernel regression and random features literature is formulated for "source/capacity" conditions, which are power-law type assumptions that contain the problem setup here, when $2\alpha > 1$ (the low-dimensional regime). For convenience, we record the parameters

$$\alpha_{\text{source}} = 2\alpha \quad \text{and} \quad r = \frac{2\alpha + 2\beta - 1}{4\alpha}.$$

Table 4: **(Nonexhaustive) Comparison of sample-complexity results.** Let $\rho \overset{\text{def}}{=} 2\alpha + 2\beta - 1$. We use our Phases with $n$ = sample size, $d$ = parameters. We will include derivation of these results in the appendix of our paper. [DLM24][1] can also be done with RR+optimal-ridge, which yields same in Phase Ia, but different in Phase II/III. [VPF21][9] obtain $\mathscr{P} \ll n^{-\min\{1/(2\alpha),(2\alpha+2\beta-1)/(2\alpha)\}}$, that is, they capture the $\mathcal{F}_{pp}$, but not $\mathcal{F}_{ac}$. The *minimax* optimal rates never achieve any of the rates (always worse), which can be connected to overly conservative, small stepsizes. For derivation of the minimax rates, we used Cor. 2 from [DB18][7]. [Lin+24][5] requires label noise order 1 and also a very small learning rate.

| | | **This work** | [DLM24][1] |
|---|---|---|---|
| **Algorithm** | | one-pass SGD | RR + $O(1)$-ridge |
| | Phase Ia | $\Theta(n^{-\rho/(2\alpha)} \vee d^{-\rho})$ | same as ours |
| **Risk,** | Phase II | $\Theta(n^{-\rho/(2\alpha)} \vee d^{-1}n^{-1+1/(2\alpha)} \vee d^{-2\alpha})$ | same as ours |
| $\mathscr{P}(n)$ | Phase III | $\Theta(n^{-2+1/(2\alpha)} \vee d^{-1}n^{-\frac{2\alpha-1}{2\alpha}} \vee d^{-2\alpha})$ | $\Theta(n^{-2} \vee d^{-1}n^{-\frac{2\alpha-1}{2\alpha}} \vee d^{-2\alpha})$ |

| | | Minimax optimal[678] | [Lin+24][5] |
|---|---|---|---|
| **Algorithm** | | one-pass SGD, very small stepsize | one-pass SGD, very small stepsize |
| | Phase Ia | $O\!\left(n^{-\rho/(2\alpha+2\beta)}\right)$ | $\Theta(d^{-\rho} + n^{-\rho/(2\alpha)} + \min\{\frac{d}{n}, n^{-1+1/(2\alpha)}\})$ |
| **Risk,** | Phase II | $O\!\left(n^{-\rho/(2\alpha+2\beta)}\right)$ | does not cover |
| $\mathscr{P}(n)$ | Phase III | $O\!\left(n^{-4\alpha/(4\alpha+1)}\right)$ | does not cover |

[6] Carratino, Rudi, Rosasco. *Learning with sgd and random features.* 2018

[7] Dieuleveut and Bach. *Nonparametric stochastic approximation with large stepsizes.* 2016.

[8] Pillaud-Vivien, Rudi, Bach. *Statistical optimality of SGD on hard learning problems through multiple passes.* 2018.

[9] Varre, Pillaud-Vivien, Flammarion. *Last iterate convergence of SGD for least squares in the interpolation regime* 2021.

Here we have taken $r$ as the limit of those $r$'s for which the source/capacity conditions hold (see Table 3). We note that in this language $r$ is often interpreted as 'hardness' (lower is harder), and that $r \in (0, 0.5)$, $r \in (0.5, 1.0)$ and $r \in (1.0, \infty)$ correspond to 3 regimes of difficulty which have appeared previously (see the citations below); they are also precisely the 3 phases Ia, II, and III.

The authors of [37] establish generalization bounds for random feature regression with power-law structures in $2\alpha > 1$ case. These bounds were sharpened in [16] and extended in [18] (see also the earlier [10] which shows kernel ridge regression is 'minimax optimal' under various 'source-capacity conditions'); we give a comparison to these bounds in Table 4, but we note that the problem setup we have is not captured by 'minimax optimality' (in particular minimax optimality is worst-case behavior over a problem class, and our problem setup is not worst-case for the traditional source/capacity conditions)

We note that this paper is fundamentally about computation, but the novel mathematical contributions could also be recast in terms of generalization bounds of one-pass SGD, some of which are new. The work of [11] compares SGD to kernel ridge regression, showing that one-pass SGD can attain the same bounds as kernel ridge regression and hence is another minimax optimal method (again under 'source-capacity' conditions). See also [20] which considers similar statements for SGD with iterate averaging and [36] for similar statements for multipass SGD; see also [38, 19] which also prove the single-batch versions of these. These bounds attain the minimax-optimal rate, which are worse than the rates attained in this paper (see Table 4 for a comparison).

**Dynamical deterministic equivalents, Volterra equations and ODEs.** Using the deterministic equivalents for random matrix resolvents [23], we in turn derive deterministic equivalents for the risk curves of SGD.

The method of analysis of the risk curves in this paper is by formulation of a convolutional Volterra equation [33]. This can be equivalently formulated as a system of coupled difference equations for weights of the SGD residual in the observed data covariance, which generalizes beyond the

least-squares context [13]; in isotropic instances, this simplifies to a finite-dimensional family of ODES [1]. This can also be generalized to momentum SGD methods [34] and large batch SGD methods [27]. Convolution-Volterra equations are convenient tools, as they are well-studied parts of renewal theory [2] and branching process theory [3].

Another method of analysis is dynamical mean field theory. The closest existing work to this one in scientific motivations is [8], which uses this technique. This formally can be considered as a type of Gaussian process approximation, but for a finite family of observables ("order parameters"). In instances of one-pass SGD (including in anisotropic cases), this is rigorously shown to hold in [21]. The analysis of the resulting self-consistent equations is nontrivial, and [8] does some of this analysis under simplifying assumptions on the structure of the solutions of these equations.

Besides these works, there is a large theory around generalization error of SGD. The work of [42] gives a direct analysis of risks of SGD under "source/capacity" type assumptions which formally capture the $F_{pp}$ parts of the Phase Ia/II loss curves. The risk bounds of [43] give non-asymptotic estimates which again reproduce tight estimates for the $F_{pp}$ parts of the loss (note that to apply these bounds to this case, substantial random matrix theory needs to be worked out first); see also concurrent work [28] where some of this is done.

# B  Derivation of Volterra equation

We begin by deriving a Volterra equation for the population loss $\mathscr{P}(\theta)$, (5). Fix a quadratic $q : \mathbb{R}^d \to \mathbb{R}$, i.e., a function $q(x) = x^T A x + e^T x + c$ for fixed matrix $A \in \mathbb{R}^{d \times d}$, vector $e \in \mathbb{R}^d$ and constant $c \in \mathbb{R}$. Let us consider the filtration $\mathcal{F}_r = \sigma(W, \theta_0, \ldots, \theta_r)$ which conditions on $W$ and the past iterates. Then we have from Taylor theorem,

$$\mathbb{E}[q(\theta_{r+1}) - q(\theta_r) \mid \mathcal{F}_r] = \mathbb{E}[\langle \nabla q(\theta_r), \theta_{r+1} - \theta_r \rangle \mid \mathcal{F}_r] + \frac{1}{2}\mathbb{E}[\langle \nabla^2 q, (\theta_{r+1} - \theta_r)^{\otimes 2} \rangle \mid \mathcal{F}_r]. \quad (22)$$

We need to plug the expression for SGD in (6) into the above. The first thing we observe is that we need moments of Gaussians via Wick's formula: for fixed vectors $v_i \in \mathbb{R}^v$, $i = 1, 2, 3, 4$, and $x \sim N(0, D)$

$$\mathbb{E}_x[x\langle x, v_1 \rangle] = \mathbb{E}_x[\text{Tr}(x^T x)]v_1 = D v_1$$
$$\mathbb{E}_x[\langle x, v_1 \rangle \langle x, v_2 \rangle \langle x, v_3 \rangle \langle x, v_4 \rangle] = \langle D, v_1 \otimes v_2 \rangle \langle D, v_3 \otimes v_4 \rangle + \langle D, v_1 \otimes v_3 \rangle \langle D, v_2 \otimes v_4 \rangle \quad (23)$$
$$+ \langle D, v_1 \otimes v_4 \rangle \langle D, v_2 \otimes v_3 \rangle.$$

Here we recall that the $(v \times v)$-matrix $D \stackrel{\text{def}}{=} \text{Diag}(j^{-2\alpha} : 1 \leq j \leq v)$. Using these moment calculations, we can compute explicitly each of the terms in (22).

**Gradient term.**   First, we consider the gradient term in (22). A simple computation yields

$$\mathbb{E}[\langle \nabla q(\theta_r), \theta_{r+1} - \theta_r \rangle \mid \mathcal{F}_r] = -\gamma \langle \nabla q(\theta_r), \mathbb{E}[\sum_{j \in B_r} W^T x^j (\langle W^T x^j, \theta_r \rangle - \langle x^j, b \rangle) \mid \mathcal{F}_r] \rangle$$
$$= -\gamma B \langle \nabla q(\theta_r), W^T D W \theta_r - W^T D b \rangle. \quad (24)$$

**Quadratic term.**   We now turn to the quadratic term in (22). Supposing $x$ and $\hat{x}$ are independent samples,

$$\frac{1}{2}\mathbb{E}[\langle \nabla^2 q, (\theta_{r+1} - \theta_r)^{\otimes 2} \rangle \mid \mathcal{F}_r]$$
$$= \frac{\gamma^2}{2}\mathbb{E}\left[\left\langle \nabla^2 q, \left(\sum_{j \in B_r} W^T x^j [\langle W^T x^j, \theta_r \rangle - \langle x^j, b \rangle]\right) \otimes \left(\sum_{k \in B_r} W^T x^k [\langle W^T x^k, \theta_r \rangle - \langle x^k, b \rangle]\right) \right\rangle \mid \mathcal{F}_r\right]$$
$$= \frac{\gamma^2}{2}\mathbb{E}\left[\sum_{j \in B_r} \left\langle \nabla^2 q, \left(\sum_{j \in B_r} W^T x^j [\langle W^T x^j, \theta_r \rangle - \langle x^j, b \rangle]\right)^{\otimes 2} \right\rangle \mid \mathcal{F}_r\right]$$
$$+ \frac{\gamma^2}{2}\mathbb{E}\left[\left\langle \nabla^2 q, 2 \sum_{j < k \in B_r} \left(W^T x^j [\langle W^T x^j, \theta_r \rangle - \langle x^j, b \rangle]\right) \otimes \left(W^T x^k [\langle W^T x^k, \theta_r \rangle - \langle x^k, b \rangle]\right) \right\rangle \mid \mathcal{F}_r\right].$$
$$(25)$$

Continuing, we have

$$\frac{1}{2}\mathbb{E}[\langle \nabla^2 q, (\theta_{r+1} - \theta_r)^{\otimes 2}\rangle \,|\, \mathcal{F}_r]$$

$$= \frac{\gamma^2 B}{2}\mathbb{E}\big[\langle \nabla^2 q, (W^T x[\langle W^T x, \theta_r\rangle - \langle x, b\rangle])^{\otimes 2}\rangle \,|\,\mathcal{F}_r\big]$$

$$+ \frac{\gamma^2 B(B-1)}{2}\mathbb{E}\big[\langle \nabla^2 q, (W^T x[\langle W^T x, \theta_r\rangle - \langle x, b\rangle]) \otimes (W^T \hat{x}[\langle W^T \hat{x}, \theta_r\rangle - \langle \hat{x}, b\rangle])\rangle \,|\,\mathcal{F}_r\big] \qquad (26)$$

$$= \frac{\gamma^2 B}{2}\mathbb{E}\big[(\langle W^T x, \theta_r\rangle - \langle x, b\rangle)^2 \langle \nabla^2 q, (W^T x)^{\otimes 2}\rangle \,|\,\mathcal{F}_r\big]$$

$$+ \frac{\gamma^2 B(B-1)}{2}\mathbb{E}\big[\langle \nabla^2 q, (W^T x[\langle W^T x, \theta_r\rangle - \langle x, b\rangle]) \otimes (W^T \hat{x}[\langle W^T \hat{x}, \theta_r\rangle - \langle \hat{x}, b\rangle])\rangle \,|\,\mathcal{F}_r\big].$$

Let $\nabla^2 q = \sum_{j=1}^v v_j \otimes \tilde{v}_j$. Now we note the following

$$(\langle W^T x, \theta_r\rangle - \langle x, b\rangle)^2 \langle \nabla^2 q, (W^T x)^{\otimes 2}\rangle$$

$$= (x^T W \nabla^2 q W^T x)[\langle W^T x, \theta_r\rangle^2 - 2\langle W^T x, \theta_r\rangle\langle x, b\rangle + \langle x, b\rangle^2] \qquad (27)$$

$$= \sum_j \langle x, W v_j\rangle\langle x, W\tilde{v}_j\rangle[\langle W^T x, \theta_k\rangle^2 - 2\langle x, W\theta_r\rangle\langle x, b\rangle + \langle x, b\rangle^2].$$

This, after taking expectations, is in the form for us to apply the moment computations in (23). Using these moments, we get the following expression:

$$\mathbb{E}\Big[\sum_j \langle x, W v_j\rangle\langle x, W\tilde{v}_j\rangle[\langle W^T x, \theta_r\rangle^2 - 2\langle x, W\theta_r\rangle\langle x, b\rangle + \langle x, b\rangle^2] \,|\,\mathcal{F}_r\Big]$$

$$= \langle \nabla^2 q, W^T D W\rangle\|D^{1/2}(W\theta_r - b)\|^2$$

$$\qquad + 2\sum_j \big[\langle D, W v_j \otimes W\theta_r\rangle\langle D, W\tilde{v}_j \otimes W\theta_r\rangle - \langle D, W v_j \otimes W\theta_r\rangle\langle D, W\tilde{v}_j \otimes b\rangle$$

$$\qquad + \langle D, W v_j \otimes b\rangle\langle D, W\tilde{v}_j \otimes b\rangle - \langle D, W\tilde{v}_j \otimes W\theta_r\rangle\langle D, W v_j \otimes b\rangle\big] \qquad (28)$$

$$= \langle \nabla^2 q, W^T D W\rangle\|D^{1/2}(W\theta_k - b)\|^2$$

$$\qquad + 2\sum_j \langle D, W v_j \otimes (W\theta_r - b)\rangle\langle D, W\tilde{v}_j \otimes (W\theta_r - b)\rangle.$$

Now we simplify the 2nd term in the summand

$$\sum_j \langle D, W v_j \otimes (W\theta_r - b)\rangle\langle D, W\tilde{v}_j \otimes (W\theta_r - b)\rangle$$

$$= \sum_j \langle W^T D, v_j \otimes (W\theta_r - b)\rangle\langle W^T D, \tilde{v}_j \otimes (W\theta_r - b)\rangle \qquad (29)$$

$$= \sum_j \sum_{i,n,\ell,m} (W^T D)_{ni} v_{jn} (W\theta_r - b)_i (W^T D)_{m\ell}\tilde{v}_{jm}(W\theta_r - b)_\ell$$

$$= 2\langle DW(\nabla^2 q)W^T D, (W\theta_r - b)^{\otimes 2}\rangle.$$

Moreover, as $x$ and $\hat{x}$ are independent, we see that

$$\mathbb{E}\big[\langle \nabla^2 q, (W^T x[\langle W^T x, \theta_r\rangle - \langle x, b\rangle]) \otimes (W^T \hat{x}[\langle W^T \hat{x}, \theta_r\rangle - \langle \hat{x}, b\rangle])\rangle \,|\,\mathcal{F}_r\big]$$

$$= \mathbb{E}\big[(W\theta_r - b)^T x x^T W \nabla^2 q W^T \hat{x}\hat{x}^T(W\theta_r - b) | \mathcal{F}_r\big] \qquad (30)$$

$$= (W\theta_r - b)^T D W \nabla^2 q W^T D(W\theta_r - b).$$

As a result, we deduce by combining (27), (28), (29), and (30) with (26) gives the following representation for the expected quadratic term

$$\frac{1}{2}\mathbb{E}[\langle \nabla^2 q, (\theta_{r+1} - \theta_r)^{\otimes 2}\rangle \,|\, \mathcal{F}_k] = \frac{\gamma^2 B}{2}\langle \nabla^2 q, W^T D W\rangle\|D^{1/2}(W\theta_r - b)\|^2$$

$$+ \frac{\gamma^2 B(B+1)}{2}\langle DW(\nabla^2 q)W^T D, (W\theta_r - b)^{\otimes 2}\rangle. \qquad (31)$$

**Volterra equation.**    Using the simplified gradient and quadratic terms, we can now state the expected change in any quadratic $q : \mathbb{R}^d \to \mathbb{R}$ evaluated at an iterate of SGD (6):

$$\mathbb{E}[q(\theta_{r+1}) - q(\theta_r) \,|\, \mathcal{F}_r] = -\gamma B \langle \nabla q(\theta_r), W^T D W \theta_r - W^T D b \rangle$$
$$+ \tfrac{\gamma^2 B}{2} \langle \nabla^2 q, W^T D W \rangle \| D^{1/2}(W\theta_r - b) \|^2 \qquad (32)$$
$$+ \tfrac{\gamma^2 B(B+1)}{2} \langle D W (\nabla^2 q) W^T D, (W\theta_r - b)^{\otimes 2} \rangle.$$

We can write $\mathbb{R}^v = \mathrm{Im}(W) \oplus W^\perp$. Thus, there exists $\check{b} \in \mathbb{R}^d$ and $\dot{b} \in \mathbb{R}^v$ such that one can write $b = W\check{b} + \dot{b}$, that is, something in the image of $W$ and something in the co-ker of $W$, i.e.,

$$b = W\check{b} + \dot{b}, \quad \text{where } W^T D \dot{b} = 0. \qquad (33)$$

> **One-step update formula.**  Using this observation, we have a formula for the expectation of the quadratic $q : \mathbb{R}^d \to \mathbb{R}$,
>
> $$\mathbb{E}[q(\theta_{r+1}) - q(\theta_r) \,|\, \mathcal{F}_r] = -\gamma B \langle \nabla q(\theta_r), W^T D W (\theta_r - \check{b}) \rangle$$
> $$+ \tfrac{\gamma^2 B}{2} \langle \nabla^2 q, W^T D W \rangle \| D^{1/2}(W\theta_r - b) \|^2 \qquad (34)$$
> $$+ \tfrac{\gamma^2 B(B+1)}{2} \langle W^T D W (\nabla^2 q) W^T D W, (\theta_r - \check{b})^{\otimes 2} \rangle.$$

We observe that all the terms on the right hand side of the above (34) involve the matrix $W^T D W \in \mathbb{R}^{d \times d}$. Consequently, let $(\lambda_j, w_j)$ for $j = 1, \ldots, d$ be the eigenvalue-eigenvector of $W^T D W$ with $\|w_j\| = 1$. Now define

$$\rho_j^2(r) \overset{\text{def}}{=} \langle w_j^{\otimes 2}, (\theta_r - \check{b})^{\otimes 2} \rangle, \quad \text{for all } j = 1, \ldots, d. \qquad (35)$$

We will write our Volterra equation in terms of $\rho_j$'s. Note we can express the loss $\mathscr{P}(\theta_r) = \| D^{1/2}(W\theta_r - b) \|^2$ by

$$\mathscr{P}(\theta_r) = \| D^{1/2}(W\theta_r - b) \|^2 = \sum_{j=1}^d \lambda_j^2 \rho_j^2(r) + \| D^{1/2}\dot{b} \|^2. \qquad (36)$$

We can now plug $\rho_j^2$ into (34). For this, we need to compute $\nabla \rho_j^2$ and $\nabla^2 \rho_j^2$:

$$\rho_j^2(r) = \langle w_j^{\otimes 2}, \theta_r - \check{b}^{\otimes 2} \rangle, \quad \nabla_\theta \rho_j^2(r) = 2 w_j \langle w_j, \theta_r - \check{b} \rangle, \quad \text{and} \quad \nabla^2 \rho_j^2(r) = 2 w_j \otimes w_j.$$

Then we have that

$$\mathrm{d}\rho_j^2(r) = -2\gamma B \langle w_j, \theta_r - \check{b} \rangle \langle w_j, W^T D W (\theta_r - \check{b}) \rangle$$
$$+ B\gamma^2 \langle w_j \otimes w_j, W^T D W \rangle \| D^{1/2}(W\theta_r - b) \|^2$$
$$+ B(B+1)\gamma^2 \langle W^T D W (w_j \otimes w_j) W^T D W, (\theta_r - \check{b})^{\otimes 2} \rangle \qquad (37)$$
$$= -2\gamma B \lambda_j \rho_j^2(r) + \gamma^2 B \lambda_j \| D^{1/2}(W\theta_r - b) \|^2 + \gamma^2 B(B+1) \lambda_j^2 \rho_j^2(r)$$

Using an integrating factor, we can implicitly solve this expression

$$\mathrm{d}\rho_j^2(k) = \left[ -2\gamma B \lambda_j + \gamma^2 B(B+1) \lambda_j^2 \right] \rho_j^2 + \gamma^2 B \lambda_j \| D^{1/2}(W\theta_k - b) \|^2, \qquad (38)$$

and thus, we have a discrete Volterra equation

$$\rho_j^2(r) = \rho_j^2(0)(1 - 2\gamma B \lambda_j + \gamma^2 B(B+1) \lambda_j^2)^r$$
$$+ \gamma^2 B \sum_{s=0}^{r-1} (1 - 2\gamma B \lambda_j + \gamma^2 B(B+1) \lambda_j^2)^{r-1-s} \lambda_j \| D^{1/2}(W\theta_s - b) \|^2. \qquad (39)$$

Let us define $\check{K} \overset{\text{def}}{=} W^T D W$. Using the expression in (36),

$$\mathbb{E}[\mathscr{P}(\theta_r) \,|\, W] = \sum_{j=1}^d \lambda_j \rho_j^2(0)(1 - 2\gamma B \lambda_j + \gamma^2 B(B+1) \lambda_j^2)^r + \| D^{1/2}\dot{b} \|^2$$
$$+ \sum_{j=1}^d \gamma^2 B \lambda_j^2 \sum_{s=0}^{r-1} (1 - 2\gamma B \lambda_j + \gamma^2 B(B+1) \lambda_j^2)^{r-1-s} \cdot \mathbb{E}[\mathscr{P}(\theta_s) \,|\, W].$$

Let us define the kernel

$$\mathscr{K}(r) \overset{\text{def}}{=} \gamma^2 B \sum_{j=1}^{d} \lambda_j^2 (1 - 2\gamma B \lambda_j + \gamma^2 B (B+1)\lambda_j^2)^r = \gamma^2 B \cdot \text{Tr}\big(\check{K}^2 (I - 2\gamma B\check{K} + \gamma^2 B(B+1)\check{K}^2)^r\big).$$

> **Discrete volterra equation for the loss for $\check{K} = W^T D W$.** Let $r$ be the number of iterates of SGD. Then
>
> $$\mathbb{E}[\mathscr{P}(\theta_r) \,|\, W] = \langle \check{K}(I - 2\gamma B\check{K} + \gamma^2 B(B+1)\check{K}^2)^r, (\theta_0 - \check{b})^{\otimes 2}\rangle + \|D^{1/2}\dot{b}\|^2$$
>
> $$+ \sum_{s=0}^{r-1} \mathscr{K}(r - 1 - s) \cdot \mathbb{E}[\mathscr{P}(\theta_s) \,|\, W],$$
>
> where $\mathscr{K}(s) = \gamma^2 B \sum_{j=1}^{d} \lambda_j^2 (1 - 2\gamma B\lambda_j + \gamma^2 B(B+1)\lambda_j^2)^s$ (40)
>
> $$= \gamma^2 B \times \text{Tr}\big(\check{K}^2(I - 2\gamma B\check{K} + \gamma^2 B(B+1)\check{K}^2)^s\big)$$
>
> and $\quad D = \text{Diag}(j^{-2\alpha} : 1 \le j \le v).$

We can also write (40) in terms of $\hat{K} \overset{\text{def}}{=} D^{1/2}WW^T D^{1/2}$. To see this, set $D^{1/2}W = V\sqrt{\Omega}U^T$ where $\check{K} = U\Omega U^T$ and $\hat{K} = V\Omega V^T$. Then we see that

$$\langle \text{poly}(\check{K})\check{K}, (\theta_0 - \check{b})^{\otimes 2}\rangle = \langle \text{poly}(\Omega)\Omega, (U(\theta_0 - \check{b})^{\otimes 2}\rangle$$

$$= \langle V\text{poly}(\Omega)V^T, (V\sqrt{\Omega}U(\theta_0 - \check{b}))^{\otimes 2}\rangle$$

$$= \langle \text{poly}(\hat{K}), (D^{1/2}W(\theta_0 - \check{b}))^{\otimes 2}\rangle.$$

> **Discrete volterra equation for the loss with $\hat{K} = D^{1/2}WW^T D^{1/2}$.** Let $r$ be the number of iterates of SGD. Then
>
> $$\mathbb{E}[\mathscr{P}(\theta_r) \,|\, W] = \langle (I - 2\gamma B\hat{K} + \gamma^2 B(B+1)\hat{K}^2)^r, (D^{1/2}(W\theta_0 - b))^{\otimes 2}\rangle + \|D^{1/2}\dot{b}\|^2$$
>
> $$+ \sum_{s=0}^{r-1} \mathscr{K}(r - 1 - s) \cdot \mathbb{E}[\mathscr{P}(\theta_s) \,|\, W],$$
>
> where $\mathscr{K}(s) = \gamma^2 B \sum_{j=1}^{d} \lambda_j^2 (1 - 2\gamma B\lambda_j + \gamma^2 B(B+1)\lambda_j^2)^s$
>
> $$= \gamma^2 B \times \text{Tr}\big(\hat{K}^2(I - 2\gamma B\hat{K} + \gamma^2 B(B+1)\hat{K}^2)^s\big)$$
>
> and $\quad D = \text{Diag}(j^{-2\alpha} : 1 \le j \le v).$
>
> (41)

## C   Analysis of Volterra Equation

From now on, we consider the setting where the initialization of SGD is $\theta_0 = 0$. Let us introduce the forcing function:

$$\mathscr{F}(r) \overset{\text{def}}{=} \langle (I - 2\gamma B\hat{K} + \gamma^2 B(B+1)\hat{K}^2)^r, (D^{1/2}(W\theta_0 - b))^{\otimes 2}\rangle + \|D^{1/2}\dot{b}\|^2 \quad (42)$$

and recall the kernel function $\mathscr{K}(s)$:

$$\mathscr{K}(s) = \gamma^2 B \cdot \text{Tr}\big(\hat{K}^2(I - 2\gamma B\hat{K} + \gamma^2 B(B+1)\hat{K}^2)^s\big).$$

While these representations are easy to see from the derivation of the Volterra equation, a more useful representation of the forcing function and the kernel function is through contour integrals over the spectrum of $\hat{K}$. With this in mind, let $\Gamma$ be a contour containing $[0, 1]$. Note that by the assumptions

on $\hat{K}$, the largest eigenvalue is normalized to be $1$; hence $\Gamma$ contains the spectrum of $\hat{K}$. Then the forcing function takes the form

$$\mathscr{F}(r) = \frac{-1}{2\pi i} \oint_\Gamma \langle (\hat{K} - z)^{-1}, (D^{1/2}b)^{\otimes 2} \rangle (1 - 2\gamma Bz + \gamma^2 B(B+1)z^2)^r \, \mathrm{d}z. \qquad (43)$$

and the kernel function

$$\mathscr{K}(r) = \gamma^2 B \cdot \mathrm{Tr}\left( \frac{-1}{2\pi i} \oint_\Gamma z^2 \big( (1 - 2\gamma Bz + \gamma^2 B(B+1)z^2)^r \big)(\hat{K} - z)^{-1} \, \mathrm{d}z \right). \qquad (44)$$

Then one can write the Volterra equation (41) as the forcing function plus a convolution with the kernel and the expected loss, *i.e.*,

$$\mathbb{E}[\mathscr{P}(\theta_r) \,|\, W] = \mathscr{F}(r) + \big( \mathscr{K} * \mathbb{E}[\mathscr{P}(\theta_s) \,|\, W] \big),$$

$$\text{where } (\mathscr{K} * \mathbb{E}[\mathscr{P}(\theta_s) \,|\, W])(r) = \sum_{s=0}^{r-1} \mathscr{K}(r - 1 - s) \mathbb{E}[\mathscr{P}(\theta_s) \,|\, W]. \qquad (45)$$

### C.1 Deterministic equivalent of the loss under SGD

The forcing functions $\mathscr{F}(r)$ and kernel function $\mathscr{K}(r)$ are random functions as they depend on the random matrix $W$. Moreover the expressions via contour integration show that both of these functions can be described in terms of the random matrix $\hat{K} = D^{1/2}WW^T D^{1/2}$. Indeed it is the resolvent of $\hat{K}$,

$$\mathscr{R}(\hat{K}, z) \overset{\text{def}}{=} (\hat{K} - z)^{-1},$$

which plays a significant role in $\mathscr{F}$ and $\mathscr{K}$ and thus in the expected loss $\mathbb{E}[\mathscr{P}(\theta_r) \,|\, W]$. To analyze the power law behavior of the expected loss, it would be helpful to remove the randomness in $\hat{K}$, i.e., $W$. We do this by finding a deterministic equivalent for the resolvent of $\hat{K}$, $\mathscr{R}(\hat{K}, z) = (\hat{K} - z)^{-1}$, using techniques from random matrix theory. Intuitively, we want to take the expectation over the random matrix $W$; though not formally true.

Formally, we define the deterministic equivalent for the resolvent $\mathscr{R}(\hat{K}, z)$, denoted by $\mathcal{R}(z)$ implicitly via a fixed point equation

$$m(z) \overset{\text{def}}{=} \frac{1}{1 + \frac{1}{d} \sum_{j=1}^v \frac{j^{-2\alpha}}{j^{-2\alpha}m(z) - z}} \quad \text{where} \quad \mathcal{R}(z) = \mathrm{Diag}\left( \frac{1}{j^{-2\alpha}m(z) - z} : 1 \le j \le v \right). \quad (46)$$

As mentioned earlier, this deterministic equivalent, $\mathcal{R}(z)$ can be viewed, roughly as,

$$\mathbb{E}_W[(\hat{K} - z)^{-1}] = \mathbb{E}_W[\mathscr{R}(\hat{K}, z)] \approx \mathcal{R}(z);$$

though it is not formally the expectation over $W$.

Using this deterministic expression for the resolvent of $\hat{K}$, we defined deterministic expressions for the forcing function via the contour representation of $\mathscr{F}(r)$ in (43)

$$\left( \begin{array}{c} \text{forcing function} \\ \text{deterministic equivalent} \end{array} \right) \mathcal{F}(r) \overset{\text{def}}{=} \frac{-1}{2\pi i} \oint_\Gamma \langle \mathcal{R}(z), (D^{1/2}b)^{\otimes 2} \rangle (1 - 2\gamma Bz + \gamma^2 B(B+1)z^2)^r \, \mathrm{d}z, \tag{47}$$

and the kernel function in (44)

$$\left( \begin{array}{c} \text{kernel function} \\ \text{deterministic equivalent} \end{array} \right) \mathcal{K}(r) \overset{\text{def}}{=} \gamma^2 B \cdot \mathrm{Tr}\left( \frac{-1}{2\pi i} \oint_\Gamma z^2 (1 - 2\gamma Bz + \gamma^2 B(B+1)z^2)^r \mathcal{R}(z) \, \mathrm{d}z \right). \tag{48}$$

Using the deterministic expressions for the forcing function $\mathcal{F}$ and kernel function $\mathcal{K}$, we define the deterministic function $\mathcal{P}(r) : \mathbb{N} \to \mathbb{R}$ as the solution to the (discrete) convolution-type Volterra equation:

$$\mathcal{P}(r) = \mathcal{F}(r) + (\mathcal{K} * \mathcal{P})(r), \quad \text{where } (\mathcal{K} * \mathcal{P})(r) = \sum_{s=0}^{r-1} \mathcal{K}(r - 1 - s)\mathcal{P}(s). \tag{49}$$

We note the similarity with the Volterra equation for SGD. We conjecture that the two processes are close: for $\{\theta_r\}$ the sequence of iterates generated by SGD with $\theta_0 = 0$ and any $\varepsilon > 0$,

$$(1 - \varepsilon) \leq \sup_{r \in \mathbb{N}} \left\{ \frac{\mathbb{E}[\mathscr{P}(\theta_r)|W]}{\mathcal{P}(r)} \right\} \leq (1 + \varepsilon),$$

for all admissible $v$, $d$ with probability going to 1 as $d \to \infty$.

We leave this for future research; we suspect it is true based upon existing of deterministic equivalence theory for random matrices and numerical evidence.

## C.2 Convergence threshold

A natural question is: for what choices of batch $B$ and learning rate $\gamma$ does $\mathcal{P}$ converge? To answer this, we introduce an additional quantity, the *kernel norm* defined as

$$\text{(kernel norm)} \quad \|\mathcal{K}\| \overset{\text{def}}{=} \sum_{s=0}^{\infty} \mathcal{K}(s). \tag{50}$$

**Proposition C.1** (Kernel norm). *The kernel norm is satisfies*

$$\|\mathcal{K}\| \sim \frac{\gamma}{2} \sum_{j=1}^{v} \frac{j^{-2\alpha}}{1 - \gamma j^{-2\alpha}}.$$

*If $2\alpha > 1$, then $v$ be taken to equal $\infty$, that is,*

$$\|\mathcal{K}\| \sim \frac{\gamma}{2} \sum_{j=1}^{\infty} \frac{j^{-2\alpha}}{1 - \gamma j^{-2\alpha}}$$

*In the case that $2\alpha < 1$, we have*

$$\|\mathcal{K}\| \sim \frac{\gamma}{2} \sum_{j=1}^{v} j^{-2\alpha} \sim \frac{\gamma}{2(1 - 2\alpha)} v^{1-2\alpha}.$$

*In all cases, we choose $\gamma$ so that the kernel norm is asymptotic to a strictly positive constant.*

A well-known result about convolution-type Volterra such as (49) is that the solution of convolution-type Volterra equation is bounded if and only the forcing function $\mathcal{F}(r)$ is bounded and the kernel norm $\|\mathcal{K}\| < 1$. This naturally leads to conditions for our specific forcing function and kernel function.

**Remark C.1** (Convergence threshold conditions.). *The forcing function $\mathcal{F}$ is bounded and the kernel norm $\|\mathcal{K}\| < 1$ for (47) and (48), respectively, if and only if*

$$\text{(i). } |1 - 2\gamma B\lambda_j + \gamma^2 B(B+1)\lambda_j^2| < 1, \text{ for all } \lambda_j \in [0,1] \text{ and (ii). kernel norm } \|\mathcal{K}\| < 1. \tag{51}$$

*The first term ensures that the forcing function of the Volterra equation in (49) goes to 0 (i.e., bounded) and the second condition is the same kernel norm bound. Moreover, we can think of condition (i). as the same condition needed for gradient descent to converge while the kernel norm is the effect of noise from SGD.*

*We also note that in light of Proposition C.1 the kernel norm does not involve the batch size $B$. Therefore the condition $\|\mathcal{K}\| < 1$ only places a condition on the learning rate (see below).*

We now state necessary/sufficient conditions on the batch size and learning rate (51) (Proof of Prop. 2.1).

**Proposition C.2** (Necessary/Sufficient conditions on learning rate and batch size). *The learning rate, $\gamma > 0$ and batch size, $B > 0$, satisfy*

$$\|\mathcal{K}\| < 1, \quad \gamma(B+1) < 2. \tag{52}$$

*if and only if the solution $\mathcal{P}(r)$ to the convolution-type Volterra equation (10) is bounded.*

*Proof.* From (51), we need that $|1 - 2\gamma B\lambda_j + \gamma^2 B(B+1)\lambda_j^2| < 1$, for all $\lambda_j \in [0, 1]$. For this, we consider two cases.

First, suppose that $1 - 2\gamma Bx + \gamma^2 B(B+1)x^2 < 1$ for all $x \in [0, 1]$. We have that

$$-2\gamma Bx + \gamma^2 B(B+1)x^2 < 0 \implies x(-2\gamma B + \gamma^2 B(B+1)x) < 0. \tag{53}$$

The roots are precisely $x = 0$ and $x = \frac{2}{\gamma(B+1)}$. If $2/(\gamma(B+1)) > 1$, then the inequality in (53) always holds. Therefore, we need that $\gamma(B+1) < 2$.

Now suppose $-1 + 2\gamma Bx - \gamma^2 B(B+1)x^2 < 1$. Then we have

$$-2 + 2\gamma Bx - \gamma^2 B(B+1)x^2 < 0, \quad \text{for all } x \in [0, 1]. \tag{54}$$

The roots of the left-hand side are complex and thus the inequality always holds. $\square$

**Remark C.2.** *Below the high-dimensional line, $2\alpha < 1$, the kernel norm diverges with $v$ for fixed constant $\gamma$, and so we must take $\gamma \to 0$ to ensure bounded solutions. Furthermore, with $\gamma \to 0$ (at any rate depending on $v$) we have the asymptotic equivalence*

$$\|\mathcal{K}\| \sim \frac{\gamma}{2} \sum_{j=1}^{v} j^{-2\alpha} \sim \frac{\gamma}{2(1-2\alpha)} v^{1-2\alpha}.$$

*For a proof of the asymptotic for $\|\mathcal{K}\|$, see Corollary G.1.*

**Remark C.3.** *Similar results hold for the expected SGD loss (via the Volterra equation (45)) by replacing $\|\mathcal{K}\|$ with $\|\mathscr{K}\|$.*

## C.3 Simplification of the Volterra Equation

While convolution-type Volterra equation such as (49) are quite nice and well studied in the literature (e.g., [22]), we need an approximation of the solution to it to have better understanding of compute-optimal curves. In this section, we show that we can bound (above and below) $\mathcal{P}(r)$ by a constant multiple of the forcing function $\mathcal{F}$ and kernel function $\mathcal{K}$.

### C.3.1 Background on Volterra equations

To do this, we need some background on general convolution-type Volterra equations of the form:

$$P(t) = f(t) + (K * P)(t), \quad \text{where } (K * P)(t) = \sum_{s=0}^{t} K(s)P(t-s). \tag{55}$$

where $f(t)$ is a non-negative forcing function and $K(t)$ is a *monotonically decreasing* non-negative kernel function.

Let us define $K^{*n} \stackrel{\text{def}}{=} \underbrace{(K * K * \ldots * K * K)}_{n \text{ times}}(t)$, the $n$-fold convolution of $K$ where $K^{*1} = K(t)$.

Under mild assumptions such as $\|K\| = \sum_{t=0}^{\infty} K(t) < 1$ and the forcing function $f$ is bounded, then there exists a unique (bounded) solution $P(t)$ to (55) and the solution is given by repeatedly convolving the forcing function with $K$ (see, e.g., [22, Theorem 3.5]),

$$P(t) = f(t) + \sum_{j=1}^{\infty} K^{*j} * f(t)$$

$$= f(t) + (K * f)(t) + (K * K * f)(t) + (K * K * K * f)(t) + \ldots.$$

This representation of the solution to (55) enables us to get good bounds on $P(t)$. First, we state and prove a lemma attributed to Kesten's Lemma [3, Lemma IV.4.7].

**Lemma C.1** (Kesten's Lemma). *Suppose the kernel function $K$ is positive and monotonically decreasing and $\|K\| < \infty$. Moreover suppose for some $\varepsilon > 0$, there exists a $T(\varepsilon) > 0$ such that*

$$\sum_{s=0}^{t} K(s)K(t-s) \leq 2(1+\varepsilon)\|K\|K(t) \quad \text{for all } t \geq T. \tag{56}$$

*Then for all $n \geq 0$,*

$$\sup_t \left\{ \frac{K^{*(n+1)}(t)}{K(t)} \right\} \leq \left( \frac{K(0)}{K(T)} + 1 \right) (2\|K\|(1+\varepsilon))^n.$$

*Proof.* Define

$$a_n \overset{\text{def}}{=} \sup_{t>0} \frac{K^{*n}(t)}{K(t)(2\|K\|)^{n-1}}.$$

Then $a_1 = 1$, and we are trying to prove

$$a_n \leq \left( \frac{K(0)}{K(T)} + 1 \right) (1+\varepsilon)^{n-1}.$$

By definition of the convolution, we have that

$$\frac{K^{*(n+1)}(t)}{(2\|K\|)^n} = \sum_{s=0}^{t} \frac{K(s)K(t-s)}{2\|K\|} \times \frac{K^{*n}(t-s)}{K(t-s)(2\|K\|)^{n-1}} \leq a_n \times \sum_{s=0}^{t} \frac{K(s)K(t-s)}{2\|K\|}.$$

By the hypothesis (56),

$$\text{for } t \geq T, \qquad \frac{K^{*(n+1)}(t)}{(2\|K\|)^n} \leq a_n(1+\varepsilon)K(t). \tag{57}$$

For $t < T$, we have

$$\frac{K^{*(n+1)}(t)}{(2\|K\|)^n} = \sum_{s=0}^{t} \frac{K(s)K^{*(n)}(t-s)}{(2\|K\|)^n}$$

$$(K \text{ monotonically decreasing}) \quad \leq K(0) \sum_{s=0}^{t} \frac{K^{*n}(t-s)}{(2\|K\|)^n}$$

$$\leq K(0) \frac{\|K^{*n}\|}{(2\|K\|)^n} = K(0) \left( \tfrac{1}{2} \right)^n,$$

where the last equality follows by the equality $\|K^{*n}\| = \|K\|^n$, [22, Theorem 2.2(i)].

In conclusion, by monotonicity, we have that

$$\frac{K^{*(n+1)}(t)}{(2\|K\|)^n K(t)} \leq \begin{cases} \frac{K(0)}{2^n K(T)}, & t \leq T \\ a_n(1+\varepsilon), & t \geq T. \end{cases}$$

Hence we have that

$$a_{n+1} \leq \frac{K(0)}{K(T)2^n} + (1+\varepsilon)a_n.$$

Developing the recursion,

$$a_{n+1} \leq \sum_{j=0}^{n-1} \frac{(1+\varepsilon)^j K(0)}{K(T)} \times \left( \frac{1}{2} \right)^{n-j} + (1+\varepsilon)^n \leq (1+\varepsilon)^n \left[ \frac{1}{1-1/2} - 1 \right] \frac{K(0)}{K(T)} + (1+\varepsilon)^n.$$

The result is proven. $\qquad \square$

**Remark C.4.** *If the assumption (56) holds only for $\hat{T} > t > T$, then the statement of Lemma C.1 still holds with*

$$\sup_{t \leq \hat{T}} \left\{ \frac{K^{*(n+1)}(t)}{K(t)} \right\} \leq \left( \frac{K(0)}{K(T)} + 1 \right) (2\|K\|(1+\varepsilon))^n.$$

We now give a non-asymptotic bound for the general convolution-type Volterra equation.

**Lemma C.2** (Non-asymptotic Volterra bound). *Let $K$ and $f$ be non-negative functions. Suppose $K$ is monotonically decreasing and for some $\varepsilon > 0$, there exists a $T(\varepsilon) > 0$ such that*

$$\sum_{s=0}^{t} K(s)K(t-s) \leq 2(1+\varepsilon)\|K\|K(t), \quad \text{for all } t \geq T.$$

*Moreover, suppose the convergence threshold condition $2(1+\varepsilon)\|K\| < 1$ holds. Then*

$$f(t) + (K * f)(t) \leq P(t) \leq f(t) + C \times (K * f)(t),$$

*where $C = \left(\frac{K(0)}{K(T)} + 1\right) \left(\frac{1}{1-2\|K\|(1+\varepsilon)}\right)$.*

*Proof.* We consider the upper and lower bound separately.

*Lower bound:* Since $K$ and $f$ is are non-negative, then $\sum_{j=1}^{\infty}(K^{*j} * f)(t) \geq (K^{*1} * f)(t) \geq (K * f)(t)$. Recall the solution to the convolution-type Volterra equation takes the form,

$$P(t) = f(t) + \sum_{j=1}^{\infty}(K^{*j} * f)(t).$$

It immediately follows from $\sum_{j=1}^{\infty}(K^{*j} * f)(t) \geq (K * f)(t)$ the lower bound.

*Upper bound:* The solution to a Volterra equation (in $L^1$) is

$$P(t) = f(t) + \sum_{j=1}^{\infty}(K^{*j} * f)(t).$$

By Lemma C.1 and the hypothesis, there exists a $T > 0$ and $\varepsilon > 0$ such that

$$K^{*j}(s) \leq K(s)\left[\frac{K(0)}{K(T)} + 1\right](2\|K\|(1+\varepsilon))^{j-1},$$

and $(2\|K\|(1+\varepsilon))^{j-1} < 1$. Hence, we have that

$$\sum_{j=1}^{\infty}(K^{*j} * f)(t) = \sum_{j=1}^{\infty}\left(\sum_{s=0}^{t} K^{*j}(s)f(t-s)\right)$$

$$\leq \left(\frac{K(0)}{K(T)} + 1\right)\sum_{j=1}^{\infty}(2\|K\|(1+\varepsilon))^{j-1}(K * f)(t)$$

$$= \left(\frac{K(0)}{K(T)} + 1\right)\left(\frac{1}{1-2\|K\|(1+\varepsilon)}\right)(K * f)(t).$$

The result is shown. $\qquad\square$

### C.3.2   Proof of Theorem 2.1

We are now ready to show one of the main tools used to analyze the loss function, Theorem 2.1. The result relies on approximations for the kernel and forcing functions found in Section F and Section G. We restate the theorem statement to remind the reader of the result.

**Theorem C.1** (Approximation solution for $\mathcal{P}$). *Suppose $\gamma$ and $B$ is at most half the convergence threshold and $\alpha > \frac{1}{4}$. There exists an $M > 0$ large and a constant $C = C(\alpha, \beta, M)$, independent of $d$, so that for all admissible $v$ and $d$, for all $M < \gamma Br$,*

$$\mathcal{F}(r) + (\mathcal{K} * \mathcal{F})(r) \leq \mathcal{P}(r) \leq \mathcal{F}(r) + C \times (\mathcal{K} * \mathcal{F})(r). \tag{58}$$

*The convolution further simplifies. For any $\epsilon > 0$, there exists an $M > 0$ and a constant $C = C(\alpha, \beta, M)$ independent of $d$ so that for all $M < \gamma Br$,*

$$(1-\epsilon)\|\mathcal{K}\| \cdot \mathcal{F}(r) + \frac{1}{C \times \gamma B} \cdot \mathcal{K}(r) \leq (\mathcal{K} * \mathcal{F})(r) \leq C \times \left(\|\mathcal{K}\| \cdot \mathcal{F}(r) + \frac{1}{\gamma B} \cdot \mathcal{K}(r)\right). \tag{59}$$

*Proof of Theorem C.1 / Theorem 2.1.* Note for all $\gamma B r > 1/Md^{2\alpha}$, we have that $c\mathcal{F}_0 \leq \mathcal{F}(r), \mathcal{K}(r) \leq C\mathcal{F}_0(r)$ for some $C, c > 0$. This is where the limiting level starts to dominate. We begin by showing (58). Fix $\varepsilon > 0$. From Proposition G.2, we have that there exists an $M > 0$ sufficiently large so that the hypothesis for Kesten's Lemma, i.e.,

$$\sum_{s=0}^{r} \mathcal{K}(s)\mathcal{K}(r-s) \leq 2(1+\varepsilon)\|\mathcal{K}\|\mathcal{K}(r), \quad \text{for all } d^{2\alpha}/M > \gamma B r > M.$$

Therefore, we get (58) by Lemma C.2.

To prove (59), we begin by

$$\sum_{s=0}^{r} \mathcal{K}(r-s)\mathcal{F}(s) = \sum_{s=0}^{r/2} \mathcal{K}(r-s)\mathcal{F}(s) + \sum_{s=r/2}^{r} \mathcal{K}(r-s)\mathcal{F}(s) \leq \mathcal{K}(\tfrac{r}{2})\sum_{s=0}^{r/2}\mathcal{F}(s) + \mathcal{F}(\tfrac{r}{2})\sum_{s=0}^{r/2}\mathcal{K}(s)$$

where we used monotonicity of $\mathcal{F}$ and $\mathcal{K}$.

Using Proposition H.2 and Proposition H.4, for large $d^{2\alpha}/M \geq \gamma B r \geq M$, we have that $\mathcal{F}(\tfrac{r}{2}) \asymp \mathcal{F}(r)$ since $\mathcal{F}$ is power law for large $r$ (see also Corollary F.1). The same holds for $\mathcal{K}$, using Proposition H.5 and Proposition G.2, $\mathcal{K}(\tfrac{r}{2}) \asymp \mathcal{K}(r)$ for $d^{2\alpha}/M \geq \gamma B r \geq M$ for some $M > 0$.

For small $\gamma B r \leq M$, we have that $\mathcal{F}(r/2) \leq C$ and $\mathcal{K}(r/2) \leq C$ for some $C > 0$. Since $\mathcal{F}$ and $\mathcal{K}$ are monotonic, we can choose a constant so that $\mathcal{F}(r/2) \lesssim \mathcal{F}(r)$ and $\mathcal{K}(r/2) \lesssim \mathcal{K}(r)$ for $\gamma B r \leq M$.

Now using Proposition C.1 and Proposition H.6, we have that

$$\sum_{s=0}^{r} \mathcal{K}(r-s)\mathcal{F}(s) \leq \mathcal{K}(\tfrac{r}{2})\sum_{s=0}^{r/2}\mathcal{F}(s) + \mathcal{F}(\tfrac{r}{2})\sum_{s=0}^{r/2}\mathcal{K}(s) \leq \frac{1}{\gamma B}\mathcal{K}(r) + \mathcal{F}(r)\|\mathcal{K}\|.$$

For the lower bound, we have that

$$\sum_{s=0}^{r} \mathcal{K}(r-s)\mathcal{F}(s) = \sum_{s=0}^{r/2} \mathcal{K}(r-s)\mathcal{F}(s) + \sum_{s=r/2}^{r} \mathcal{K}(r-s)\mathcal{F}(s) \geq \mathcal{K}(r)\sum_{s=0}^{r/2}\mathcal{F}(s) + \mathcal{F}(r)\sum_{s=0}^{r/2}\mathcal{K}(s),$$

where we used monotonicity of $\mathcal{K}$ and $\mathcal{F}$.

We note that $\mathcal{F}(s) \asymp C$ for $\gamma B s \leq M$ for all $M > 0$. Therefore,

$$\sum_{s=0}^{r/2} \mathcal{F}(s) \geq \sum_{s=0}^{M/(2\gamma B)} \mathcal{F}(s) \geq \frac{1}{\gamma B}.$$

On the other hand, by Proposition C.1, for any $\epsilon > 0$, there is an $M$ so that for any $\gamma B r \geq M$,

$$\sum_{s=0}^{r/2} \mathcal{K}(s) \geq (1-\epsilon)\|\mathcal{K}\|.$$

This proves the lower bound.

$\square$

## C.4 Details of risk curves for the phases

We can now put together a coherent picture of the effect of different choices of $\alpha$ and $\beta$ and their impact on the Pareto frontier. We will have 4 distinct phases where the expected loss will exhibit a power law decay and 1 region ($\alpha + \beta \leq 0.5$) for which the expected loss has no power law decay (see Figure 2a). We will describe each of the 4 power law phases below marked by their boundaries.

First, we recall the forcing function $\mathcal{F}$ and kernel function $\mathcal{K}$ introduced in Section 2.1.

Table 5: **Decomposition of the forcing and kernel functions**. We express the forcing function $\mathcal{F}(r)$ as the sum of **three** functions, $\mathcal{F}_{pp}, \mathcal{F}_0, \mathcal{F}_{ac}$, up to errors and kernel function $\mathcal{K}(r)$ as $\mathcal{K}_{pp}(r)$, up to errors. These functions arise from the different parts of the spectrum of the deterministic equivalent for the resolvent of $\hat{K}$.

| Function | Part of spectrum |
|---|---|
| $\mathcal{F}_0(r) \overset{\text{def}}{=} \dfrac{-1}{2\pi i} \oint_{\Gamma_0} \langle \mathcal{R}(z), (D^{1/2}\hat{\beta})^{\otimes 2}\rangle (1 - 2\gamma Bz + \gamma^2 B(B+1)z^2)^r \, dz; \quad \text{see (84)}$ 

 $\mathcal{F}_0(r) = \displaystyle\sum_{j=1}^{v} \dfrac{j^{-2\alpha-2\beta}}{1 + j^{-2\alpha}d^{2\alpha}\kappa(v/d)}\big(1 + \mathcal{O}(d^{-1})\big), \text{ where } \kappa \text{ solves } \displaystyle\int_0^{v/d} \dfrac{\kappa \, dx}{\kappa + x^{2\alpha}} = 1$ 

 $\mathcal{F}_0(r) \sim \begin{cases} \dfrac{d^{-2\alpha}}{\kappa}\left(\sum_{j=1}^v j^{-2\beta}\right), & \text{if } 2\beta > 1 \\ d^{1-2(\alpha+\beta)}\int_0^{v/d} \dfrac{u^{-2\beta}}{\kappa+u^{2\alpha}}\, du, & \text{if } 2\beta < 1; \end{cases}$ (Prop. H.3) | Point mass at $z=0$ 

 (Prop. F.1) |
| $\mathcal{F}_{pp}(r) \overset{\text{def}}{=} \dfrac{1}{2\alpha}\displaystyle\int_0^1 u^{(2\beta-1)/(2\alpha)} \exp(-2\gamma Bru) \, du; \quad \text{see (85)}$ 

 $\mathcal{F}_{pp}(r) \sim (2\alpha)^{-1}(2\gamma B)^{1/(2\alpha)-\beta/\alpha-1} \times \Gamma\left(\dfrac{\beta}{\alpha}-\dfrac{1}{2\alpha}+1\right) \times r^{-(1+\beta/\alpha)+1/(2\alpha)};$ (Prop. H.2) | Pure point |
| $\mathcal{F}_{ac}(r) \overset{\text{def}}{=} \dfrac{c_\beta}{2\alpha}\displaystyle\int_{d^{-2\alpha}}^1 u^{-1/(2\alpha)}d^{-1}\exp(-2\gamma Bru)\, du, \text{ where } c_\beta = \sum_{j=1}^v j^{-2\beta} \text{ if } 2\beta > 1$ 
 and otherwise 0; see (85) 
 If $2\alpha > 1$ and $2\beta > 1$, 
 $\mathcal{F}_{ac}(r) \sim c_\beta(2\gamma B)^{-1+1/(2\alpha)}(2\alpha)^{-1}\Gamma\left(1-\dfrac{1}{2\alpha}\right) \times r^{-1+1/(2\alpha)} \times d^{-1};$ (Prop. H.4) | Abs. con't |
| $\mathcal{K}_{pp}(r) \overset{\text{def}}{=} \dfrac{\gamma^2 B}{2\alpha}\displaystyle\int_0^1 u^{1-1/(2\alpha)}\exp(-2\gamma Bur)\, du, \quad \text{if } \alpha > 1/4; \quad \text{see (86)}$ 
 $\mathcal{K}_{pp}(r) \sim \gamma^2 B \times (2\alpha)^{-1}(2\gamma B)^{-2+1/(2\alpha)} \times \Gamma\left(2-\dfrac{1}{2\alpha}\right) \times r^{-2+1/(2\alpha)};$ (Prop. H.5) | Pure point |

**Forcing function.** For the forcing function,

$$\mathcal{F}(r) = \mathcal{F}_0(r) + \mathcal{F}_{pp}(r) + \mathcal{F}_{ac}(r) + \text{errors}_{\mathcal{F}}. \tag{60}$$

The function $\mathcal{F}_0(r)$ is the component of the forcing function corresponding to the point mass at 0, $\mathcal{F}_{pp}(r)$ is the component of the forcing function corresponding to the pure point part of the spectrum, and lastly, the most complicated part of the spectrum, the forcing function corresponding to the distorted features. In particular, we will show in Section F the exact definitions of $\mathcal{F}_0, \mathcal{F}_{pp}, \mathcal{F}_{ac}$ and $|\text{error}_{\mathcal{F}}|$ are small, and, in Section H, we derive asymptotic-like behaviors for these functions. See Table 5 for definitions and asymptotics.

**Kernel function.** Similarly, the kernel function $\mathcal{K}$ is

$$\mathcal{K}(r) = \mathcal{K}_{pp}(r) + \text{errors}_{\mathcal{K}}.$$

Note here that the kernel function has a multiplication by the eigenvalue of $\hat{K}$ and so the point mass at 0 will not contribute. In Section G, we will give an explicit definition of $\mathcal{K}_{pp}$ and show error terms are small and, in Proposition H.5, we give the asymptotic-like behavior of $\mathcal{K}_{pp}$.

Now we describe in detail the different risk curves that arise for the different phases.

### C.4.1 Above the high-dimensional line (Phases Ia, II, III)

This setting is commonly known as the *trace class*. It is characterized by four components:

- learning rate $\gamma$ can be picked independent of dimension;
- loss curve does not self average, that is, the loss curve does not concentrate around a deterministic function;
- $v \geq d$, but $v$ has no upper bound and so we can take $v \to \infty$;
- batch, $B$, is constrained to be small (see Proposition C.2).

When $2\alpha > 1$, or the trace class phase, the loss will exhibit 3 different phases. We described these phases in detail below.

**Phase Ia:** $(2\beta < 1, 2\alpha > 1)$**.**  In this phase, it notable for three characteristics:

- absolutely continuous part of the forcing function does not participate;
- level at which SGD saturates is affected by $\beta$;
- SGD noise does not participate.

In this case, the loss curve is just a constant multiple of gradient flow. Hence, we have that

$$\mathcal{P}(r) \asymp \mathcal{F}_{pp}(r) + \mathcal{F}_0(r).$$

**Proposition C.3** (Phase Ia: $2\beta < 1$, $2\alpha > 1$)**.** *Suppose $2\beta < 1$ and $2\alpha > 1$. Suppose the learning rate $\gamma$ and batch $B > 0$ satisfy at most half the convergence threshold in Proposition C.2. Then there exists an $M > 0$ large and constants $C = C(\alpha, \beta, M)$ and $c = c(\alpha, \beta, M)$, independent of $d$, so that for all admissible $v$ and $d$, for all $\gamma Br > M$*

$$c \times \big(\mathcal{F}_{pp}(r) + \mathcal{F}_0(r)\big) \leq \mathcal{P}(r) \leq C \times \big(\mathcal{F}_{pp}(r) + \mathcal{F}_0(r)\big). \tag{61}$$

*Proof.* By Theorem C.1, we know that it suffices to look at the forcing function $\mathcal{F}$ and kernel function $\mathcal{K}$. Moreover, in this regime, we have that $\gamma$ and $B$ are constant (see Proposition C.2).

The rest of the argument relies on the bounds found in Proposition H.2, ($\mathcal{F}_{pp}$), Proposition H.4 ($\mathcal{F}_{ac}$), Proposition H.3, ($\mathcal{F}_0$), and Proposition H.5 ($\mathcal{K}_{pp}$).

For the forcing function, $\mathcal{F}_{ac}(r) = 0$ as $2\beta < 1$ (Proposition H.4). Therefore the forcing function is composed of $\mathcal{F}_{pp}(r)$ and $\mathcal{F}_0(r)$.

First, we have that $(\gamma Br)^{-2+1/(2\alpha)} < (\gamma Br)^{-(1+\beta/\alpha)+1/(2\alpha)}$ as $\beta < \alpha$ in this phase. Thus, using Proposition H.2 and Proposition H.5, for $\gamma Br > M$, where $M$ is some constant, we have that $\frac{1}{\gamma B}\mathcal{K}_{pp}(r) \leq C \times \mathcal{F}_{pp}(r)$ for some $C > 0$. Hence the result is shown. $\square$

As a consequence of the argument above, we know that

$$\mathcal{P}(r) \approx \begin{cases} \mathcal{F}_{pp}(r), & \text{if } \gamma Br \leq D_0 \\ \mathcal{F}_0(r), & \text{if } \gamma Br \geq D_0 \end{cases} \quad \text{for some } D_0 \text{ that depends on } d.$$

**Phase II:** $(2\beta > 1, 2\alpha > 1, \beta < \alpha)$  For this phase, we see that

- limit level is unaffected by $\beta$;
- absolutely continuous spectrum takes over for $r \in (Md^\alpha, d^{2\alpha}/M)$ for some $M$;
- SGD noise does not participate.

Therefore, in this case, we have that

$$\mathcal{P}(r) \asymp \mathcal{F}_{pp}(r) + \mathcal{F}_{ac}(r) + \mathcal{F}_0(r).$$

**Proposition C.4** (Phase II: $2\beta > 1$, $2\alpha > 1$, $\beta < \alpha$)**.** *Suppose $2\beta > 1$, $2\alpha > 1$, and $\beta < \alpha$. Suppose the learning rate $\gamma$ and batch $B > 0$ satisfy at most half the convergence threshold in Proposition C.2. Then there exists an $M > 0$ large and constants $C = C(\alpha, \beta, M)$ and $c = c(\alpha, \beta, M)$, independent of $d$, so that for all admissible $v$ and $d$, for all $\gamma Br > M$*

$$c \times \big(\mathcal{F}_{pp}(r) + \mathcal{F}_{ac}(r) + \mathcal{F}_0(r)\big) \leq \mathcal{P}(r) \leq C \times \big(\mathcal{F}_{pp}(r) + \mathcal{F}_{ac}(r) + \mathcal{F}_0(r)\big). \tag{62}$$

*Proof.* By Theorem C.1, we know that it suffices to look at the forcing function $\mathcal{F}$ and kernel function $\mathcal{K}$. Moreover, in this regime, we have that $\gamma$ and $B$ are constant (see Proposition C.2).

The rest of the argument relies on the bounds found in Proposition H.2, ($\mathcal{F}_{pp}$), Proposition H.4 ($\mathcal{F}_{ac}$), Proposition H.3, ($\mathcal{F}_0$), and Proposition H.5 ($\mathcal{K}_{pp}$).

$\gamma Br \leq M_0$, *for some $M_0$:* First, we have that $\frac{1}{\gamma B}\mathcal{K}_{pp}(r) \leq C_0 \times \mathcal{F}_{pp}(r)$ (Proposition H.5) and $\mathcal{F}_{ac}(r) \leq C_0 \times \mathcal{F}_{pp}(r)$ (Proposition H.4) for some constant $C_0 > 0$. The constant $M_0$ is where the asymptotic of $\mathcal{F}_{pp}$ starts to apply.

$M_0 \leq \gamma Br \leq M_1$, *for some $M_0$ and for all $M_1 > M_0$:* We see that $(\gamma Br)^{-2+1/(2\alpha)} < (\gamma Br)^{-(1+\beta/\alpha)+1/(2\alpha)}$ as $\beta < \alpha$ in this phase. Thus, using Proposition H.2 and Proposition H.5, we have that $\frac{1}{\gamma B}\mathcal{K}_{pp}(r) \leq C_1 \times \mathcal{F}_{pp}(r)$ for some $C_1 > 0$. A quick computation shows that $\mathcal{F}_{ac}(r) \leq \mathcal{F}_{pp}(r)$.

$M_1 \leq \gamma Br \leq M_2 d^{2\alpha}$, *for any $M_1$ and some $M_2$:* The $M_2$ is the smallest of the two endpoints for the asymptotics of $\mathcal{F}_{pp}$ and $\mathcal{F}_{ac}$. As in the previous regime, we have that $\frac{1}{\gamma B}\mathcal{K}_{pp}(r) \lesssim \mathcal{F}_{pp}(r)$. In this region, $\mathcal{F}_{ac}(r) \asymp d^{-1}(\gamma Br)^{-1+1/(2\alpha)}$ and $\mathcal{F}_{pp}(r) \asymp (\gamma Br)^{-(1+\beta/\alpha)+1/(2\alpha)}$. We see at $(\gamma Br) = d^{2\alpha}$ that $(\gamma Br)^{-\beta/\alpha} \leq (d^{2\alpha})^{-\beta/\alpha} = d^{-2\beta} \leq d^{-1}$ as $2\beta > 1$. Therefore, at $\gamma Br = d^{2\alpha}$, $\mathcal{F}_{pp}(r) \lesssim \mathcal{F}_{ac}(r)$ and we started, i.e., when $r = M_1$ with $\mathcal{F}_{ac}(r) \lesssim \mathcal{F}_{pp}(r)$. Therefore, we must change in this regime to being $\mathcal{F}_{ac}$ dominate.

$M_2 d^{2\alpha} \leq \gamma Br$ *for all $M_2$:* In this case, all terms are bounded above by $\mathcal{F}_0(r)$. $\qquad\square$

As a consequence of the argument above, we know that

$$\mathcal{P}(r) \approx \begin{cases} \mathcal{F}_{pp}(r), & \text{if } \gamma Br \leq D_0 \\ \mathcal{F}_{ac}(r), & \text{if } D_0 \leq \gamma Br \leq D_1 \\ \mathcal{F}_0(r), & \text{if } \gamma Br \geq D_1 \end{cases} \quad \text{for some } D_0, D_1 \text{ that depend on } d. \qquad (63)$$

**Phase III: SGD noise appears,** $(2\beta > 1, 2\alpha > 1, \beta > \alpha)$ In this case, we see that SGD changes the dynamics over gradient flow. In particular,

- limit level is unaffected by $\beta$;
- absolutely continuous forcing function takes over for iterations $r \in (Md, d^{2\alpha}/M)$ for some $M$;
- SGD noise regulates convergence.

Thus, we have that

$$\mathcal{P}(r) \asymp \mathcal{F}_{ac}(r) + \mathcal{F}_0(r) + \frac{1}{\gamma B}\mathcal{K}_{pp}(r). \qquad (64)$$

**Proposition C.5** (Phase III: $2\beta > 1$, $2\alpha > 1$, $\beta > \alpha$). *Suppose $2\beta > 1$, $2\alpha > 1$, and $\beta > \alpha$. Suppose the learning rate $\gamma$ and batch $B > 0$ satisfy at most half the convergence threshold in Proposition C.2. Then there exists an $M > 0$ large and constants $C = C(\alpha, \beta, M)$ and $c = c(\alpha, \beta, M)$, independent of $d$, so that for all admissible $v$ and $d$, for all $\gamma Br > M$*

$$c \times \left(\mathcal{F}_{ac}(r) + \mathcal{F}_0(r) + \frac{1}{\gamma B}\mathcal{K}_{pp}(r)\right) \leq \mathcal{P}(r) \leq C \times \left(\mathcal{F}_{ac}(r) + \mathcal{F}_0(r) + \frac{1}{\gamma B}\mathcal{K}_{pp}(r)\right). \qquad (65)$$

*Proof.* By Theorem C.1, we know that it suffices to look at the forcing function $\mathcal{F}$ and kernel function $\mathcal{K}$. Moreover, in this regime, we have that $\gamma$ and $B$ are constant (see Proposition C.2).

The rest of the argument relies on the bounds found in Proposition H.2, ($\mathcal{F}_{pp}$), Proposition H.4 ($\mathcal{F}_{ac}$), Proposition H.3, ($\mathcal{F}_0$), and Proposition H.5 ($\mathcal{K}_{pp}$).

$\gamma Br \leq M_0$, *for some $M_0$:* First, we have that $\mathcal{F}_{pp}(r) \leq C_0 \times \frac{1}{\gamma B}\mathcal{K}_{pp}(r)$ (Proposition H.5) and $\mathcal{F}_{ac}(r) \leq C_0 \times \frac{1}{\gamma B}\mathcal{K}_{pp}(r)$ (Proposition H.4) for some constant $C_0 > 0$. The constant $M_0$ is where the asymptotic of $\mathcal{K}_{pp}$ starts to apply.

$M_0 \leq \gamma Br \leq M_1$, *for some $M_0$ and for all $M_1 > M_0$:* We see that $(\gamma Br)^{-2+1/(2\alpha)} > (\gamma Br)^{-(1+\beta/\alpha)+1/(2\alpha)}$ as $\beta > \alpha$ in this phase. Thus, using Proposition H.2 and Proposition H.5, we have that $\mathcal{F}_{pp} \leq C_1 \times \frac{1}{\gamma B}\mathcal{K}_{pp}(r)$ for some $C_1 > 0$. A quick computation shows that $\mathcal{F}_{ac}(r) \leq \mathcal{K}_{pp}(r)$.

$M_1 \leq \gamma Br \leq M_2 d^{2\alpha}$, *for any $M_1$ and some $M_2$:* The $M_2$ is the smallest of the two endpoints for the asymptotics of $\mathcal{K}_{pp}$ and $\mathcal{F}_{ac}$. As in the previous regime, we have that $\frac{1}{\gamma B}\mathcal{K}_{pp}(r) \lesssim \mathcal{F}_{pp}(r)$. In this region, $\mathcal{F}_{ac}(r) \asymp d^{-1}r^{-1+1/(2\alpha)}$ and $\mathcal{K}_{pp}(r) \asymp r^{-2+1/(2\alpha)}$. We see at $\gamma Br = d^{2\alpha}$ that $(\gamma Br)^{-1} \leq (d^{2\alpha})^{-1} = d^{-2\alpha} \leq d^{-1}$ as $2\alpha > 1$. Therefore, at $(\gamma Br) \asymp d^{2\alpha}$, $\mathcal{K}_{pp}(r) \lesssim \mathcal{F}_{ac}(r)$ and we started, i.e., when $r = M_1$ with $\mathcal{F}_{ac}(r) \lesssim \mathcal{K}_{pp}(r)$. Therefore, we must change in this regime to

being $\mathcal{F}_{ac}$ dominate.

$M_2 d^{2\alpha} \leq r\gamma B$ *for all* $M_2$: In this case, all terms are bounded above by $\mathcal{F}_0(r)$. □

As a consequence of the argument above, we know that

$$\mathcal{P}(r) \approx \begin{cases} \mathcal{K}_{pp}(r), & \text{if } \gamma Br \leq D_0 \\ \mathcal{F}_{ac}(r), & \text{if } D_0 \leq \gamma Br \leq D_1 \\ \mathcal{F}_0(r), & \text{if } \gamma Br \geq D_1 \end{cases} \quad \text{for some } D_0, D_1 \text{ that depend on } d. \qquad (66)$$

### C.4.2 Below the high-dimensional line (Phases IVa, IVb, Ib, Ic)

One of the main differences between the previous regime and this regime is that $V$ can not be taken to $\infty$ independent of $d$. As a result, we call this below the *high-dimensional line* and it is precisely bounded by whether $2\alpha$ is summable or not.

The four main characteristics of this regime are:

- learning rate $\gamma$ scales like $v^{-1+2\alpha}$;
- SGD loss, i.e., $\mathbb{E}\left[\mathscr{P}(\theta_r)\right]$ self-concentrates;
- $v$ can not be too large, i.e., $d$ and $v$ are proportional;
- batch can be large (i.e., $\gamma B \leq 1$) since the learning rate is small ($\gamma \sim v^{-1+2\alpha}$).

In Phases IV, Ib, and Ic, because $j^{-2\alpha}$ is not summable, the summation of the $j$ depends on the dimension $v$. Thus, the kernel norm is

$$\|\mathcal{K}\| \sim \frac{\gamma}{2} \sum_{j=1}^{v} j^{-2\alpha} \sim \frac{\gamma}{2(1-2\alpha)} v^{1-2\alpha},$$

where the learning rate $\gamma$ is chosen so that $\|\mathcal{K}\|$ is constant, i.e., $\gamma = \frac{\tilde{\gamma}}{\|\mathcal{K}\|}$ where $\tilde{\gamma} > 0$ is a constant.

**Phase IV,** $(2\beta > 1, \frac{1}{4} < \alpha < \frac{1}{2})$. In this phase, we have the following

- limiting value of the loss that SGD converges to is unaffected by $\beta$;
- pure point forcing function plays a role;
- absolutely continuous part of the spectrum does not contribute to the forcing function;
- SGD noise affect the loss curves.

In this phase, the loss curve is

$$\mathcal{P}(r) \asymp \mathcal{F}_{pp}(r) + \mathcal{F}_0(r) + \frac{1}{\gamma B}\mathcal{K}_{pp}(r).$$

The following gives the precise statement.

**Proposition C.6** (Phase IV: $2\beta > 1$, $\frac{1}{4} < \alpha < \frac{1}{2}$)**.** *Suppose* $2\beta > 1$ *and* $\frac{1}{4} < \alpha < \frac{1}{2}$. *Suppose the learning rate* $\gamma$ *and batch* $B > 0$ *satisfy at most half the convergence threshold in Proposition C.2. Then there exists an* $M > 0$ *large and constants* $C = C(\alpha, \beta, M)$ *and* $c = c(\alpha, \beta, M)$, *independent of* $d$, *so that for all admissible* $v$ *and* $d$, *for all* $\gamma Br > M$

$$c \times \left(\mathcal{F}_{pp}(r) + \mathcal{F}_0(r) + \frac{1}{\gamma B}\mathcal{K}_{pp}(r)\right) \leq \mathcal{P}(r) \leq C \times \left(\mathcal{F}_{pp}(r) + \mathcal{F}_0(r) + \frac{1}{\gamma B}\mathcal{K}_{pp}(r)\right). \qquad (67)$$

*Proof.* By Theorem C.1, we know that it suffices to look at the forcing function $\mathcal{F}$ and kernel function $\mathcal{K}$. Moreover, in this regime, we have that $\gamma$ decreases like $d^{2\alpha-1}$ (see Proposition C.2).

The rest of the argument relies on the bounds found in Proposition H.2, ($\mathcal{F}_{pp}$), Proposition H.4 ($\mathcal{F}_{ac}$), Proposition H.3, ($\mathcal{F}_0$), and Proposition H.5 ($\mathcal{K}_{pp}$).

We first note there is no $\mathcal{F}_{ac}(r) \lesssim \mathcal{F}_0$ and therefore it is too small to contribute.

$\gamma Br \leq M_0$, *for some $M_0$:* First, we have that $\frac{1}{\gamma B}\mathcal{K}_{pp}(r) \leq C_0 \times \mathcal{F}_{pp}(r)$ for some constant $C_0 > 0$. The constant $M_0$ is where the asymptotic of $\mathcal{F}_{pp}$ starts to apply.

$M_0 \leq \gamma Br \leq M_1$, *for some $M_0$ and for all $M_1 > M_0$:* We see that $\gamma(\gamma Br)^{-2+1/(2\alpha)} < (\gamma Br)^{-(1+\beta/\alpha)+1/(2\alpha)}$ since $\gamma \asymp d^{2\alpha-1}$ and $2\alpha < 1$ in this phase. Thus, using Proposition H.2 and Proposition H.5, we have that $\frac{1}{\gamma B}\mathcal{K}_{pp}(r) \leq C_1 \times \mathcal{F}_{pp}$ for some $C_1 > 0$.

$M_1 \leq \gamma Br \leq M_2 d^{2\alpha}$, *for any $M_1$ and some $M_2$:* The $M_2$ is the smallest of the two endpoints for the asymptotics of $\mathcal{K}_{pp}$ and $\mathcal{F}_{pp}$. In this region, $\mathcal{F}_{pp}(r) \asymp (\gamma Br)^{-(1+\beta/\alpha)+1/(2\alpha)}$ and $\gamma \times \frac{1}{\gamma B}\mathcal{K}_{pp}(r) \asymp \gamma \times (\gamma Br)^{-2+1/(2\alpha)} \asymp d^{2\alpha-1} \times (\gamma Br)^{-2+1/(2\alpha)}$. We see at $r = d^{2\alpha}$ that $d^{2\alpha-1}(\gamma Br)^{-1} = d^{-1} \geq d^{-2\beta} = (d^{2\alpha})^{-\beta/\alpha}$. Thus $\mathcal{F}_{pp}(r) \lesssim \frac{1}{\gamma B}\mathcal{K}_{pp}(r)$ and we started, i.e., when $r = M_1$ with $\mathcal{K}_{pp}(r) \lesssim \mathcal{F}_{pp}(r)$. Therefore, we must change in this regime to being $\mathcal{K}_{pp}$ dominate.

$M_2 d^{2\alpha} \leq \gamma Br$ *for all $M_2$:* In this case, all terms are bounded above by $\mathcal{F}_0(r)$. $\qquad\square$

As a consequence of the argument above, we know that

$$\mathcal{P}(r) \approx \begin{cases} \mathcal{F}_{pp}(r), & \text{if } \gamma Br \leq D_0 \\ \frac{1}{\gamma B}\mathcal{K}_{pp}(r), & \text{if } D_0 \leq \gamma Br \leq D_1 \\ \mathcal{F}_0(r), & \text{if } \gamma Br \geq D_1 \end{cases} \quad \text{for some } D_0, D_1 \text{ that depend on } d. \quad (68)$$

**Phase Ib,** $(2\beta < 1, 0.25 < \alpha < 0.5, 2(\alpha + \beta) > 1)$. Phase Ia, Ib, and Ic are quite similar as the dynamics of SGD only depend on the forcing function pure point and limiting value. In this phase, the learning rate $\gamma$ is dimension dependent, unlike Phase Ia, and the following hold

- limiting value of the loss that SGD converges to is $d^{-2\alpha+1-2\beta}$;
- absolutely continuous part of the spectrum does not contribute to the forcing function;
- SGD noise not does affect the loss curves.

In this phase, the loss curve is

$$\mathcal{P}(r) \asymp \mathcal{F}_{pp}(r) + \mathcal{F}_0(r).$$

Although we did not prove the statement for $\alpha < 0.25$ as we do not have estimates for the kernel function, we believe that statement still holds. We believe that the kernel function stops becoming power law when $\alpha < 0.25$, but the forcing function is still power law.

The following gives the precise statement.

**Proposition C.7** (Phase Ib: $2\beta < 1, \frac{1}{4} < \alpha < \frac{1}{2}, 2(\alpha + \beta) > 1$). *Suppose $2\beta < 1$, $2(\alpha + \beta) > 1$, and $\frac{1}{4} < \alpha < \frac{1}{2}$. Suppose the learning rate $\gamma$ and batch $B > 0$ satisfy at most half the convergence threshold in Proposition C.2. Then there exists an $M > 0$ large and constants $C = C(\alpha, \beta, M)$ and $c = c(\alpha, \beta, M)$, independent of $d$, so that for all admissible $v$ and $d$, for all $\gamma Br > M$*

$$c \times \left(\mathcal{F}_{pp}(r) + \mathcal{F}_0(r)\right) \leq \mathcal{P}(r) \leq C \times \left(\mathcal{F}_{pp}(r) + \mathcal{F}_0(r)\right). \quad (69)$$

*Proof.* By Theorem C.1, we know that it suffices to look at the forcing function $\mathcal{F}$ and kernel function $\mathcal{K}$. Moreover, in this regime, we have that $\gamma$ decreases like $d^{2\alpha-1}$ (see Proposition C.2).

The rest of the argument relies on the bounds found in Proposition H.2, ($\mathcal{F}_{pp}$), Proposition H.4 ($\mathcal{F}_{ac}$), Proposition H.3, ($\mathcal{F}_0$), and Proposition H.5 ($\mathcal{K}_{pp}$).

We first note there is no $\mathcal{F}_{ac}(r)$.

$\gamma Br \leq M_0$, *for some $M_0$:* First, we have that $\frac{1}{\gamma B}\mathcal{K}_{pp}(r) \leq C_0 \times \mathcal{F}_{pp}(r)$ for some constant $C_0 > 0$. The constant $M_0$ is where the asymptotic of $\mathcal{F}_{pp}$ starts to apply.

$M_0 \leq \gamma Br \leq M_1 d^{2\alpha}$, *for any $M_0$ and some $M_1$:* The $M_1$ is the smallest of the two endpoints for the asymptotics of $\mathcal{K}_{pp}$ and $\mathcal{F}_{pp}$. In this region, $\mathcal{F}_{pp}(r) \asymp (\gamma Br)^{-(1+\beta/\alpha)+1/(2\alpha)}$ and $\gamma \times \frac{1}{\gamma B}\mathcal{K}_{pp}(r) \asymp \gamma \times (\gamma Br)^{-2+1/(2\alpha)} \asymp d^{2\alpha-1} \times (\gamma Br)^{-2+1/(2\alpha)}$. We see at $r = d^{2\alpha}$ that $d^{2\alpha-1}(\gamma Br)^{-1} = d^{-1} \leq d^{-2\beta} = (d^{2\alpha})^{-\beta/\alpha}$. Thus $\frac{1}{\gamma B}\mathcal{K}_{pp}(r) \lesssim \mathcal{F}_{pp}(r)$ and we started, i.e., when

$r = M_1$ with $\mathcal{K}_{pp}(r) \lesssim \mathcal{F}_{pp}(r)$. Therefore, $\mathcal{F}_{pp}$ must dominate.

$M_1 d^{2\alpha} \leq \gamma Br$ *for all* $M_1$: In this case, all terms are bounded above by $\mathcal{F}_0(r)$. $\qquad\square$

We expect Prop. C.7 to hold with the same conclusions for $2\beta < 1$, $2\alpha < 1$, and $2(\alpha + \beta) > 1$.

As a consequence of the argument above, we know that

$$\mathcal{P}(r) \approx \begin{cases} \mathcal{F}_{pp}(r), & \text{if } \gamma Br \leq D_0 \\ \mathcal{F}_0(r), & \text{if } \gamma Br \geq D_0 \end{cases} \quad \text{for some } D_0 \text{ that depends on } d. \tag{70}$$

**Phase Ic,** $(2\beta > 1, 0 < \alpha < \frac{1}{4}$. Lastly, we consider Phase Ic, which is similar to Phases Ia and Ib. The following holds in this phase.

- limiting value of the loss that SGD converges to is $d^{-2\alpha+1-2\beta}$;
- absolutely continuous part of the spectrum does not contribute to the forcing function;
- SGD noise not does affect the loss curves.

In this phase, the loss curve is
$$\mathcal{P}(r) \asymp \mathcal{F}_{pp}(r) + \mathcal{F}_0(r).$$
Under the assumption that Theorem C.1 holds for $\alpha > 1/4$, we get the following.

**Proposition C.8** (Phase Ic: $2\beta > 1$, $0 < \alpha < \frac{1}{4}$). *Suppose* $2\beta > 1$ *and* $0 < \alpha < \frac{1}{4}$ *and Theorem C.1 holds. Suppose the learning rate* $\gamma$ *and batch* $B > 0$ *satisfy at most half the convergence threshold in Proposition C.2. Then there exists an* $M > 0$ *large and constants* $C = C(\alpha, \beta, M)$ *and* $c = c(\alpha, \beta, M)$, *independent of* $d$, *so that for all admissible* $v$ *and* $d$, *for all* $\gamma Br > M$

$$c \times \big(\mathcal{F}_{pp}(r) + \mathcal{F}_0(r)\big) \leq \mathcal{P}(r) \leq C \times \big(\mathcal{F}_{pp}(r) + \mathcal{F}_0(r)\big). \tag{71}$$

We can not prove this statement as we do not have sharp bounds on the kernel function in this region. We believe that the kernel function stops becoming power law, but the forcing function is still power law. Thus, it should become even more forcing function dominate.

We believe the loss curve follows similar behavior to Phase Ia and Phase Ib, that is,

$$\mathcal{P}(r) \approx \begin{cases} \mathcal{F}_{pp}(r), & \text{if } \gamma Br \leq D_0 \\ \mathcal{F}_0(r), & \text{if } \gamma Br \geq D_0 \end{cases} \quad \text{for some } D_0 \text{ that depends on } d. \tag{72}$$

## D  Compute-optimal curves

Throughout this section, consider the deterministic equivalent loss function $\mathcal{P}(r) = \mathcal{P}(r, d)$. Moreover as batch size $B$ is order 1, it only effects the compute-optimal curves by a constant. Therefore, we can set $B = 1$. For each iteration $r$, the SGD costs $d$ flops, or equivalently $r/d =$ flops, $\mathfrak{f}$. The goal is to find the optimal compute line as a function of the number of flops $\mathfrak{f}$:

$$\min_d \mathcal{P}(\tfrac{\mathfrak{f}}{d}, d).$$

If $d^\star(\mathfrak{f}) \stackrel{\text{def}}{=} \arg\min_d \mathcal{P}(\tfrac{\mathfrak{f}}{d}, d)$, the optimal compute line is precisely $\mathcal{P}\big(\tfrac{\mathfrak{f}}{d^\star(\mathfrak{f})}, d^\star(\mathfrak{f})\big)$.

To do this, we simplify the loss curve $\mathcal{P}(\tfrac{\mathfrak{f}}{d}, d)$. While it is possible to minimize this as a function of $d$, an alternative function considered is the following

$$\tilde{\mathcal{P}}(r, d) \stackrel{\text{def}}{=} \mathcal{F}_{pp}(r, d) \vee \mathcal{F}_{ac}(r, d) \vee \mathcal{F}_0(r, d) \vee \tfrac{1}{\gamma B}\mathcal{K}_{pp}(r, d), \tag{73}$$

which achieves the right power law behavior as the true compute-optimal curve and deviates from this true curve by an absolutely constant (independent of $d, \mathfrak{f}$) (see Theorem C.1). Note here some of the terms should be taken as 0 when not defined for the different phases.

Using this alternative loss function, $\tilde{\mathcal{P}}(r, d)$, the compute-optimal line must occur at one of the corner points, i.e., where any pair of functions equal each other. The following lemma gives a useful characterization of these points.

Table 6: **Summary of the compute-optimal curves for $\tilde{\mathcal{P}}(\frac{\mathfrak{f}}{d}, d)$ for above the high-dimensional line,** $2\alpha > 1$. This includes Phases Ia, II, and III.

| | **Trade off** | **Compute-optimal Curves** |
|---|---|---|
| **Phase Ia** (Prop. D.1) | $\mathcal{F}_{pp} = \mathcal{F}_0$ | $\tilde{\mathcal{P}}^\star_{\text{Phase Ia}}(\mathfrak{f}) \asymp \mathfrak{f}^{\left(\frac{1}{2\alpha+1}-1\right)(1+\beta/\alpha-1/(2\alpha))}$ 
 $d^\star_{\text{Phase Ia}} \asymp \mathfrak{f}^{1/(2\alpha+1)}$ |
| **Phase II** (Prop. D.2) | $\mathcal{F}_{pp} = \mathcal{F}_{ac}$ | $\tilde{\mathcal{P}}^\star_{\text{Phase II}}(\mathfrak{f}) \asymp \mathfrak{f}^{-\frac{2\alpha+2\beta-1}{2(\alpha+\beta)}}$ 
 $d^\star_{\text{Phase II}} \asymp \mathfrak{f}^{(\beta/\alpha)/(1+\beta/\alpha)}$ |
| **Phase III** (Prop. D.3) | $\frac{1}{\gamma B}\mathcal{K}_{pp} = \mathcal{F}_{ac}$ | $\tilde{\mathcal{P}}^\star_{\text{Phase III}}(\mathfrak{f}) \asymp \mathfrak{f}^{(1-4\alpha)/(4\alpha)}$ 
 $d^\star_{\text{Phase III}} \asymp \mathfrak{f}^{1/2}$ |

**Lemma D.1.** *Suppose $\mathcal{C}_0, \mathcal{C}_1 > 0$ are constants and $\gamma_0, \gamma_1, p_0, p_1 > 0$ exponents such that a function $\hat{\mathcal{P}}(r, d)$ equals*

$$\hat{\mathcal{P}}(r, d) = \max\left\{\mathcal{C}_0 r^{-\gamma_0} d^{-p_0}, \mathcal{C}_1 r^{-\gamma_1} d^{-p_1}\right\}.$$

*Then replacing $r \mapsto \frac{\mathfrak{f}}{d}$ the minimizer in $d$ satisfies*

$$d^\star \stackrel{def}{=} \arg\min_d \{\hat{\mathcal{P}}(\mathfrak{f}, d)\} = \left(\frac{\mathcal{C}_0}{\mathcal{C}_1}\right)^{1/(\gamma_1 - p_1 - \gamma_0 + p_0)} \times \mathfrak{f}^{(-\gamma_0 + \gamma_1)/(\gamma_1 - p_1 - \gamma_0 + p_0)}$$

*and the optimal value is*

$$\min_d \hat{\mathcal{P}}(\mathfrak{f}, d) = \mathcal{C}_0 \times \mathfrak{f}^{-\gamma_0} \times (d^\star)^{\gamma_0 - p_0}.$$

*Proof.* The proof is a straightforward computation. The minimizer of $\hat{\mathcal{P}}(\mathfrak{f}, d)$ in $d$ must occur where the two terms in the maximum are equal, i.e.,

$$\mathcal{C}_0\left(\frac{\mathfrak{f}}{d}\right)^{-\gamma_0} d^{-p_0} = \mathcal{C}_1\left(\frac{\mathfrak{f}}{d}\right)^{-\gamma_1} d^{-p_1}.$$

Solving for this $d$ gives $d^\star$. Plugging in the value of $d^\star$ into $\hat{\mathcal{P}}(\mathfrak{f}, d)$ gives the optimal value. $\square$

**Remark D.1.** *The possible minimal values of* (73), *i.e., where pairs of functions in the max are equal, can be reduced further. For instance, if $\mathcal{F}_{ac}(r, d)$ exist for the phase, then for some $0 < r_0 < r_1 < r_2$*

$$\tilde{\mathcal{P}}(r, d) \approx \begin{cases} \mathcal{F}_{pp}(r, d), & 0 < r \leq r_0 \\ \frac{1}{\gamma B}\mathcal{K}_{pp}(r, d) & r_0 < r \leq r_1 \\ \mathcal{F}_{ac}(r, d), & r_1 < r < r_2 \\ \mathcal{F}_0(r, d), & r_2 < r \end{cases}$$

*Thus, there are only a maximum of three points to check in order to find the optimal compute curve.*

**Remark D.2.** *In view of Lemma D.1, to find the optimal compute curves, we first find the potential curves (i.e., all the possible combinations of two functions in the loss curve are equal while still lying on the loss curve). Then the curve which has the smallest exponent on the flops, $\mathfrak{f}$, is the optimal compute curve.*

### D.1 Compute-optimal curves: Above the high-dimensional line (Phases Ia, II, III).

To ease notation, we introduce several constants that will be used only in this Section D.1:

$$\mathcal{F}_{pp}(r, d) \asymp (\gamma Br)^{-(1+\beta/\alpha)+1/(2\alpha)}, \qquad \mathcal{F}_{ac}(r, d) \asymp d^{-1}(\gamma Br)^{-1+1/(2\alpha)},$$

$$\frac{1}{\gamma B}\mathcal{K}_{pp}(r, d) \asymp \gamma \times (\gamma Br)^{-2+1/(2\alpha)}, \quad \text{and} \quad \mathcal{F}_0(r, d) \asymp d^{-2\alpha+\max\{0, 1-2\beta\}},$$

where the asymptotics only hold in specific regions of the space of $\gamma Br$. For additional details on the derivation of these asymptotics and the constraints on $\gamma Br$ where asymptotics hold, see Section H.

**Remark D.3.** *The constants in the asymptotics are dimension independent and only depend on $\alpha, \beta$.*

The compute-optimal curves are summarized in Table 6.

### D.1.1   Phase Ia

In this case, the approximate loss curve is given by

$$\tilde{\mathcal{P}}(\tfrac{f}{d}, d) = \max\{\mathcal{F}_{pp}(\tfrac{f}{d}, d), \mathcal{F}_0(\tfrac{f}{d}, d)\} \asymp \max\{\big(\tfrac{f}{d}\big)^{-(1+\beta/\alpha)+1/(2\alpha)}, \times d^{-2\alpha+1-2\beta}\}. \qquad (74)$$

With this, we give a description of the optimal compute curve.

**Proposition D.1** (Phase Ia: Compute-optimal Curve). *Suppose we are in Phase Ia, that is, $2\beta < 1$ and $2\alpha > 1$. The compute-optimal curve using $\tilde{\mathcal{P}}(\tfrac{f}{d}, d)$ in (74) occurs when $\mathcal{F}_{pp}(\tfrac{f}{d}, d) = \mathcal{F}_0(\tfrac{f}{d}, d)$. Precisely, the optimal $d^\star$ which minimizes $\tilde{\mathcal{P}}(\tfrac{f}{d}, d)$ is*

$$d^\star_{Phase\ Ia} \asymp f^{1/(2\alpha+1)},$$

*and the compute-optimal curve is*

$$\tilde{\mathcal{P}}^\star_{Phase\ Ia}(f) \asymp f^{\left(\frac{1}{2\alpha+1}-1\right)(1+\beta/\alpha-1/(2\alpha))}.$$

*Proof.* We apply Lemma D.1 with

$$\mathcal{C}_0 = 1, \quad \gamma_0 = 1 + \beta/\alpha - 1/(2\alpha), \quad p_0 = 0$$
$$\text{and}\quad \mathcal{C}_1 = 1, \quad \gamma_1 = 0, \quad p_1 = 2\alpha - 1 + 2\beta.$$

$\square$

### D.1.2   Phase II

In this case, the approximate loss curve has three terms (Proposition C.4 with (63))

$$\tilde{\mathcal{P}}(\tfrac{f}{d}, d) = \max\big\{\mathcal{F}_{pp}(\tfrac{f}{d}, d), \mathcal{F}_{ac}(\tfrac{f}{d}, d), \mathcal{F}_0(\tfrac{f}{d}, d)\big\}$$
$$\asymp \max\big\{\big(\tfrac{f}{d}\big)^{-(1+\beta/\alpha)+1/(2\alpha)}, \big(\tfrac{f}{d}\big)^{-1+1/(2\alpha)} \times d^{-1}, d^{-2\alpha}\big\} \qquad (75)$$

**Proposition D.2** (Phase II: Compute-optimal Curve). *Suppose we are in Phase II, that is, $2\beta > 1$, $2\alpha > 1$, and $\beta < \alpha$. The compute-optimal curve using $\tilde{\mathcal{P}}(\tfrac{f}{d}, d)$ in (75) occurs when $\mathcal{F}_{pp}(\tfrac{f}{d}, d) = \mathcal{F}_{ac}(\tfrac{f}{d}, d)$. Precisely, the optimal $d^\star$ which minimizes $\tilde{\mathcal{P}}(\tfrac{f}{d}, d)$ is*

$$d^\star_{Phase\ II} \asymp f^{(\beta/\alpha)/(1+\beta/\alpha)},$$

*and the compute-optimal curve is*

$$\tilde{\mathcal{P}}^\star_{Phase\ II}(f) \asymp f^{-\frac{2\alpha+2\beta-1}{2(\alpha+\beta)}}.$$

*Proof.* Using the Remark D.2 after Lemma D.1 and Proposition C.4 with (63), we only need to check two intersections: $\mathcal{F}_{pp} = \mathcal{F}_{ac}$ and $\mathcal{F}_{ac} = \mathcal{F}_0$. The curve which has the smallest (i.e., largest negative) exponent (i.e, steepest curve on a log-log plot) is the compute-optimal curve.

*Case 1: Consider $\mathcal{F}_{pp}(\tfrac{f}{d}, d) = \mathcal{F}_{ac}(\tfrac{f}{d}, d)$.* We apply Lemma D.1 with

$$\mathcal{C}_0 = 1, \quad \gamma_0 = 1 + \beta/\alpha - 1/(2\alpha), \quad p_0 = 0$$
$$\text{and}\quad \mathcal{C}_1 = 1, \quad \gamma_1 = 1 - 1/(2\alpha), \quad p_1 = 1$$

to get that the minimum is

$$d_1^\star \asymp f^{(\beta/\alpha)/(1+\beta/\alpha)}$$

and the optimal value is

$$\tilde{\mathcal{P}}_1^\star(f) \asymp f^{-\frac{2\alpha+2\beta-1}{2(\alpha+\beta)}}.$$

*Case 2: Consider $\mathcal{F}_{ac}(\tfrac{f}{d}, d) = \mathcal{F}_0(\tfrac{f}{d}, d)$.* As before, we apply Lemma D.1 with

$$\mathcal{C}_0 = 1, \quad \gamma_0 = 0, \quad p_0 = 2\alpha$$
$$\text{and}\quad \mathcal{C}_1 = 1, \quad \gamma_1 = 1 - 1/(2\alpha), \quad p_1 = 1$$

to get that the minimum is

$$d_2^\star \asymp f^{1/(2\alpha+1)}$$

and the optimal value is

$$\tilde{\mathcal{P}}_2^\star(f) \asymp f^{-2\alpha/(2\alpha+1)}.$$

One can check that

$$-\frac{2\alpha + 2\beta - 1}{2(\alpha + \beta)} < \frac{-2\alpha}{2\alpha + 1}, \qquad \text{for all } 2\beta > 1, 2\alpha > 1, \beta < \alpha.$$

Therefore, Case 1 is the optimal overall. □

### D.1.3 Phase III

In this case, the approximate loss curve has three terms (Proposition C.5 with (66))

$$\tilde{\mathcal{P}}(\tfrac{f}{d}, d) = \max\left\{ \tfrac{1}{\gamma_B}\mathcal{K}_{pp}(\tfrac{f}{d}, d), \mathcal{F}_{ac}(\tfrac{f}{d}, d), \mathcal{F}_0(\tfrac{f}{d}, d) \right\}$$
$$\asymp \left\{ \left(\tfrac{f}{d}\right)^{-2+1/(2\alpha)}, \left(\tfrac{f}{d}\right)^{-1+1/(2\alpha)} \times d^{-1}, d^{-2\alpha} \right\}. \tag{76}$$

**Proposition D.3** (Phase III: Compute-optimal Curve). *Suppose we are in Phase III, that is, $2\beta > 1$, $2\alpha > 1$, and $\beta > \alpha$. The compute-optimal curve using $\tilde{\mathcal{P}}(\tfrac{f}{d}, d)$ in (76) occurs when $\tfrac{1}{\gamma_B}\mathcal{K}_{pp}(\tfrac{f}{d}, d) = \mathcal{F}_{ac}(\tfrac{f}{d}, d)$. Precisely, the optimal $d^\star$ which minimizes $\tilde{\mathcal{P}}(\tfrac{f}{d}, d)$ is*

$$d_{Phase\ III}^\star \asymp f^{1/2},$$

*and the compute-optimal curve is*

$$\tilde{\mathcal{P}}_{Phase\ III}^\star(f) \asymp f^{(1-4\alpha)/(4\alpha)}.$$

*Proof.* Using the Remark D.2 after Lemma D.1 and Proposition C.5 with (66), we only need to check two curves: $\tfrac{1}{\gamma_B}\mathcal{K}_{pp} = \mathcal{F}_{ac}$ and $\mathcal{F}_{ac} = \mathcal{F}_0$. The curve which has the smallest (i.e., largest negative) exponent (i.e, steepest curve on a log-log plot) is the compute-optimal curve.

*Case 1: Consider $\mathcal{F}_{ac}(\tfrac{f}{d}, d) = \mathcal{F}_0(\tfrac{f}{d}, d)$.* We did this for Phase II in the proof of Proposition D.2. Thus, we have

$$\mathcal{C}_0 = 1, \quad \gamma_0 = 0, \quad p_0 = 2\alpha$$
$$\text{and} \quad \mathcal{C}_1 = 1, \quad \gamma_1 = 1 - 1/(2\alpha), \quad p_1 = 1$$

to get that the minimum is

$$d_1^\star \asymp f^{1/(2\alpha+1)}$$

and the optimal value is

$$\tilde{\mathcal{P}}_1^\star(f) \asymp f^{-2\alpha/(2\alpha+1)}.$$

*Case 2: Consider $\tfrac{1}{\gamma_B}\mathcal{K}_{pp}(\tfrac{f}{d}, d) = \mathcal{F}_{ac}(\tfrac{f}{d}, d)$.* We apply Lemma D.1 with

$$\mathcal{C}_0 = 1, \quad \gamma_0 = 2 - 1/(2\alpha), \quad p_0 = 0$$
$$\text{and} \quad \mathcal{C}_1 = 1, \quad \gamma_1 = 1 - 1/(2\alpha), \quad p_1 = 1$$

to get that the minimum is

$$d_2^\star \asymp f^{1/2}$$

and the optimal value is

$$\tilde{\mathcal{P}}_2^\star(f) \asymp f^{(1-4\alpha)/(4\alpha)}.$$

One can check that

$$\frac{1 - 4\alpha}{4\alpha} < \frac{-2\alpha}{2\alpha + 1}, \qquad \text{for all } 2\beta > 1, 2\alpha > 1, \beta > \alpha.$$

Therefore, Case 2 is the optimal overall. □

Table 7: **Summary of the compute-optimal curves for** $\tilde{\mathcal{P}}(\frac{f}{d}, d)$ **for below the high-dimensional line,** $2\alpha < 1$. This includes Phases IV, Ib, and Ic.

| | **Trade off** | **Compute-optimal Curves** |
|---|---|---|
| **Phase IVa** (Prop. D.4) | $\frac{1}{\gamma B}\mathcal{K}_{pp} = \mathcal{F}_0$ | $\tilde{\mathcal{P}}^{\star}_{\text{Phase IVa}}(f) \asymp f^{-\alpha}$ 
 $d^{\star}_{\text{Phase IVa}} \asymp f^{1/2}$ |
| **Phase IVb** (Prop. D.5) | $\frac{1}{\gamma B}\mathcal{K}_{pp} = \mathcal{F}_{pp}$ | $\tilde{\mathcal{P}}^{\star}_{\text{Phase IVb}}(f) \asymp f^{(1-2\alpha)(2\alpha+2\beta-1)/(2(2\alpha\beta+\alpha-2\beta))}$ 
 $d^{\star}_{\text{Phase IVb}} \asymp f^{(\alpha-\beta)/(2\alpha\beta+\alpha-2\beta)}$ |
| **Phase Ib** (Prop. D.6) | $\mathcal{F}_{pp} = \mathcal{F}_0$ | $\tilde{\mathcal{P}}^{\star}_{\text{Phase Ib}}(f) \asymp f^{1/2-\alpha-\beta}$ 
 $d^{\star}_{\text{Phase Ib}} \asymp f^{1/2},$ |
| **Phase Ic** (Prop. D.7) | $\mathcal{F}_{pp} = \mathcal{F}_0$ | $\tilde{\mathcal{P}}^{\star}_{\text{Phase Ic}}(f) \asymp f^{\frac{\alpha(2\alpha+2\beta-1)}{\alpha(2\beta-3)-2\beta+1}}$ 
 $d^{\star}_{\text{Phase Ic}} \asymp f^{\frac{1-2(\alpha+\beta)}{2(\alpha(2\beta-3)-2\beta+1)}}$ |

## D.2 Compute-optimal curves: Below the high-dimensional line (Phase IV, Ib, Ic), $2\alpha < 1$

This section main distinction with above the high-dimensional line section is the dependency of the learning rate on $v$. In deed, we have that $v/d \to c \in (0, \infty)$ and the learning rate is chosen so that the kernel norm is constant, i.e.,

$$\gamma \sim 2(1-2\alpha)\|\mathcal{K}\|v^{2\alpha-1} \quad \Rightarrow \quad \gamma \overset{\text{def}}{=} \tilde{\gamma} \times v^{2\alpha-1},$$

where $\tilde{\gamma}$ is the positive constant so that $\gamma = \tilde{\gamma} \times v^{2\alpha-1}$. Consequently, we also need to keep track of the learning rate in the various terms.

In this section, we assume that $v/d \to r \in (1, \infty)$ as $v, d \to \infty$.

We state for completeness the $d$ and $r$ dependency on the forcing and kernel function, including the learning rate $\gamma$. We note that these asymptotics only hold for a set of $\gamma B r$ values which depend on the spectral properties of $K$ (see the propositions listed next to the terms for details).

$$\mathcal{F}_{pp}(r) \asymp (\gamma \times r)^{-(1+\beta/\alpha)+1/(2\alpha)} \asymp \left(\frac{v^{2\alpha-1}}{d^{2\alpha-1}}\right)^{-(1+\beta/\alpha)+1/(2\alpha)} (d^{2\alpha-1} \times r)^{-(1+\beta/\alpha)+1/(2\alpha)}$$

$$\asymp (d^{2\alpha-1} \times r)^{-(1+\beta/\alpha)+1/(2\alpha)}, \quad \text{(see Proposition H.2)}$$

$$\frac{1}{\gamma B}\mathcal{K}_{pp}(r) \asymp \gamma^{-1+1/(2\alpha)} \times r^{-2+1/(2\alpha)} \asymp \left(\tilde{\gamma} \times \frac{V^{2\alpha-1}}{d^{2\alpha-1}}\right)^{-1+1/(2\alpha)} \times d^{2-1/(2\alpha)-2\alpha} \times r^{-2+1/(2\alpha)}$$

$$\asymp d^{2-1/(2\alpha)-2\alpha} \times r^{-2+1/(2\alpha)}, \quad \text{(see Proposition H.5)}$$

$$\mathcal{F}_0(r) \asymp (v^{1-1/(2\alpha)} \times d^{1/(2\alpha)})^{-2\alpha+\max\{0,1-2\beta\}}$$

$$\asymp \left(\frac{v^{1-1/(2\alpha)}}{d^{1-1/(2\alpha)}}\right)^{-2\alpha+\max\{0,1-2\beta\}} \times d^{-2\alpha+\max\{0,1-2\beta\}}$$

$$\asymp d^{-2\alpha+\max\{0,1-2\beta\}}. \quad \text{(see Proposition H.3)}$$

### D.2.1 Phase IV (a) and (b)

In these cases, the approximation loss curve is given by (Proposition C.6 with (68))

$$\tilde{\mathcal{P}}(\tfrac{f}{d}, d) \asymp \max\{\mathcal{F}_{pp}(\tfrac{f}{d}, d), \tfrac{1}{\gamma B}\mathcal{K}_{pp}(\tfrac{f}{d}, d), \mathcal{F}_0(\tfrac{f}{d}, d)\}$$

$$\asymp \max\left\{\left(\tfrac{f}{d}\right)^{-(1+\beta/\alpha)+1/(2\alpha)} \times d^{(2\alpha-1)(-(1+\beta/\alpha)+1/(2\alpha))},\right. \tag{77}$$

$$\left. d^{2-1/(2\alpha)-2\alpha} \times \left(\tfrac{f}{d}\right)^{-2+1/(2\alpha)}, d^{-2\alpha}\right\}.$$

As one crosses the $2\alpha = 1$, line the $\mathcal{F}_{ac}$ disappears and $\mathcal{F}_{pp}$ emerges. Consequently, there leaves two possible corners where the compute-optimal value could occur at. When $\alpha$ goes below $\alpha = 1/4$, the $\mathcal{K}_{pp}$ decreases.

The difference between IVa and IVb is simply where the compute-optimal occurs. In IVa, the tradeoff occurs between $\mathcal{K}_{pp}$ and $\mathcal{F}_0$, whereas in IVb, the tradeoff occurs at $\mathcal{F}_{pp}$ and $\mathcal{K}_{pp}$.

We give the compute-optimal curve for Phase IVa.

**Proposition D.4** (Phase IVa: Compute-optimal Curve). *Suppose we are in Phase IVa, that is,* $2\beta > 1$ *and* $\frac{\sqrt{2}-1}{\sqrt{2}} < \alpha < \frac{1}{2}$. *The compute-optimal curve using* $\tilde{\mathcal{P}}(\frac{\mathfrak{f}}{d}, d)$ *in* (77) *occurs when* $d^{1-2\alpha}\mathcal{K}_{pp}(\frac{\mathfrak{f}}{d}, d) = \mathcal{F}_0(\frac{\mathfrak{f}}{d}, d)$. *Precisely, the optimal* $d^\star$ *which minimizes* $\tilde{\mathcal{P}}(\frac{\mathfrak{f}}{d}, d)$ *is*

$$d^\star_{Phase\ IVa} \asymp \mathfrak{f}^{1/2},$$

*and the compute-optimal curve is*

$$\tilde{\mathcal{P}}^\star_{Phase\ IVa}(\mathfrak{f}) \asymp \mathfrak{f}^{-\alpha}.$$

*Proof.* Using the Remark D.2 after Lemma D.1 and Proposition C.6 with (68), we only need to check two curves: $\mathcal{F}_{pp} = d^{1-2\alpha} \times \mathcal{K}_{pp}$ and $d^{1-2\alpha} \times \mathcal{K}_{pp} = \mathcal{F}_0$. The curve which has the smallest (i.e., largest negative) exponent (i.e, steepest curve on a log-log plot) is the compute-optimal curve.

*Case 1: Consider* $\mathcal{F}_{pp}(\frac{\mathfrak{f}}{d}, d) = d^{1-2\alpha}\mathcal{K}_{pp}(\frac{\mathfrak{f}}{d}, d)$. We apply Lemma D.1 with

$$\mathcal{C}_0 = 1, \quad \gamma_0 = 1 + \beta/\alpha - 1/(2\alpha), \quad p_0 = (2\alpha - 1)(1 + \beta/\alpha - 1/(2\alpha))$$
$$\text{and} \quad \mathcal{C}_1 = 1, \quad \gamma_1 = 2 - 1/(2\alpha), \quad p_1 = -2 + 1/(2\alpha) + 2\alpha,$$

to get that the minimum is

$$d^\star_1 \asymp \mathfrak{f}^{(\alpha-\beta)/(2\alpha\beta+\alpha-2\beta)}$$

and the optimal value is

$$\tilde{\mathcal{P}}^\star_1(\mathfrak{f}) = c_f c_{pp} \left(\frac{c_f c_{pp}}{c_k \tilde{c}_{pp}}\right)^{(1-\alpha)(2\alpha+2\beta-1)/(2\alpha\beta+\alpha-2\beta)} \times \mathfrak{f}^{(1-2\alpha)(2\alpha+2\beta-1)/(2(2\alpha\beta+\alpha-2\beta))}.$$

*Case 2: Consider* $d^{1-2\alpha}\mathcal{K}_{pp}(\frac{\mathfrak{f}}{d}, d) = \mathcal{F}_0(\frac{\mathfrak{f}}{d}, d)$. We apply Lemma D.1 with

$$\mathcal{C}_0 = 1, \quad \gamma_0 = 0, \quad p_0 = 2\alpha$$
$$\text{and} \quad \mathcal{C}_1 = 1, \quad \gamma_1 = 2 - 1/(2\alpha), \quad p_1 = -2 + 1/(2\alpha) + 2\alpha,$$

to get that the minimum is

$$d^\star_1 \asymp \mathfrak{f}^{1/2}$$

and the optimal value is

$$\tilde{\mathcal{P}}^\star_1(\mathfrak{f}) \asymp \mathfrak{f}^{-\alpha}.$$

In this region of $\alpha$'s and $\beta$'s, Case 2 has a smaller exponent on $\mathfrak{f}$ than in Case 1. $\qquad\square$

As for Phase IVb, we have the following.

**Proposition D.5** (Phase IVb: Compute-optimal Curve). *Suppose we are in Phase IVb, that is,* $2\beta > 1$ *and* $\frac{1}{4} < \alpha < \frac{\sqrt{2}-1}{\sqrt{2}}$. *The compute-optimal curve using* $\tilde{\mathcal{P}}(\frac{\mathfrak{f}}{d}, d)$ *in* (77) *occurs when* $c_k d^{1-2\alpha}\mathcal{K}_{pp}(\frac{\mathfrak{f}}{d}, d) = c_f \mathcal{F}_{pp}(\frac{\mathfrak{f}}{d}, d)$. *Precisely, the optimal* $d^\star$ *which minimizes* $\tilde{\mathcal{P}}(\frac{\mathfrak{f}}{d}, d)$ *is*

$$d^\star_{Phase\ IVb} \asymp \mathfrak{f}^{(\alpha-\beta)/(2\alpha\beta+\alpha-2\beta)},$$

*and the compute-optimal curve is*

$$\tilde{\mathcal{P}}^\star_{Phase\ IVb}(\mathfrak{f}) \asymp \mathfrak{f}^{(1-2\alpha)(2\alpha+2\beta-1)/(2(2\alpha\beta+\alpha-2\beta))}.$$

*Proof.* The computations are exactly the same as in Proposition D.4. For the $\alpha$'s and $\beta$'s in this region, we see that Case 1 has the smaller exponent on $\mathfrak{f}$ than Case 2. $\qquad\square$

### D.2.2 Phase Ib

In this case, the approximate loss curve is given by (Proposition C.7 with (70))

$$\tilde{\mathcal{P}}(\tfrac{f}{d}, d) \asymp \max\{\mathcal{F}_{pp}(\tfrac{f}{d}, d), \mathcal{F}_0(\tfrac{f}{d}, d)\}$$
$$\asymp \max\{\left(\tfrac{f}{d}\right)^{-(1+\beta/\alpha)+1/(2\alpha)} \times d^{(2\alpha-1)(-(1+\beta/\alpha)+1/(2\alpha))}, d^{-2\alpha+1-2\beta}\}. \tag{78}$$

Note this is true for $\alpha > 1/4$, but we expect this to hold without this extra assumption.

With this, we give a description of the optimal compute curve.

**Proposition D.6** (Phase Ib: Compute-optimal Curve)**.** *Suppose we are in Phase Ib, that is, $2\beta < 1$, $\frac{1}{4} < \alpha < \frac{1}{2}$, and $\alpha + \beta > \frac{1}{2}$. The compute-optimal curve using $\tilde{\mathcal{P}}(\tfrac{f}{d}, d)$ in (78) occurs when $\mathcal{F}_{pp}(\tfrac{f}{d}, d) = \mathcal{F}_0(\tfrac{f}{d}, d)$. Precisely, the optimal $d^\star$ which minimizes $\tilde{\mathcal{P}}(\tfrac{f}{d}, d)$ is*

$$d^\star_{Phase\ Ib} \asymp f^{\frac{1}{2}},$$

*and the compute-optimal curve is*

$$\tilde{\mathcal{P}}^\star_{Phase\ Ib}(f) \asymp f^{\frac{1}{2}-\alpha-\beta}.$$

*Proof.* We apply Lemma D.1 with

$$\mathcal{C}_0 = 1, \quad \gamma_0 = 1 + \beta/\alpha - 1/(2\alpha), \quad p_0 = (2\alpha - 1)(1 + \beta/\alpha - 1/(2\alpha))$$
$$\text{and} \quad \mathcal{C}_1 = 1, \quad \gamma_1 = 0, \quad p_1 = 2\alpha - 1 + 2\beta.$$

$\square$

We expect the conclusions of Prop. D.6 to hold for the $(\alpha, \beta)$ pairs where $2\beta < 1$, $2\alpha < 1$, and $2(\alpha + \beta) > 1$.

### D.2.3 Phase Ic

In this case, the approximate loss curve is given by (Proposition C.8 with (72))

$$\tilde{\mathcal{P}}(\tfrac{f}{d}, d) \asymp \max\{\mathcal{F}_{pp}(\tfrac{f}{d}, d), \mathcal{F}_0(\tfrac{f}{d}, d)\}$$
$$\asymp \max\{\left(\tfrac{f}{d}\right)^{-(1+\beta/\alpha)+1/(2\alpha)} \times d^{(2\alpha-1)(-(1+\beta/\alpha)+1/(2\alpha))}, d^{-2\alpha}\}. \tag{79}$$

Note again that this is speculative as we do not have the bounds for the kernel. However we do believe that this is correct. With this, we give a description of the compute-optimal curve.

**Proposition D.7** (Phase Ic: Compute-optimal Curve)**.** *Suppose we are in Phase Ic, that is, $2\beta > 1$ and $0 < \alpha < \frac{1}{4}$ and suppose (79) is true. The compute-optimal curve using $\tilde{\mathcal{P}}(\tfrac{f}{d}, d)$ in (77) occurs when $\mathcal{F}_{pp}(\tfrac{f}{d}, d) = \mathcal{F}_0(\tfrac{f}{d}, d)$. Precisely, the optimal $d^\star$ which minimizes $\tilde{\mathcal{P}}(\tfrac{f}{d}, d)$ is*

$$d^\star_{Phase\ Ic} \asymp f^{\frac{1-2(\alpha+\beta)}{2(\alpha(2\beta-3)-2\beta+1)}},$$

*and the compute-optimal curve is*

$$\tilde{\mathcal{P}}^\star_{Phase\ Ic}(f) \asymp f^{\frac{\alpha(2\alpha+2\beta-1)}{\alpha(2\beta-3)-2\beta+1}}.$$

*Proof.* We apply Lemma D.1 with

$$\mathcal{C}_0 = 1, \quad \gamma_0 = 1 + \beta/\alpha - 1/(2\alpha), \quad p_0 = (2\alpha - 1)(1 + \beta/\alpha - 1/(2\alpha))$$
$$\text{and} \quad \mathcal{C}_1 = 1, \quad \gamma_1 = 0, \quad p_1 = 2\alpha.$$

$\square$

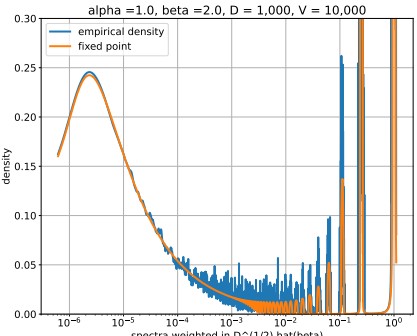

Figure 6: **Spectra of empirical and theory weighted by** $D^{1/2}\hat{\beta}$. Empirical spectra (blue) averaged over 100 randomly generated matrices $W \in \mathbb{R}^{v \times d}$. Point mass at $z = 0$ was manually removed. Theory (orange) computed using the resolvent formula (9) and solved with Newton method (10 iterations for each $z$; $z$-values were spaced at $0.1 d^{-2\alpha}$ with an imaginary part at $d^{-2\alpha}$). There is a continuous part that evolves into a pure point outliers.

# E    Spectrum of $\hat{K}$: random matrix theory

In this section, we analyze the spectrum of $\hat{K} = D^{1/2}WW^T D^{1/2}$. For this, we use standard tools from random matrix theory to derive a *fixed point equation* for the Stieljes transform of $\hat{K}$. Indeed, by knowing the Stieljes transform of $\hat{K}$, one can recover the spectral properties.

In particular, we will need the spectra of $\hat{K}$ decomposes into 3 parts:

1. *Point mass at $z = 0$:* There will be a point mass at $z = 0$ of mass $v - d$ for trivial reasons since $v \gg d$.

2. *Pure point outliers:* There will be a set of outliers, the pure point spectra, which are at constant order and nearly equal to $j^{-2\alpha}$ for $j = 1, 2, \ldots$,

3. *Absolutely continuous part:* The spectral bulk, the absolutely continuous part, which form a density on a shrinking window.

In fact, we will not need to give a complete picture about all the spectra.

## E.1    Self-consistent approximation for $(\hat{K} - z)^{-1} = (D^{1/2}WW^T D^{1/2} - z)^{-1}$.

In this section, we state the deterministic equivalent for the random matrix $(\hat{K} - z)^{-1}$ and give some properties of its "self-consistent spectra." The starting point for this is the self-consistent equation

$$m(z) = \frac{1}{1 + \frac{1}{d}\sum_{j=1}^{V} \frac{j^{-2\alpha}}{j^{-2\alpha}m(z) - z}} \quad \text{where} \quad (D^{1/2}WW^T D^{1/2} - z)_{jj}^{-1} \longleftrightarrow \frac{1}{j^{-2\alpha}m(z) - z}. \tag{80}$$

The identification $\longleftrightarrow$ can be made rigorous by showing that

$$\left|\mathrm{Tr}\left(A(D^{1/2}WW^T D^{1/2} - z)^{-1}\right) - \mathrm{Tr}\left(A\,\mathrm{Diag}(j^{-2\alpha}m(z) - z : j)^{-1}\right)\right| \xrightarrow[d\to\infty]{\mathrm{Pr}} 0,$$

for deterministic sequences of test matices $\{A\}$ with bounded nuclear norm and generally with very high probability in $d$. We note that we would need a more precise quantification of errors to be useful for establishing the scaling law for the actual random matrices. In Figure 6, we solve the theoretical spectrum by solving the fixed point equation for $m(z)$ using a Newton method on a grid of complex $z$.

The function $m$ can also be related to the trace of $(\hat{K} - z)^{-1}$. From the definition of $m$, we can derive the explicit representation theorem.

**Lemma E.1.** *For any $v, d$ and $z \in \mathbb{C}$ with $\mathrm{Im}(m(z)) > 0$*

$$\sum_{j=1}^{v} \frac{1}{j^{-2\alpha}m(z) - z} = \frac{-v + (1 - m(z))d}{z}.$$

*Proof.* Multiplying through by $z$ we should evaluate

$$\sum_{j=1}^{v} \frac{z}{j^{-2\alpha}m(z) - z}.$$

Adding and subtracting $j^{-2\alpha}m(z)$

$$\sum_{j=1}^{v} \frac{z}{j^{-2\alpha}m(z) - z} = -v + m(z)\sum_{j=1}^{v} \frac{j^{-2\alpha}}{j^{-2\alpha}m(z) - z}.$$

Using the definition of $m$,

$$\sum_{j=1}^{v} \frac{j^{-2\alpha}}{j^{-2\alpha}m(z) - z} = d\left(\frac{1}{m(z)} - 1\right).$$

Substituting this in completes the proof. $\qquad\square$

While an explicit solution of $m$ is unavailable, we can derive many properties of $m$, starting with:

**Proposition E.1.** *Suppose $X$ is a real random variable compactly supported in $[0, \infty)$. For every $z \in \mathbb{H} = \{z \in \mathbb{C} : \operatorname{Im} z > 0\}$, there is a unique solution of* (80) *satisfying* $\operatorname{Im} m(z) < 0$. *Moreover, this solution is an analytic function, and it can be solved by iterating the fixed point map*

$$(m(z) : z \in \mathbb{H}) \mapsto \left(\frac{1}{1 + \frac{v}{d}\mathbb{E}\left(\frac{X}{Xm(z) - z}\right)} : z \in \mathbb{H}\right)$$

*initialized with $m \equiv 1$. Furthermore, if we consider the equation for $z \in \mathbb{H}$*

$$F(m; z) \stackrel{def}{=} m + \frac{v}{d}\mathbb{E}\left(\frac{Xm}{Xm - z}\right) = 1,$$

*then this is solved uniquely by $m(z)$ and moreover it is stable in that $\partial_m F \neq 0$ in a neighborhood of the solution.*

*Proof.* Let $G$ be the mapping

$$G(m; z) \stackrel{def}{=} \frac{1}{1 + \frac{v}{d}\mathbb{E}\left(\frac{X}{Xm(z) - z}\right)}.$$

For each fixed $z \in \mathbb{H}$, this is a strict self map into the lower half-plane of $m$. Self-maps are strict contractions in the hyperbolic metric (from the Schwarz-Pick lemma), and thus there is a unique solution of $m(z) = G(m(z); z)$.

We now introduce $F$ according to the formula

$$F(m; z) = m + \frac{v}{d}\mathbb{E}\left(\frac{Xm}{Xm - z}\right),$$

so that $F(m; z) = m/G(m; z)$. Introduce the stability operator

$$\partial_m F = 1 - \frac{v}{d}\mathbb{E}\left(\frac{Xz}{(Xm - z)^2}\right).$$

Then we have

$$\partial_m F = \frac{G - m\partial_m G}{G^2}.$$

By the Schwarz-Pick lemma (in the half-plane version), we have

$$|\partial_m G| < \frac{|\operatorname{Im} G(m; z)|}{|\operatorname{Im} m|},$$

and hence in some sufficiently small neighborhood of $G(m(z)) = m(z)$, we therefore have

$$|\partial_m G| < 1.$$

Hence also, in a sufficiently small neighborhood of $m(z)$

$$\partial_m F = m\frac{1 - \partial_m G}{G^2} + \frac{G - m}{G^2} \neq 0.$$

$\qquad\square$

While this proposition does not state what happens on the real line, in regions of the line where $m$ has a finite, real-valued limit, it agrees with its reflection in the lower half-plane. Hence in any open subset of $\mathbb{R}$ where $\lim_{\eta \to 0} m(x + i\eta)$ exists and is real, $m$ will be analytic.

From Proposition E.1, we can derive some explicit estimates on $m$, which will be sufficient for deriving the estimates on the forcing and kernel functions. We summarize these properties in the following:

**Proposition E.2.** *Let $X$ be any random variable with support in $(0, 1]$. Then the following hold:*

1. *Near 0: Suppose that $a < 0$ is a real-valued solution of*

$$\frac{v}{d}\mathbb{E}\left(\frac{Xa}{Xa - 1}\right) = 1.$$

*Then $m$ is analytic in a neighborhood of $z = 0$ and $m(z) = az + O(z^2)$. If furthermore if for some interval $[0, z_0]$ the equation*

$$-za + \frac{v}{d}\mathbb{E}\left(\frac{Xa}{Xa - 1}\right) = 1$$

*is solvable for $a < 0$, then $m$ is analytic in a complex neighborhood of the whole interval $[0, z_0]$.*

2. *Far away: Let $\varrho_0(z)$ be the distance of $z$ to $[0, 1]$. Suppose that $z$ is such that $16\frac{v}{d}\mathbb{E}\, X \leq \varrho_0(z)^2$. Then for some absolute constant $C > 0$,*

$$|m(z) - 1| \leq 8\frac{v}{d}\frac{\mathbb{E}\, X}{\varrho_0(z)}.$$

*Moreover suppose that $\left|\frac{v}{d}\mathbb{E}\left(\frac{X}{X-z}\right)\right| \leq \epsilon < \frac{1}{4}$, and suppose that*

$$\delta \overset{def}{=} \sup_{m:|m-1|\leq 2\epsilon} \frac{v}{d}\mathbb{E}\left(\frac{X^2}{|Xm - z|^2}\right) \leq \frac{1}{4} \quad and \quad \sup_{m:|m-1|\leq 2\epsilon} \frac{v}{d}\mathbb{E}\left(\frac{X|z|}{|Xm - z|^2}\right) \leq \frac{1}{4}.$$

*Then*

$$\left|m(z) - 1 + \frac{v}{d}\mathbb{E}\left(\frac{X}{X - z}\right)\right| \leq 2\delta\left|\frac{v}{d}\mathbb{E}\left(\frac{X}{X - z}\right)\right|.$$

We need a lemma on stability of solutions.

**Lemma E.2.** *Suppose $F$ is an analytic map of $-\mathbb{H} \to \mathbb{C}$, Suppose there is a $c, \delta > 0$ and $m_0 \in -\mathbb{H}$ so that for all $m \in -\mathbb{H}$ with $|m - m_0| \leq \delta$ and $\delta < |\operatorname{Im} m_0|$,*

$$|\partial_m F(m, z)| \geq c > 0$$

*If $m_0$ is an approximate solution of $F(m) = 1$ in that it satisfies*

$$|1 - F(m_0)| < c\delta,$$

*then there is an solution $m \in -\mathbb{H}$ with $F(m) = 1$ so $|m - m_0| \leq \delta$.*

*Proof.* We introduce an ODE (which is the continuous limit of the damped Newton's method)

$$\frac{\mathrm{d}m_t}{\mathrm{d}t} = -\frac{1 - F(m_t)}{\partial_m F(m_t)}, \quad \text{where} \quad m_t|_{t=0} = m_0.$$

Then we note that along this ODE

$$\frac{\mathrm{d}(1 - F(m_t))}{\mathrm{d}t} = \partial_m F(m_t)\frac{\mathrm{d}m_t}{\mathrm{d}t} = -(1 - F(m_t, z)).$$

Hence we have $(1 - F(m_t)) = (1 - F(m_0))e^{-t}$ for however long the ODE exists. In particular, if we have that on some open set $U$ of admissible $m$ containing $m_0$ that

$$|\partial_m F(m)| \geq c,$$

then provided the ODE does not exit $U$,

$$|m - m_0| = |m_\infty - m_0| \leq \int_0^\infty |(1 - F(m_0))| e^{-t}/c = |(1 - F(m_0))|/c.$$

Hence as there is a neighborhood of $m_0$ of size $\delta$ on which $\partial_m F(m)| \geq c$ then as $|(1 - F(m_0, z))| < c\delta$ we have

$$|m - m_0| \leq |(1 - F(m_0, z))|/c = \delta.$$

$\square$

*Proof of Proposition.* **Part 1, Near 0**: For the component near 0, we consider a change of variables and look at $q(z) = m(z)/z$ which is therefore the unique solution of the fixed point equation:

$$F(q(z), z) := zq(z) + \frac{v}{d} \mathbb{E}\left(\frac{Xq(z)}{Xq(z) - 1}\right) = 1.$$

Then by hypothesis, we have a solution at $z = 0$ given by $F(a, 0) = 1$. We wish to continue this solution to a neighborhood of $(a, 0)$, and so it would suffice to know that the differential equation

$$\partial_q F(q, z) \frac{\mathrm{d}q}{\mathrm{d}z} + \partial_z F = 0$$

has a solution in a neighborhood of the point. Solving for the derivative of $q$,

$$\frac{\mathrm{d}q}{\mathrm{d}z} = \frac{-q}{\partial_q F(q, z)}, \quad \text{where} \quad \partial_q F(q, z) = z - \frac{v}{d} \mathbb{E}\left(\frac{X}{(Xq(z) - 1)^2}\right).$$

We note that if we solve the equation along the real line, then $q$ stays real–valued. Furthermore, for all $q$ with $q < 0$ we have

$$\partial_q F(q, z) = z - \frac{v}{d} \mathbb{E}\left(\frac{X}{(Xq(z) - 1)^2}\right).$$

By analyticity, we can extend this solution into a neighborhood in the upper half plane and on an interval of the real line where this solvable. Hence $m$ is analytic in this neighborhood and has boundary values given by $m(z)/z \to a$.

**Part 2, Far away**: For the other parts, we use that $m$ is the solution of

$$F(m(z), z) := m(z) + \frac{v}{d} \mathbb{E}\left(\frac{Xm(z)}{Xm(z) - z}\right) = 1.$$

Hence we have

$$\frac{\mathrm{d}m}{\mathrm{d}z} = \frac{-\partial_z F(m, z)}{\partial_m F(m, z)} = \frac{-\frac{v}{d} \mathbb{E}\left(\frac{Xm(z)}{(Xm(z) - z)^2}\right)}{1 - \frac{v}{d} \mathbb{E}\left(\frac{Xz}{(Xm(z) - z)^2}\right)}.$$

Define a distance

$$f\varrho(z) := \varrho_\delta(z) := \inf\{|Xm - z| : |m - 1| \leq \delta, X \in [0, 1]\}.$$

Provided that $|m - 1| \leq \delta$ and $\frac{v}{d}(\mathbb{E}\,X)|z| \leq \varrho(z)^2/2$ (which is trivially satisfied if $\frac{v}{d}(\mathbb{E}\,X)/\varrho(z) \leq \frac{1}{2}$)

$$\left|\frac{\mathrm{d}m}{\mathrm{d}z}\right| \leq 2(1 + \delta) \frac{v}{d} \frac{\mathbb{E}(X)}{\varrho(z)^2}.$$

For any $z$ with $\varrho(z) > 0$, we can find a unit speed geodesic $\sigma(t)$ connecting $z$ to $\infty$ so that $\varrho(\sigma(t)) = \varrho(z) + t$ (this will just be a straight line). Then along this geodesic, $\frac{v}{d}(\mathbb{E}\,X)|\sigma(t)| \leq \varrho(\sigma(t))^2/2$, since

$$\tfrac{v}{d}(\mathbb{E}\,X)|\sigma(t)| \leq \tfrac{v}{d}(\mathbb{E}\,X)(|z| + t) \leq \tfrac{v}{d}(\mathbb{E}\,X)(|z| + t|z|/\varrho(z)) \leq (\varrho(z))^2(1 + t/\varrho(z))/2 \leq (\varrho(z) + t)^2/2.$$

Hence along the geodesic, and provided that $|m - 1| \leq \delta$, we conclude

$$\left|\frac{\mathrm{d}m}{\mathrm{d}z}(\sigma(t))\right| \leq 2(1 + \delta) \frac{v}{d} \frac{\mathbb{E}(X)}{(\varrho(z) + t)^2}.$$

Integrating along this line segment from infinity, we conclude that provided the right hand side is less than $\delta$,

$$\left| m(z) - 1 \right| \leq \int_0^\infty \left| \frac{\mathrm{d}m}{\mathrm{d}z}(\sigma(t)) \right| \mathrm{d}t \leq 2(1+\delta) \frac{v}{d} \frac{\mathbb{E}\, X}{\varrho(z)}.$$

For the second conclusion, we use Lemma E.2 with this $F$ (holding $z$ fixed). We let $m_0 = 1 - \frac{v}{d} \mathbb{E}\left( \frac{X}{(X-z)} \right)$, and we consider an $2\epsilon$ neighborhood of $1$, $\mathcal{U}$ By assumption, on $\mathcal{U}$

$$\partial_m F(m, z) = 1 - \frac{v}{d} \mathbb{E}\left( \frac{Xz}{(Xm(z) - z)^2} \right)$$

satisfies $|\partial_m F| \geq \frac{3}{4}$. We also have that

$$|1 - F(m_0, z)| = \left| \frac{v}{d} \mathbb{E}\left( \frac{X^2(1 - m_0)}{(Xm_0(z) - z)(X - z)} \right) \right|.$$

Hence applying Cauchy-Schwarz

$$|1 - F(m_0, z)| \leq |1 - m_0|\delta = \epsilon\delta.$$

The conclusion now follows from Lemma E.2.

$\square$

## E.2  The region near $0$ and the spectral bulk

We now bound the contribution of the region near $0$.

**Proposition E.3.** *The function $m(z)$ is analytic in a neighborhood of $z = 0$ of radius $c(\alpha)d^{-2\alpha}$ for some $c(\alpha) > 0$. Furthermore, $m$ is negative on $(0, cd^{-2\alpha})$, vanishes at $0$, and has $|m'(0) + \kappa(v/d)d^{2\alpha}| \leq Cd^{2\alpha-1}$ for all $d$ sufficiently large where*

$$\kappa(v/d) \quad \text{solves} \quad \int_0^{v/d} \frac{\kappa\, \mathrm{d}x}{\kappa + x^{2\alpha}} = 1.$$

*Moreover, we introduce $f(z; V/d)$ where $f : \mathbb{H} \to -\mathbb{H}$ which solves*

$$f(z; a) + \int_0^a \frac{f(z; a)\, \mathrm{d}x}{f(z; a) - x^{2\alpha}z} = 1.$$

*This extends analytically to the interval $[0, c)$. Then $f$ and $m$ are close in that for any compact subset $K \subset \left( \mathbb{C} \setminus ([c, \infty] \cup [-\infty, 0]) \right)$, we have*

$$|m(zd^{-2\alpha}) - f(z)| \leq C(K)/d.$$

*We furthermore have that in the case $2\beta < 1$*

$$\left| \sum_{j=1}^v \frac{j^{-2\alpha-2\beta}}{j^{-2\alpha}m(zd^{-2\alpha}) - zd^{-2\alpha}} - d^{1-2\beta} \int_0^a \frac{x^{-2\beta}\, \mathrm{d}x}{f(z) - zx^{2\alpha}} \right| \leq C(K),$$

*and in the case $2\beta > 1$*

$$\left| \sum_{j=1}^v \frac{j^{-2\alpha-2\beta}}{j^{-2\alpha}m(zd^{-2\alpha}) - zd^{-2\alpha}} - \frac{c_\beta}{f(z)} \right| \leq C(K)d^{-\min\{1, 2\alpha, 2\beta-1\}}.$$

*Proof.* For the first part, we look to apply Proposition E.2 part 1. The equation we need to solve is

$$-za + \frac{1}{d}\sum_{j=1}^v \frac{j^{-2\alpha}a}{j^{-2\alpha}a - 1} = 1,$$

for $a < 0$. We change variables by setting $a = -d^{2\alpha}\varkappa$ and $z = d^{-2\alpha}\mathfrak{z}$, in terms of which

$$\mathfrak{z}\varkappa + \frac{1}{d}\sum_{j=1}^v \frac{(j/d)^{-2\alpha}\varkappa}{(j/d)^{-2\alpha}\varkappa + 1} = 1. \tag{81}$$

By monotonicity, for $\varkappa$ positive,

$$\int_{1/d}^{v/d} \frac{\varkappa\,dx}{\varkappa + x^{2\alpha}} \le \frac{1}{d}\sum_{j=1}^{v} \frac{(j/d)^{-2\alpha}\varkappa}{(j/d)^{-2\alpha}\varkappa + 1} \le \int_0^{v/d} \frac{\varkappa\,dx}{\varkappa + x^{2\alpha}},$$

and moreover the lower bound is only less than the upper bound by at most $\frac{1}{d}$ uniformly in $\varkappa > 0$. In the case that $2\alpha < 1$, we can bound for $\varkappa \in [0,1]$

$$\varkappa\frac{(v/d)}{1 + (v/d)^{2\alpha}} \le \int_0^{v/d} \frac{\varkappa\,dx}{\varkappa + x^{2\alpha}} \le \varkappa\frac{(v/d)^{1-2\alpha}}{1 - 2\alpha}.$$

Hence there there is an interval $[0, c_0]$ (bounded solely in terms of $(v/d)$) on which (81) is solvable and is uniformly bounded away from 0 over all $d$. Hence, the solution of $\kappa$ of $\int_0^{v/d} \frac{\kappa\,dx}{\kappa + x^{2\alpha}} = 1$ satisfies

$$\frac{1}{d}\sum_{j=1}^{v} \frac{(j/d)^{-2\alpha}\kappa}{(j/d)^{-2\alpha}\kappa + 1} \in [1 - \tfrac{1}{d}, 1 + \tfrac{1}{d}].$$

Following the same bounds on $\int_0^{V/d} \frac{dx}{\kappa + x^{2\alpha}}$ shown above, we conclude that the true solution $\varkappa$ of (81) with $\mathfrak{z} = 0$ satisfies $|\varkappa - \kappa| = O(1/d)$. This concludes the proof when $2\alpha < 1$.

In the case that $2\alpha > 1$,

$$\int_0^{v/d} \frac{\varkappa\,dx}{\varkappa + x^{2\alpha}} \le \int_0^{\infty} \frac{\varkappa\,dx}{\varkappa + x^{2\alpha}} = \varkappa^{1/(2\alpha)}\int_0^{\infty} \frac{dx}{1 + x^{2\alpha}}.$$

On the other hand for $\varkappa \in [0,1]$

$$\int_0^{v/d} \frac{\varkappa\,dx}{\varkappa + x^{2\alpha}} \ge \varkappa^{1/(2\alpha)}\int_0^{(v/d)\varkappa^{-1/(2\alpha)}} \frac{dx}{1 + x^{2\alpha}} \ge \varkappa^{1/(2\alpha)}\int_0^{(v/d)} \frac{dx}{1 + x^{2\alpha}}$$

and hence once more there is an interval $[0, c_0]$ independent of $V/d$ on which this is solvable and moreover the conclusions now follow in the same way as in the case that $2\alpha < 1$.

**Convergence to f.** The existence and uniqueness of $f$ follows from Proposition E.1, where we define

$$\mathcal{F}(f; z) \overset{\text{def}}{=} f + \int_0^a \frac{f\,dx}{f - x^{2\alpha}z} = 1,$$

(making appropriate choices of $v/d$ and $X$).

We further have, from the previous part, that $f$ takes negative values on an the interval $(0, c)$. In what follows we fix a compact set $K \subset \big(\mathbb{C} \setminus ([c, \infty] \cup [-\infty, 0])\big)$. Further, we claim the stability operator

$$\partial_f \mathcal{F} = 1 - \int_0^a \frac{x^{2\alpha}z\,dx}{(f(z;a) - x^{2\alpha}z)^2}$$

is nonvanishing on $K$ in a neighborhood of $f$. Off of the real line, this follows from Proposition E.1. On the real line, it follows from monotonicity of $\mathcal{F}$ for $f < 0$.

Hence it follows that on $K$, there is a constant $C(K)$ and a $\delta_0 > 0$ so that if $z \in K$ and $m$ satisfies

$$|\mathcal{F}(m; z) - 1| < \delta_0,$$

then

$$|m - f| \le C(K)|\mathcal{F}(m; z) - 1|.$$

Define

$$S(m; z, \beta) \overset{\text{def}}{=} \sum_{j=1}^{v} \frac{j^{-2\alpha - 2\beta}}{j^{-2\alpha}m - zd^{-2\alpha}}.$$

Then with $\beta = 0$

$$\frac{1}{d}S(m; z, 0) = \frac{1}{d}\sum_{j=1}^{v} \frac{j^{-2\alpha}}{j^{-2\alpha}m - zd^{-2\alpha}}.$$

We will define $a = v/d$, and we will define $\Delta$ as the separation

$$\Delta \overset{\text{def}}{=} \min\left\{\min\{t \in [0, (v/d)^{2\alpha}] : |m - zt|\}, 1\right\}$$

Then by bounding the errors in a trapezoid rule approximation

$$\left|\frac{1}{d}S(m; z, 0) - \int_0^a \frac{\mathrm{d}x}{m - zx^{2\alpha}}\right| \lesssim \frac{1}{d\Delta^2},$$

where we have used a bound on the $x$ derivative of the integrand

$$\int_0^a \frac{|z|2\alpha x^{2\alpha-1}\,\mathrm{d}x}{|m - zx^{2\alpha}|^2} \lesssim \frac{1}{\Delta^2}$$

(which relies on $z$ being bounded away from $0$ and on $z$ being bounded in modulus). Hence if $m(zd^{-2\alpha})$ is the solution of

$$m + \frac{m}{d}S(m; z, 0) = 1,$$

then we have

$$|\mathcal{F}(m; z) - 1| = \left|m + \int_0^a \frac{m\,\mathrm{d}x}{m - zx^{2\alpha}} - 1\right| \lesssim \frac{|m|d^{-1}}{\Delta(m)},$$

provided $m$ is bounded.

To see that $m$ remains bounded, we let $\partial_m F$ be the stability operator of the equation

$$1 = F(m; z) = m + \frac{m}{d}S(m; z, 0).$$

Then once more

$$\partial_m F = 1 + \frac{1}{d}S(m; z, 0) + \frac{m}{d}\partial_m S(m; z, 0).$$

Approximating the sum for $\partial_m S$

$$|\partial_m F(m; z) - \partial_m \mathcal{F}(m; z)| \lesssim d^{-1}\left(\frac{1}{\Delta^2} + \frac{|m|}{\Delta^3}\right).$$

Differentiating the fixed point equation, we have the differential equation for $m$

$$\frac{\mathrm{d}m}{\mathrm{d}z} = \frac{m}{z}\frac{1 - \partial_m F(m; z)}{\partial_m F(m; z)}.$$

As the same equation holds for $f$ and is non-degenerate in a neighborhood of $f$ (as the stability operator does not vanish in a neighborhood of the solution), we conclude that

$$\left|\frac{\mathrm{d}m}{\mathrm{d}(zd^{-2\alpha})}\right| = O(1) \quad \text{and} \quad |\mathcal{F}(m; z) - 1| \lesssim \frac{d^{-1}}{\Delta(m)}$$

uniformly on compact sets for all $d$ sufficiently large

**Sum formula.** Hence having approximated $f$, we can turn to estimating $S(m; z, \beta)$. When $2\beta < 1$, we may repeat the Riemann sum approximation argument. Specifically, we have

$$\left|d^{2\beta-1}S(m; z, \beta) - \int_0^a \frac{x^{-2\beta}\,\mathrm{d}x}{m - zx^{2\alpha}}\right| \lesssim \frac{1}{d\Delta^2},$$

where to bound the errors, we now must estimate

$$d^{2\beta-1}\int_1^v \left|\frac{\mathrm{d}}{\mathrm{d}x}\frac{x^{-2\alpha-2\beta}}{x^{-2\alpha}m - zd^{-2\alpha}}\right|\mathrm{d}x \lesssim d^{2\beta-1}\int_1^v \left(\left|\frac{x^{-2\alpha-2\beta-1}}{x^{-2\alpha}m - zd^{-2\alpha}}\right| + \left|\frac{x^{-4\alpha-2\beta-1}m}{(x^{-2\alpha}m - zd^{-2\alpha})^2}\right|\right)\mathrm{d}x.$$

Setting $x = wd$, we arrive at

$$\left|d^{2\beta-1}S(m; z, \beta) - \int_0^a \frac{x^{-2\beta}\,\mathrm{d}x}{m - zx^{2\alpha}}\right| \lesssim d^{-1}\int_{(1/d)}^a \left(\left|\frac{w^{-2\beta-1}}{m - zw^{2\alpha}}\right| + \left|\frac{w^{-2\beta-1}m}{(m - zw^{2\alpha})^2}\right|\right)\mathrm{d}x$$

$$\lesssim d^{2\beta-1}\left(\frac{1}{\Delta} + \frac{|m|}{\Delta^2}\right).$$

We may subsequently replace in this expression $m$ by $f$.

In the case that $2\beta > 1$, we subtract from $S$ the divergence $c_\beta/m$ and then express

$$S(m; z, \beta) - 1/m \sum_{j=1}^{v} j^{-2\beta} \sum_{j=1}^{v} \left( \frac{j^{-2\alpha-2\beta}}{j^{-2\alpha}m - zd^{-2\alpha}} - \frac{j^{-2\alpha-2\beta}}{j^{-2\alpha}m} \right).$$

Bounding the difference leads to

$$|S(m; z, \beta) - c_\beta/m| \lesssim c_\beta d^{-2\alpha}/\Delta + d^{1-2\beta}/|m|.$$

$\square$

**Remark E.1.** *There is an exactly solvable case where even more can be said. Note that when $2\alpha > 1$ and $v/d = \infty$, the equation for $f$ becomes*

$$f + \int_0^\infty \frac{f \, dx}{f - x^{2\alpha}z} = 1.$$

*Changing variables (which requires a contour deformation which restricts the branches considered) by letting $x^{2\alpha}z = -fy^{2\alpha}$, so that $x = (-f/z)^{1/(2\alpha)}y$. Then*

$$f + (-f/z)^{1/(2\alpha)} \int_0^\infty \frac{dx}{1 + x^{2\alpha}} = 1$$

*Hence with $c_\alpha = \int_0^\infty \frac{dx}{1+x^{2\alpha}}$, we have that $f$ is the solution of*

$$f + (-f/z)^{1/(2\alpha)}c_\alpha = 1.$$

*If for example $\alpha = 1$, then with $g = (-f)^{1/2}$ we have $g$ satisfies the quadratic equation*

$$-g^2 + gz^{-1/2}c_1 = 1,$$

*or solving*

$$g = z^{-1/2}\frac{c_1}{2} \pm \sqrt{c_1^2 z^{-1}/4 - 1},$$

*with $\pm$ chosen so that $\operatorname{Im} g \geq 0$ and $\operatorname{Re} g > 0$. We note that $c_1 = \frac{\pi}{2}$ and conclude that*

$$f = -\frac{1}{z}\left( \frac{\pi}{4} \pm \sqrt{(\pi/4)^2 - z} \right)^2,$$

*with the branch chosen to ensure $\operatorname{Im} f < 0$ when $\operatorname{Im} z > 0$.*

### E.3   The mesoscopic region

We will need the following technical estimate on sums over lattice points.

**Lemma E.3.** *Suppose that $z$ and $w$ are complex numbers and $-z/w \notin \mathbb{Z}$*

$$\operatorname{pv} \sum_n \frac{1}{wn + z} = -\frac{\pi}{w} \cot(\pi z/w). \tag{82}$$

*Moreover, if we suppose $|\operatorname{Im}(z/w)| \geq |\operatorname{Re}(z/w)|$ then there is an absolute constant $C > 0$ so that for any $N \in \mathbb{N}$*

$$\left| \operatorname{pv} \sum_n \frac{1}{wn + z} - \sum_{n=-N}^{N} \frac{1}{wn + z} \right| \leq \frac{C|z|}{|w|^2 N}$$

*Proof.* Note that we may remove a factor $\frac{1}{w}$ from all statements and instead look at the case (with $y = -z/w$)

$$\operatorname{pv} \sum_n \frac{1}{wn + z} = \frac{1}{w} \operatorname{pv} \sum_n \frac{1}{n - y}.$$

Then by a residue computation (applied to the function $\pi \cot(\pi z)\frac{1}{z-y}$)

$$\frac{1}{w}\,\mathrm{pv}\sum_n \frac{1}{n-y} = \frac{1}{w}\pi\cot(\pi y) = -\frac{\pi}{w}\cot(\pi z/w),$$

where we have used that $\cot$ is odd.

Now by pairing terms, we have

$$\left|\mathrm{pv}\sum_n \frac{1}{wn+z} - \sum_{n=-N}^{N}\frac{1}{wn+z}\right| \le \sum_{n=N+1}^{\infty}\left|\frac{1}{wn+z}+\frac{1}{-wn+z}\right|.$$

Making a common fraction, we have

$$\left|\mathrm{pv}\sum_n \frac{1}{wn+z} - \sum_{n=-N}^{N}\frac{1}{wn+z}\right| \le \sum_{n=N+1}^{\infty}\left|\frac{2z}{-(wn)^2+z^2}\right|$$

$$\le \frac{|2z|}{|w|^2}\sum_{n=N+1}^{\infty}\left|\frac{1}{-n^2+(z/w)^2}\right|.$$

Now $\mathrm{Re}(z/w)^2 < 0$, and hence the claim follows. $\qquad\square$

**Proposition E.4.** *Let $\alpha,\beta \ge 0$ with neither equal to $\frac{1}{2}$. We further assume $2\alpha+\beta \ne \frac{1}{2}$. For $u,\eta,a,b \ge 0$ consider with $m = a - ib$*

$$A + iB = \sum_{j=1}^{v} \frac{j^{-2\alpha-2\beta}}{-u-i\eta+j^{-2\alpha}m}$$

*for real $A,B$. We suppose that the $a,b,u,\eta$ and $\epsilon$ satisfy*

1. *$|1-a| \le \frac{1}{2}$,*

2. *$\eta + ub < \epsilon^4 u$,*

3. *$0 \le b$,*

4. *$\log(1/\epsilon)u^{1+1/(2\alpha)} \le c\eta$,*

5. *$\eta \le \epsilon u$,*

6. *$0 < u < c$.*

*Then there is an $\epsilon_0 > 0$, $c > 0$, and $C > 0$ so that if $\epsilon \in (0,\epsilon_0)$ such that*

$$\left|B - \frac{\pi(u/a)^{\beta/\alpha-1/(2\alpha)}}{2\alpha a} - c_\beta\frac{b}{a^2} - o_{\alpha+\beta}\frac{(\eta+ub)v^{1-2\alpha-2\beta}}{u^2(1-2\alpha-2\beta)}\right|$$

$$\le C\left(\epsilon u^{\beta/\alpha-1/(2\alpha)} + c_\beta(\eta+b)u\log(1/u) + \epsilon o_{\alpha+\beta}\frac{(\eta+ub)v^{1-2\alpha-2\beta}}{u^2}\right),$$

*where $c_\beta = \sum_{j=1}^{\infty}j^{-2\beta}$ (if $\beta > \frac{1}{2}$) or $c_\beta = 0$ otherwise and where $o_{\alpha+\beta}$ is the indicator function of $\alpha+\beta < \frac{1}{2}$. Furthermore, let $\mathcal{A} = \mathcal{A}(u+i\eta)$ be the same sum with $m \to 1$. Then*

$$|A - \mathcal{A}| \le C|1-a|\left(\frac{1}{\epsilon}u^{\beta/\alpha-1/(2\alpha)} + c_\beta + o_{\alpha+\beta}\epsilon\frac{v^{1-2\alpha-2\beta}}{u}\right),$$

*and moreover*

$$|\mathcal{A}| \le C\left(u^{\beta/\alpha-1/(2\alpha)} + c_\beta + o_{\alpha+\beta}\frac{v^{1-2\alpha-2\beta}}{u}\right).$$

*Proof.* We look to estimate the expression

$$A + iB = \sum_{j=1}^{v}\frac{j^{-2\alpha-2\beta}}{-u-i\eta+j^{-2\alpha}m},$$

on the regime considered, where $m = a - ib$. The dominant contribution of the sum will either come from $j^{-2\alpha}a \approx u$ or possibly, when $2\beta > 1$, from small $j$. So the analysis will be done by separately considering windows around the transition window $j^{-2\alpha}a \approx u$, and another analysis for large/small $j$. We use the notation for $I \subset \{1, 2, \ldots, v\}$ that $A_I$ and $B_I$ are the restrictions of this sum to the range of $j \in I$.

**The transition window.** We begin by setting $j_0$ to be the integer which minimizes $|j_0^{-2\alpha}a - u|$. We can estimate this difference, noting that

$$|j_0^{-2\alpha}a - u| \le \max\{|j_0^{-2\alpha} - (j_0 \pm 1)^{-2\alpha}|a\} \le C(\alpha)j_0^{-2\alpha-1}a \le C(\alpha)'u^{1+1/(2\alpha)}. \tag{83}$$

We can estimate $j^{-2\alpha}$ by Taylor approximation, giving

$$j^{-2\alpha} = j_0^{-2\alpha} - 2\alpha(j - j_0)j_0^{-2\alpha-1} + O((j - j_0)^2 j_0^{-2\alpha-2}).$$

Now we divide $j$ according to whether

$$(j_0^{-2\alpha}a - j^{-2\alpha}a)^2 \le M(\eta + j_0^{-2\alpha}b)^2$$

or if not, for a large $M = M(\epsilon) \asymp 1/\epsilon^2$. Let $I$ the largest possible symmetric interval of $j$ around $j_0$ that satisfies the above display.

On this interval, we would like to justify that the Taylor approximation holds. For this, we shall require that $\sqrt{M}(\eta + j_0^{-2\alpha}b)j_0^{2\alpha} \le \epsilon$. Note under this condition

$$|j - j_0|/j_0 + O((j - j_0)^2 j_0^{-2}) \asymp |j_0^{-2\alpha} - j^{-2\alpha}|j_0^{2\alpha} \le \sqrt{M}(\eta + j_0^{-2\alpha}b)j_0^{2\alpha} \le \epsilon.$$

Thus the largest difference of $|j - j_0|$ on $I$ is bounded above, up to constants, by $\sqrt{M}(\eta + ub)u^{-1-1/(2\alpha)}$. Hence the Taylor approximation is justified in that on $I$

$$|j^{-2\alpha} - j_0^{-2\alpha}| = (2\alpha)|j - j_0|u^{1+1/(2\alpha)}(1 + O(\epsilon)),$$

with the implied constants bounded in terms of $|1 - a|$ and $c$. It follows that for terms outside of $I$, we have $(j_0^{-2\alpha}a - j^{-2\alpha}a)^2 > c'M(\eta + j_0^{-2\alpha}b)^2$ for some absolute $c'$ (provided $c(\alpha)$ was picked sufficiently small).

The contribution of $I$ terms now follows the same path as was done in the first case:

$$\sum_{j \in I} \frac{j^{-2\alpha-2\beta}}{(a - ib)j^{-2\alpha} - (u + i\eta)} = \sum_{j \in I} \frac{j^{-2\beta}}{(a - ib) - (u + i\eta)(j_0^{2\alpha} + (j - j_0)j_0^{2\alpha-1}2\alpha)} + \xi_1.$$

The error terms $\xi_1$ are bounded by

$$|\xi_1| \lesssim j_0^{-2\beta} \sum_{j \in I} \frac{|u + i\eta|j_0^{2\alpha-2}|j - j_0|^2}{(b + j_0^{2\alpha}\eta)^2} \lesssim \frac{M^{3/2}u^{b/\alpha+1/\alpha}u^{-3-3/(2\alpha)}(\eta + ub)^3}{(b + j_0^{2\alpha}\eta)^2}$$

$$\lesssim M^{3/2}(\eta + ub)u^{\beta/\alpha-1-1/(2\alpha)}.$$

We then do a second replacement, freezing the $j^{-2\beta}$ in the numerator, and so we need to estimate

$$\xi_2 := \sum_{j \in I} \frac{j^{-2\beta} - j_0^{-2\beta}}{(a - ib) - (u + i\eta)(j_0^{2\alpha} + (j - j_0)j_0^{2\alpha-1}2\alpha)},$$

which we do simply by

$$|\xi_2| \lesssim j_0^{-2\beta-1} \max_{j \in I} \frac{|j - j_0|^2}{b + \eta/u} \lesssim M(\eta + ub)u^{\beta/\alpha-1-1/(2\alpha)}.$$

Thus with $M \asymp 1/\epsilon^2$ and using the second assumption of the lemma, we get

$$|\xi_1| + |\xi_2| \le \epsilon u^{\beta/\alpha-1/(2\alpha)},$$

where we have expressed

$$\sum_{j \in I} \frac{j^{-2\alpha-2\beta}}{(a - ib)j^{-2\alpha} - (u + i\eta)} = \sum_{j \in I} \frac{j_0^{-2\beta}}{z + w(j - j_0)} + \xi_1 + \xi_2 \quad \text{where} \quad \begin{cases} z = a - ib - (u + i\eta)j_0^{2\alpha}, \\ w = (u + i\eta)j_0^{2\alpha-1}2\alpha \end{cases}$$

The sum we can now evaluate using Lemma E.3. Note this makes $z$ nearly $-i(b + \eta/u)$ and $w$ nearly $-u^{1/(2\alpha)}(2\alpha)$, and hence $z/w$ is almost purely imaginary. Thus the error estimate in the Lemma applies and we have (using $|(\eta + bu)u^{-1-1/(2\alpha)}| \gtrsim \log(1/\epsilon)$ and $\eta < \epsilon u$)

$$\left| \sum_{j \in I} \frac{j^{-2\alpha-2\beta}}{(a - ib)j^{-2\alpha} - (u + i\eta)} - \frac{i\pi(u/a)^{\beta/\alpha-1/(2\alpha)}}{2\alpha} \right| \leq C\epsilon u^{\beta/\alpha-1/(2\alpha)}.$$

**The small $j$ regime, imaginary part.**   Recall the terms of small $j$, which is to say those with $j$ less than those in $I$, are denoted $S$. For these terms, we have $j^{-2\alpha}a - u \geq c\sqrt{M}(\eta + ub)$. For the real and imaginary parts of the sum we have

$$A_S + iB_S = \sum_{j \in S} \frac{j^{-2\alpha-2\beta}}{-u + j^{-2\alpha}a - i(\eta + j^{-2\alpha}b)} = \sum_{j \in S} \frac{j^{-2\alpha-2\beta}(-u + j^{-2\alpha}a + i(\eta + j^{-2\alpha}b))}{(-u + j^{-2\alpha}a)^2 + (\eta + j^{-2\alpha}b)^2}.$$

We shall focus on the imaginary part first. We introduce an approximation for this sum, coming from approximating the denominator by $j^{-4\alpha}a^2$. Thus we introduce

$$iB'_S \stackrel{\text{def}}{=} \frac{1}{a^2} \sum_{j \in S} j^{2\alpha-2\beta}(i(\eta + j^{-2\alpha}b)).$$

Let $c_\beta$ be as in the statement of the Proposition. Then

$$|iB'_S - c_\beta(ib/a^2)| \lesssim j_0^{1+2\alpha-2\beta}|\eta| + j_0^{1-2\beta}(b) \lesssim \epsilon u^{\beta/\alpha-1/2\alpha}.$$

We turn to estimating the difference of $B_S - iB'_S$. Using that $(j^{-2\alpha}a)^2 - (-u + j^{-2\alpha}a)^2 \leq 2u(j^{-2\alpha}a)$, we can estimate

$$|B_S - B'_S| \lesssim \sum_{j \in S} \frac{uj^{-2\beta}(\eta + j^{-2\alpha}b)}{(-u + j^{-2\alpha}a)^2}.$$

To estimate these differences, we break these sums into scales. We let $S_k$ to be those $j$ for which

$$S_k = \left\{ (\eta + ub)2^{k-1} \leq j^{-2\alpha}a - u \leq (\eta + ub)2^k \right\}.$$

Then we can estimate the number of terms in each of these $k$ by

$$|S_k| \leq C(\alpha)(\eta + ub)2^k(u + (\eta + ub)2^k)^{-1-1/(2\alpha)}.$$

For small $k$, i.e. those for which $(\eta + ub)2^k \leq u$, we can estimate $|S_k| \leq C(\alpha)(\eta + ub)2^k u^{-1-1/(2\alpha)}$. Call the small $k$ terms $S'$ and the remainder $S''$. Then for larger $S''$ terms,

$$|S_k| \leq C(\alpha)((\eta + ub)2^k)^{-1/(2\alpha)}.$$

For the difference of the imaginary parts on small $k$, we may bound $j^{-2\beta}$ as a multiple of $j_0^{-2\beta}$ and so we arrive at

$$|B_{S'} - B'_{S'}| \lesssim (u^{\beta/\alpha+1}) \sum_k \frac{|S_k|(\eta + ub)}{2^{2k}(\eta + ub)^2} \lesssim (u^{\beta/\alpha-1/(2\alpha)}) \sum_k \frac{1}{2^k} \lesssim \epsilon u^{\beta/\alpha-1/(2\alpha)}.$$

Then for the difference of the imaginary parts on large $k$

$$|B_{S''} - B'_{S''}| \lesssim \sum_k u\frac{|S_k|(\eta((\eta + ub)2^k)^{\beta/\alpha} + b((\eta + ub)2^k)^{\beta/\alpha+1})}{2^{2k}(\eta + ub)^2}.$$

This we further estimate

$$|B_{S''} - B'_{S''}| \lesssim u \sum_k \eta((\eta + ub)2^k)^{\beta/\alpha-2-1/(2\alpha)} + b((\eta + ub)2^k)^{\beta/\alpha-1-1/(2\alpha)}.$$

In the event that the exponents are non-negative, which can only occur when $2\beta > 1$, we may lose a factor which is boundable by the largest $k$ term (which is constant order) or by a logarithm in the case the exponent is $0$. If either exponent is negative, the expression is dominated by its smallest $k$ term, for which $(\eta + ub)2^k \asymp u$. In all we have

$$|B_S - B'_S| \lesssim \epsilon u^{\beta/\alpha-1/(2\alpha)} + (\eta + ub)u^{\beta/\alpha-1-1/(2\alpha)} + c_\beta(\eta + b)u\log(1/u).$$

**The large $j$ regime, imaginary part.** We break the sum into two parts $L'$ and $L''$, those with $j < 1.1j_0$ and those with $j \geq 1.1j_0$. For the terms in $L'$ we again break into scales, much like in the small $j$ regime. We let $L_k$ to be those $j$ for which

$$L_k = \left\{ (\eta + ub)2^{k-1} \leq u - j^{-2\alpha}a \leq (\eta + ub)2^k \right\}.$$

Then we can estimate the number of terms in each of these $k$ by

$$|L_k| \leq C(\alpha)(\eta + ub)2^k(u)^{-1-1/(2\alpha)}.$$

Then for the imaginary part

$$|B_{L'}| \lesssim \sum_k \frac{u^{\beta/\alpha+1}|L_k|(\eta + ub)}{2^{2k}(\eta + ub)^2}$$

$$\lesssim \sum_k \frac{u^{\beta/\alpha-1/(2\alpha)}}{2^k}.$$

This sum is always dominated by the smallest $k$, and so we have

$$|B_{L'}| \lesssim \epsilon u^{\beta/\alpha-1/(2\alpha)}.$$

As for larger $j$, we first remove a potentially divergent term, and so define

$$B'_{L''} = \sum_{j \in L''} \frac{j^{-2\alpha-2\beta}(\eta + ub)}{u^2}.$$

In the case that $\alpha + \beta < 1/2$, we have that (comparing to an integral and using monotonicity)

$$\left| B'_{L''} - \frac{(\eta + ub)V^{1-2\alpha-2\beta}}{u^2(1-2\alpha-2\beta)} \right| \lesssim (\eta + ub)\left( \frac{j_0^{-2\alpha-2\beta}}{u^2} + \frac{j_0^{1-2\alpha-2\beta}}{u^2} \right) \lesssim (\eta + ub)u^{-1+\beta/\alpha-1/(2\alpha)}$$

$$\lesssim \epsilon u^{\beta/\alpha-1/(2\alpha)}.$$

Otherwise,

$$|B'_{L''}| \lesssim (\eta + ub)u^{-1+\beta/\alpha-1/(2\alpha)} \lesssim \epsilon u^{\beta/\alpha-1/(2\alpha)}.$$

As for comparing the this divergence with the sum, we have

$$|B_{L''} - B'_{L''}| \lesssim \sum_{j \in L''} \frac{j^{-2\alpha-2\beta}(j^{-2\alpha}(\eta/u + b))}{(-u + j^{-2\alpha}a)^2} \lesssim \sum_{j \in L''} \frac{j^{-4\alpha-2\beta}((\eta + ub))}{u^3}.$$

Then if $2\alpha + \beta > 1/2$, this leaves

$$|B_{L''}| \lesssim (\eta + ub)u^{\beta/\alpha-1-1/(2\alpha)} \lesssim \epsilon u^{\beta/\alpha-1/(2\alpha)}$$

or in the case $2\alpha + \beta < 1/2$

$$|B_{L''}| \lesssim (\eta + ub)u^{-3}v^{1-4\alpha-2\beta} \lesssim (\eta + ub)\left(v^{-2\alpha}/u\right)u^{-2}v^{1-2\alpha-2\beta}.$$

**The real part.** For the real part, we shall prove a comparison with

$$\mathcal{A} = \mathrm{Re}\left( \sum_j \frac{j^{-2\alpha-2\beta}}{j^{-2\alpha} - u - i\eta} \right),$$

which we note is a special case of $A$ with $a - ib = 1$. The arguments are now very similar in all regimes to the imaginary parts, and so we just give a summary of the arguments.

The main difference is for $j \approx j_0$. Note that using the previous bounds on the transition window, we may discard an interval of $|j - j_0| \leq \sqrt{M}\eta j_0^{1+2\alpha}$ from $\mathcal{A}$ and incur an error of only $\epsilon u^{\beta/\alpha-1/(2\alpha)}$. On a larger interval, $J$, given by those $j$ with

$$\eta j_0^{1+2\alpha} \leq |j - j_0| \leq \epsilon j_0,$$

by pairing $j_0 + r$ with $j_0 - r$, we can bound

$$|\mathcal{A}_J| + |A_J| \lesssim \epsilon j_0^{1-2\beta} \lesssim \epsilon u^{\beta/\alpha-1/(2\alpha)}.$$

Moreover, the difference we can bound by

$$|\mathcal{A}_J - A_J| \lesssim |1-a| \sum_{j \in J} \frac{j^{-4\alpha - 2\beta}}{|(j^{-2\alpha}a - u)(j^{-2\alpha} - u)|} \lesssim \frac{|1-a|}{\epsilon} u^{\beta/\alpha - 1/(2\alpha)}.$$

For small $j$, where we redefine $S$ as those $j$ smaller than those in $J$, we further divide to hose $j$ with $|j - j_0| \le j_0/2$ and those $S'$ which are further from $j_0$.

$$|\mathcal{A}_S - A_S| \lesssim \frac{|1-a|}{\epsilon} j_0^{1-2\beta} + |1-a| \sum_{j \in S'} j^{-2\beta}.$$

Hence we arrive at

$$|\mathcal{A}_S - A_S| \lesssim \epsilon u^{\beta/\alpha - 1/(2\alpha)} + c_\beta |1-a|.$$

For the large $j$ terms, we redefine $L$ as those $j$ larger than those in $J$. Again dividing to those with $|j - j_0| \le j_0/2$ and otherwise, we arrive at

$$|\mathcal{A}_L - A_L| \lesssim \frac{|1-a|}{\epsilon} j_0^{1-2\beta} + |1-a| \sum_{j \in L'} \frac{j^{-4\beta - 2\beta}}{u^2}.$$

This, as in the large terms for the imaginary part, leads to

$$|\mathcal{A}_L - A_L| \lesssim \frac{|1-a|}{\epsilon} j_0^{1-2\beta} + |1-a| \epsilon u^{\beta/\alpha - 1/(2\alpha)} + |1-a| o_{\alpha+\beta} \epsilon \frac{v^{1-2\alpha - 2\beta}}{u}.$$

Finally, we observe that $\mathcal{A}$ satisfies an estimate of the form

$$|\mathcal{A}| \lesssim u^{\beta/\alpha - 1/(2\alpha)} + c_\beta + o_{\alpha+\beta} \frac{v^{1-2\alpha - 2\beta}}{u},$$

which arise from the transitionary region, the small $j$ region and the large $j$ region.

$\square$

**Proposition E.5.** *Assume $\alpha \ne \frac{1}{4}$ and $\alpha \ne \frac{1}{2}$. With $z = u + i\eta(u)$, with*

$$\eta = (\log(1/\epsilon)/c) \max \left\{ u^{1+1/(2\alpha)}, \frac{\pi}{2\alpha} \frac{u^{1-1/(2\alpha)}}{d} \right\},$$

*there is a $c > 0$ and an $\epsilon_0$ so that for all $\epsilon \in (0, \epsilon_0)$ there is a $c_\epsilon > 0$ so for all $u \in [d^{-2\alpha}/c_\epsilon, c_\epsilon]$ (with $\mathcal{A}$ as in Proposition E.4)*

$$\left| m(z(u)) - 1 + d^{-1} \mathcal{A}(u + i\eta) + i \frac{\pi}{2\alpha} u^{-1/(2\alpha)} d^{-1} \right| \le C(\alpha) \epsilon u^{-1/(2\alpha)} d^{-1}.$$

*Proof.* We claim that $m$ is approximately equal to

$$m_0 \stackrel{\text{def}}{=} 1 - \frac{\mathcal{A}(u + i\eta)}{d} - i \frac{\pi u^{-1/(2\alpha)}}{2\alpha d},$$

where $m$ is the solution of

$$F(m(z), z) := m(z) + \frac{v}{d} \mathbb{E}\left( \frac{Xm(z)}{Xm(z) - z} \right) = 1,$$

with $\text{Im}\, m < 0$. Hence the result boils down to checking:

$$|F(m_0, z) - 1| \le C\epsilon \frac{u^{-1/(2\alpha)}}{d}$$

and secondly that

$$|1 - \partial_m(F)| \le \tfrac{1}{2}$$

in a neighborhood of $m_0$, using Lemma E.2.

For showing that $|F(m_0, z) - 1|$ we first observe that on the contour selected, if $\alpha < \frac{1}{2}$ and $\epsilon_0, c_\epsilon$ is chosen sufficiently small

$$\frac{v^{1-2\alpha}}{u^2 d}(\eta + ub) \leq \epsilon \frac{\pi u^{-1/(2\alpha)}}{d}.$$

Moreover the claimed estimates on $1 - F$ now follow directly from Proposition E.4.

For the stability, we have that

$$1 - \partial_m F = \frac{1}{d} \sum_{j=1}^{v} \frac{j^{-2\alpha} z}{(j^{-2\alpha} m - z)^2}.$$

Taking modulus, we have

$$|1 - \partial_m F| \leq \frac{1}{d} \sum_{j=1}^{v} \frac{j^{-2\alpha}(u + \eta)}{(j^{-2\alpha} a - u)^2 + (j^{-2\alpha} b + \eta)^2} =: X$$

Now we break the estimation of the sum into regions of $j$, as in the proof of Proposition E.4. We let $j_0$ be the integer which minimizes $(j_0^{-2\alpha} a - u)^2$. We define $S, L, I, J$ to be the sets where

$$j < \delta j_0, \quad j > j_0/\delta, \quad (j^{-2\alpha} a - u)^2 \leq (ub + \eta)^2,$$

and the rest in $J$, we let $X_A$ be the restriction of the sum $X$ to the set of indices $A$. For the terms in $S$,

$$X_S \lesssim \frac{1}{d} \sum_{j \in S} \frac{j^{2\alpha}(u + \eta)}{a^2(1 - O(\delta))} \lesssim \frac{u^{-1/2\alpha}}{d} \delta^{1+2\alpha},$$

with the final sum holding for all $\delta > 0$ sufficiently small. For the terms in $L$,

$$X_L \lesssim \frac{1}{d} \sum_{j \in L} \frac{j^{-2\alpha}(u + \eta)}{u^2(1 - O(\delta))} \lesssim \frac{u^{-1/2\alpha}}{d} \delta^{1+2\alpha} + o_\alpha \frac{v^{1-2\alpha}}{d} \frac{u}{u^2 + \eta^2},$$

where $o_\alpha$ is the indicator of $2\alpha < 1$. For the terms in $I$ we have

$$X_I \lesssim \frac{1}{d} \sum_{j \in I} \frac{j^{-2\alpha}(u + \eta)}{(ub + \eta)^2} \lesssim \frac{1}{d} \frac{u^{-1/(2\alpha)}(u + \eta)}{(ub + \eta)},$$

where we have used that the number of terms in this regions is on order of $j_0^{2\alpha+1}(ub + \eta)$. Now taking $\eta$ a sufficiently large multiple of $u\beta$, we conclude that the terms in $X_I \leq \frac{1}{8}$. For the terms in $J$

$$X_J \lesssim \frac{1}{d} \sum_{j \in J} \frac{j^{-2\alpha}(u + \eta)}{(j^{-2\alpha} a - u)^2} \lesssim C(\delta) \frac{1}{d} \sum_r \frac{j_0^1(u + \eta)}{j_0^{-2\alpha-1}(r)^2};$$

here the range of $r$ is such that at its smallest value $j_0^{-2\alpha-1}(r) \asymp (ub + \eta)$, and so we arrive at

$$X_J \leq C(\delta) \frac{1}{d} \frac{u^{-1/(2\alpha)}(u + \eta)}{(ub + \eta)}.$$

Hence picking $\delta$ sufficiently small that $X_S, X_L$ are both less than $\frac{1}{8}$, and subsequently increasing the lower bound on $\eta/(ub)$ sufficiently far, we conclude that all four components can be made less than $\frac{1}{8}$ and hence that

$$|1 - \partial_m F| \leq \frac{1}{2}.$$

$\square$

## E.4 The large $z$ region

**Proposition E.6.** *For any compact set $U \subset \mathbb{C}$ of distance at least $\delta > 0$ from $[0, 1]$ and any $\alpha \neq 1/2$ there is a $C(\alpha)$ such that*

$$|m(z) - 1| \leq \frac{C(\alpha)}{\delta \min\{d, d^{2\alpha}\}}$$

*and such that*

$$\left| m(z) - 1 + \frac{1}{d} \sum_{j=1}^{v} \frac{j^{-2\alpha}}{j^{-2\alpha} - z} \right| \le \frac{C(\alpha)}{\delta \min\{d^2, d^{4\alpha}\}}$$

*Furthermore, on the same set*

$$\left| \sum_{j=1}^{v} \frac{j^{-2\alpha-2\beta}}{j^{-2\alpha}m - (u + i\eta)} - \sum_{j=1}^{v} \frac{j^{-2\alpha-2\beta}}{j^{-2\alpha} - (u + i\eta)} \right| \le \frac{C(\alpha)}{\delta \min\{d, d^{2\alpha}\}}.$$

*Proof.* We apply Proposition E.2 part 2. We have

$$\frac{v}{d} \mathbb{E} X = \frac{1}{d} \sum_{j=1}^{v} j^{-2\alpha} \lesssim \frac{1}{\min\{d, d^{2\alpha}\}},$$

and the result follows directly from Proposition E.2.

We turn to evaluating the sum

$$S(m, u + i\eta) \stackrel{\text{def}}{=} \sum_{j=1}^{v} \frac{j^{-2\alpha-2\beta}}{j^{-2\alpha}m - (u + i\eta)}.$$

Then taking the partial derivative in $m$,

$$\partial_m S(m, u + i\eta) = \sum_{j=1}^{v} \frac{j^{-4\alpha-2\beta}}{(j^{-2\alpha}m - (u + i\eta))^2},$$

which is uniformly bounded on $U$ and on the set $m$ so $|m - 1| < \delta/2$. It follows that on $U$

$$|S(m, u + i\eta) - S(1, u + i\eta)| \lesssim \frac{1}{\min\{d, d^{2\alpha}\}}.$$

For the second part, we start by observing that we can estimate

$$\left| \frac{v}{d} \mathbb{E}\left( \frac{X}{X - z} \right) \right| = \frac{1}{d} \left| \sum_{j=1}^{v} \left( \frac{j^{-2\alpha}}{j^{-2\alpha} - u - i\eta} \right) \right| \lesssim \frac{1}{\delta \min\{d, d^{2\alpha}\}}.$$

We further have

$$\left| \frac{v}{d} \mathbb{E}\left( \frac{X^2}{(Xm - z)^2} \right) \right| \lesssim \frac{1}{d} \left| \sum_{j=1}^{v} j^{-4\alpha}/\delta^2 \right| \lesssim \frac{1}{\delta^2 \min\{d, d^{4\alpha}\}}.$$

Hence, combining all these errors we conclude the claim. $\qquad\square$

# F    Approximation of the forcing function

We now apply the technical estimate to find good approximations for the function $\mathcal{F}$. Recall

$$\mathcal{F}(r) \stackrel{\text{def}}{=} \frac{-1}{2\pi i} \oint_{\Gamma + \Gamma_0} \langle \mathcal{R}(z), (D^{1/2}b)^{\otimes 2} \rangle (1 - 2\gamma Bz + \gamma^2 B(B+1)z^2)^r \, dz,$$

where $\quad \mathcal{R}(z) = \text{Diag}\left( \frac{j^{-2\alpha}}{-z + j^{-2\alpha}m(z)} : 1 \le j \le v \right) \quad$ and $\quad m(z) = \dfrac{1}{1 + \frac{1}{d} \sum_{j=1}^{v} \frac{j^{-2\alpha}}{j^{-2\alpha}m(z) - z}}.$

We decompose the forcing function into a sum of three functions

$$\mathcal{F}(r) = \mathcal{F}_{pp}(r) + \mathcal{F}_{ac}(r) + \mathcal{F}_0(r) + \text{errors},$$

which will be introduced in the course of the approximation.

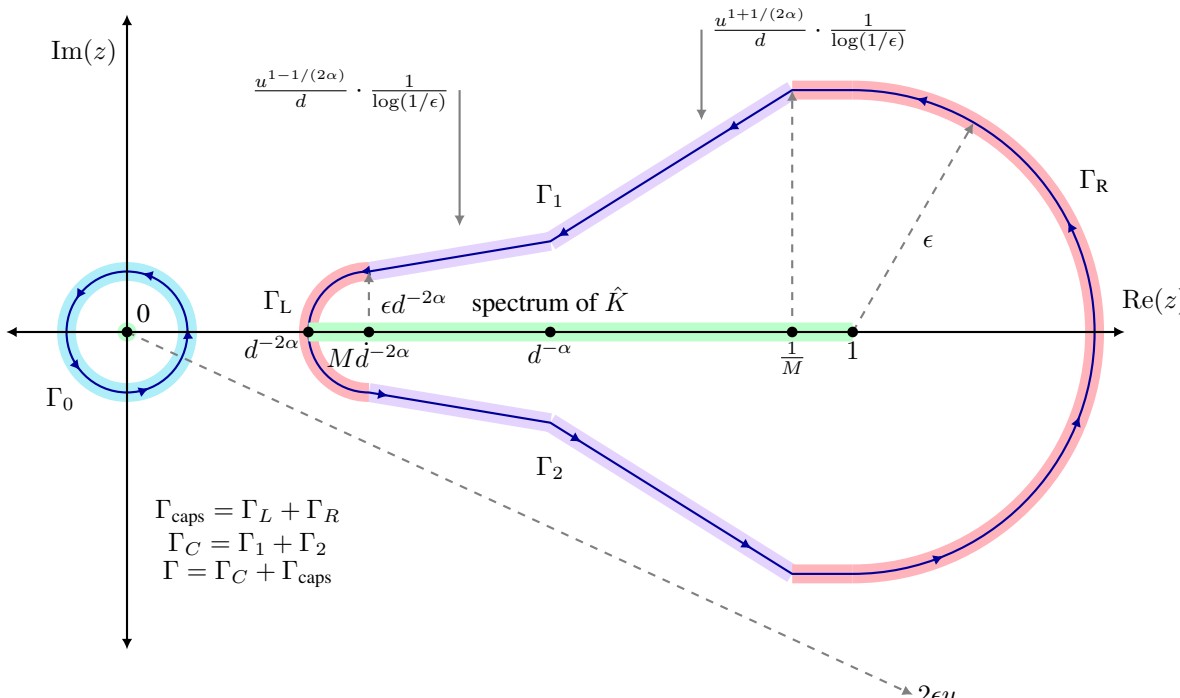

Figure 7: **Contour of** $\Gamma + \Gamma_0$. This is used to estimate the $m$ and derive expressions for the forcing function and kernel function. The important part of the contour is $\Gamma_0$, which contains the point mass at 0 (blue) and $\Gamma_C$ (purple) which contains the bulk of the spectrum of deterministic equivalent of $\hat{K}$. There is a left spectral gap which occurs at $d^{-2\alpha}$. Moreover we have a change of behavior at $d^{-\alpha}$ in the contour to account for the change of behavior from pure point to absolutely continuous bulk part of the spectrum.

The contour we select will come in three parts. The contour $\Gamma_0$ is an arbitrarily small contour enclosing 0. The contour $\Gamma$ will be in three parts which is symmetric under the reflection $z \mapsto -z$. The main part will be $\Gamma_C$ parameterized by $z = u + i\eta(u)$ with $\eta(u)$ as in Proposition E.5 for $u \in [u_0, u_1]$ where $u_0 = u_0(d) = Cd^{-2\alpha}$ for some large $C > 0$ and $u_1$ is a small positive constant. This is connected by two curves, one which is a smooth curve $\Gamma_L$ which is on scale $d^{-2\alpha}$ and which is reflection symmetric, connects $u_0 + i\eta(u_0)$ to its conjugates and crosses the imaginary axis on $[0, cd^{-2\alpha}]$ (with $c$ as in Proposition E.3). The other $\Gamma_R$ connects $u_1 + i\eta(u_1)$ to its conjugate by a smooth curve which avoids an $\epsilon$ neighborhood of $[0, 1]$.

For $\Gamma_0$, using Proposition E.3, we have

$$\mathcal{F}_0(r) \overset{\text{def}}{=} \frac{-1}{2\pi i} \oint_{\Gamma_0} \langle \mathcal{R}(z), (D^{1/2}b)^{\otimes 2} \rangle (1 - 2\gamma Bz + \gamma^2 B(B+1)z^2)^r \, dz. \tag{84}$$

This can be evaluated explicitly in terms of a residue at 0.

**Proposition F.1.** *The function $\mathcal{F}_0(r)$ is constant and*

$$\left| \mathcal{F}_0(0) - \sum_{j=1}^{v} \frac{j^{-2\alpha-2\beta}}{1 + j^{-2\alpha}d^{2\alpha}\kappa(v/d)} \right| \leq Cd^{-2\alpha+(2\beta-1)_+ - 1}.$$

*Proof.* From Proposition E.3, we can apply the residue formula. Evaluating the residue and bounding the sum produces the statement. □

The contours $\Gamma_R$ and $\Gamma_L$ both contribute error terms. Define the sum of the two as

$$\mathcal{F}_{caps} \overset{\text{def}}{=} \frac{-1}{2\pi i} \oint_{\Gamma_R + \Gamma_L} \langle \mathcal{R}(z), (D^{1/2}b)^{\otimes 2} \rangle (1 - 2\gamma Bz + \gamma^2 B(B+1)z^2)^r \, dz.$$

**Proposition F.2.** *There are positive functions $f(r)$ and $g(r)$ satisfying $f(r) \leq C \exp(-c\gamma Brd^{-2\alpha})$ and $g(r) \leq C \exp(-c\gamma Br)$ so that*

$$\left| \mathcal{F}_{caps}(r) \right| \leq Cf(r)d^{-2\alpha+(1-2\beta)_+} + Cg(r).$$

*Furthermore, for any $M > 1$ we can choose $u_0 = Td^{-2\alpha}$ and $u_1 = 1/T$ with $T$ sufficiently large that $\mathcal{F}_{caps}(r)$ satisfies for $\gamma Br \leq M$ and $\gamma Br \geq Md^{2\alpha}$ and some other $C > 0$*

$$\mathcal{F}_{caps}(r) \geq f(Cr)d^{-2\alpha+(1-2\beta)_+}/C + g(r)/C.$$

Hence this will appear as essentially constant on the loss curves. When combined with $\mathcal{F}_0$ we have that $\mathcal{F}_0(r) + \mathcal{F}_{caps}(r)$ is bounded above and below by constants times $d^{-2\alpha+(2\beta-1)_+-1}$.

*Proof.* Both the contributions from $\Gamma_L$ and $\Gamma_R$ give exponentially decaying errors, albeit at much different scales and lead to the $f$ and $g$ terms respectively. For the $\Gamma_R$ terms, we simply bound, using Proposition E.6,

$$|m(z) - 1| \lesssim d^{-\min\{2\alpha,1\}}.$$

On the $\Gamma_R$ contour, having picked the contour sufficiently close to $[0,1]$ (independent of $v, d$), we have for some $\delta > 0$

$$\left| (1 - 2\gamma Bz + \gamma^2 B(B+1)z^2)^r \right| \leq e^{-2\gamma B \operatorname{Re} zr(1-\delta)}.$$

Then we have

$$\left| \frac{-1}{2\pi i} \oint_{\Gamma_R} \left( \langle \mathcal{R}(z), (D^{1/2}b)^{\otimes 2} \rangle - \sum_{j=1}^{v} \frac{j^{-2\alpha-2\beta}}{j^{-2\alpha} - z} \right) (1 - 2\gamma Bz + \gamma^2 B(B+1)z^2)^r \, dz \right|$$

$$\lesssim d^{-\min\{2\alpha,1\}} \times \left| \oint_{\Gamma_R} e^{-2\gamma B \operatorname{Re} zr(1-\delta)} |dz| \right|.$$

Hence this decays exponentially.

By construction of the $\Gamma_R$ contour, we can close the contour with an additional (nearly vertical) segment $\Gamma_V$ with real part $u$ and height $\epsilon u$. Moreover this can be chosen to evenly divide two poles $\{j^{-2\alpha}\}$, by adding small horizontal segments. Then we can estimate on $\Gamma_V$ (essentially by Proposition E.4, with an extension for very small imaginary part when we split two poles)

$$\left| \sum_{j=1}^{v} \frac{j^{-2\alpha-2\beta}}{j^{-2\alpha} - z} \right| \lesssim u_1^{\beta/\alpha-1/(2\alpha)}.$$

Then integrating over $\Gamma_V$, we get

$$\left| \frac{-1}{2\pi i} \oint_{\Gamma_V} \left( \sum_{j=1}^{v} \frac{j^{-2\alpha-2\beta}}{j^{-2\alpha} - z} \right) (1 - 2\gamma Bz + \gamma^2 B(B+1)z^2)^r \, dz \right| \lesssim \epsilon u_1^{1+\beta/\alpha-1/(2\alpha)} e^{-2\gamma Bu_1 r(1-\delta)}.$$

Having enclosed the poles, we can apply the residue formula, and we have

$$\frac{-1}{2\pi i} \oint_{\Gamma_V + \Gamma_R} \left( \sum_{j=1}^{v} \frac{j^{-2\alpha-2\beta}}{j^{-2\alpha} - z} \right) (1 - 2\gamma Bz + \gamma^2 B(B+1)z^2)^r \, dz$$

$$= \sum_{j=1}^{j_0} j^{-2\alpha-2\beta}(1 - 2\gamma Bj^{-2\alpha} + \gamma^2 B(B+1)j^{-4\alpha})^r,$$

for some $j_0$ with $j_0 \asymp u_1^{-1/(2\alpha)}$. Hence both contributions of $\Gamma_V$ and $\Gamma_R$ decay like $g(r)$ for an appropriate choice of $\delta, C$.

For $\Gamma_L$, we use similar arguments. We use Proposition E.3 to replace the summation by a $d$-independent quantity, which also requires rescaling the contour by $d^{-2\alpha}$. Then we have

$$\left| \frac{-1}{2\pi i} \oint_{\Gamma_L} \left( \langle \mathcal{R}(z), (D^{1/2}b)^{\otimes 2} \rangle - d^{1-2\beta} \int_0^a \frac{x^{-2\beta}\,\mathrm{d}x}{f(zd^{2\alpha}) - zd^{2\alpha}x^{2\alpha}} \right) (1 - 2\gamma Bz + \gamma^2 B(B+1)z^2)^r \,\mathrm{d}z \right|$$
$$\lesssim d^{-2\alpha} \left| \oint_{\Gamma_L d^{2\alpha}} e^{-2\gamma Bcd^{-2\alpha}r} \,|dz| \right|.$$

Hence we are left with a dominant contribution of

$$d^{1-2\beta-2\alpha} \frac{-1}{2\pi i} \oint_{\Gamma_L d^{2\alpha}} \left( \int_0^a \frac{x^{-2\beta}\,\mathrm{d}x}{f(z) - zx^{2\alpha}} \right) \exp(-2\gamma Brzd^{-2\alpha})\,\mathrm{d}z$$

In the case that $2\beta > 1$ we instead are left with

$$c_\beta d^{-2\alpha} \frac{-1}{2\pi i} \oint_{\Gamma_L d^{2\alpha}} \left( \frac{1}{f(z)} \right) \exp(-2\gamma Brzd^{-2\alpha})\,\mathrm{d}z.$$

As the spectral support of $f$ has a left edge, these decay exponentially. In either case, we can then deform the contour to run twice along the real axis and then vertically to the ends of the $\Gamma_L$ contour. The component along the vertical portion can be estimated by

$$O(d^{(1-2\beta)_+}(u_0^{1-1/(2\alpha)}/d)) \exp(-(2-\delta)\gamma Bru_0)$$

(and using the boundedness of $f, 1/f$). This can be made to decay faster than the contribution from $f$.

$\square$

Finally, the dominant contributions arise from the contour $\Gamma_C$. We define:

$$\begin{aligned}
\mathcal{F}_{pp}(r) &\stackrel{\text{def}}{=} \frac{1}{2\alpha} \int_0^1 u^{(2\beta-1)/(2\alpha)} \exp(-2\gamma Bru)\,\mathrm{d}u, \\
\mathcal{F}_{ac}(r) &\stackrel{\text{def}}{=} \frac{c_\beta}{2\alpha} \int_{d^{-2\alpha}}^1 u^{-1/(2\alpha)} d^{-1} \exp(-2\gamma Bru)\,\mathrm{d}u,
\end{aligned}$$
(85)

where $c_\beta$ is as in Proposition E.5. Then this gives us the principal contribution to the limit:

**Proposition F.3.** *Set for $r \geq 0$*

$$\mathcal{F}_C(r) \stackrel{\text{def}}{=} \frac{-1}{2\pi i} \oint_{\Gamma_C} \langle \mathcal{R}(z), (D^{1/2}b)^{\otimes 2} \rangle (1 - 2\gamma Bz + \gamma^2 B(B+1)z^2)^r \,\mathrm{d}z.$$

*Then $\mathcal{F}_C(r)$ is real-valued and satisfies for some constant $C$ independent of $u_1, u_0, \alpha, \beta$*

$$|\mathcal{F}_C(r)| \leq C(\mathcal{F}_{pp}(r) + \mathcal{F}_{ac}(r)).$$

*Moreover, there is an $M = M(u_0, u_1) > 0$ and a positive bounded function $C(r)$ so that if $\gamma Br \in [M, d^{2\alpha}/M]$ then*

$$\frac{1}{C(r)}(\mathcal{F}_{pp}(r) + \mathcal{F}_{ac}(r)) \leq \mathcal{F}_C(r) \leq C(r)(\mathcal{F}_{pp}(r) + \mathcal{F}_{ac}(r))$$

*Furthermore, for any $\epsilon > 0$ there is a $M(\epsilon, u_0, u_1)$ large enough that $C(r) \leq 1 + \epsilon$ for $\gamma Br \in [M, d^{2\alpha}/M]$.*

*Proof.* These follow in a similar way to the earlier Propositions, and so we do not enter the details. Instead, we give a brief overview, using the estimates given in Proposition E.5 and Proposition E.4.

Along $\mathcal{F}_C$, we can approximate $m$ uniformly by

$$\left| m(z(u)) - \left( 1 - (c(u) + i)\frac{\pi}{2\alpha}u^{-1/(2\alpha)}d^{-1} \right) \right| \leq \epsilon u^{-1/(2\alpha)}d^{-1},$$

where $c$ is real-valued and bounded and $M = M(\epsilon)$. Hence using Proposition E.4,

$$\sum_{j=1}^{v} \frac{j^{-2\alpha-2\beta}}{-u - i\eta + j^{-2\alpha}m(z(u))} = (1 + O(\epsilon))\mathcal{A}(u) + i(1 + O(\epsilon))\frac{\pi u^{\beta/\alpha - 1/(2\alpha)}}{2\alpha}$$

$$+ i(1 + O(\epsilon))c_\beta \frac{\pi}{2\alpha} u^{-1/(2\alpha)} d^{-1}$$

for real valued $\mathcal{A}$ and $c_\beta = \sum_{j=1}^{\infty} j^{-2\beta}$ if $\beta > 1/2$ or 0 otherwise as in Proposition E.4. Integrating each of these imaginary terms over $\Gamma_C$ produces $\mathcal{F}_{pp}$ and $\mathcal{F}_{ac}$ respectively. The real part is negligible, as the contour is close to the real axis (in particular the imaginary part of the contour is smaller than the real part by a factor of $\epsilon$). $\qquad\square$

Combining all of these propositions, we have the following conclusion

**Corollary F.1.** *For any $\alpha, \beta$ with $\alpha, \beta \neq 1/2$ and $\alpha + \beta > \frac{1}{2}$ there is a function $C(r)$ bounded above for all $r$ so that*

$$\frac{1}{C(r)}(\mathcal{F}_{pp}(r) + \mathcal{F}_{ac}(r) + \mathcal{F}_0(r)) \leq \mathcal{F}(r) \leq C(r)(\mathcal{F}_{pp}(r) + \mathcal{F}_{ac}(r) + \mathcal{F}_0(r)),$$

*Moreover, for any $\epsilon > 0$ there is a $M(\epsilon)$ large enough that $C(r) \leq 1 + \epsilon$ for $\gamma Br \in [M, d^{2\alpha}/M]$ and for $\gamma Br > Md^{2\alpha}$.*

*Proof.* This follows directly from Proposition F.3, F.2 and F.1, and needs that the $\mathcal{F}$ curve is monotone to fill the gaps on which the approximations are made. ( There are potentially two windows on which the various approximations do not overlap: when $\gamma Br$ is a large constant and when it is on order $d^{2\alpha}$) $\qquad\square$

# G  Estimation of Kernel function

We can now give the approximation of the kernel function, which is represented by

$$\mathcal{K}(r) \stackrel{\text{def}}{=} \frac{-\gamma^2 B}{2\pi i} \oint_{\Gamma + \Gamma_0} z^2 \text{Tr}(\mathcal{R}(z))(1 - 2\gamma Bz + \gamma^2 B(B+1)z^2)^r \, \mathrm{d}z,$$

with the same contours that were used for the forcing function.

Using Lemma E.1, we therefore can represent the kernel function as

$$\mathcal{K}(r) \stackrel{\text{def}}{=} \frac{-\gamma^2 B}{2\pi i} \oint_{\Gamma + \Gamma_0} z(-v + (1 - m(z))d)(1 - 2\gamma Bz + \gamma^2 B(B+1)z^2)^r \, \mathrm{d}z.$$

By the residue theorem, the contribution from $\Gamma_0$ disappears, as does the $Vz$ term. Hence we are left with the representations

$$\mathcal{K}(r) \stackrel{\text{def}}{=} \frac{-\gamma^2 Bd}{2\pi i} \oint_{\Gamma} z(1 - m(z))(1 - 2\gamma Bz + \gamma^2 B(B+1)z^2)^r \, \mathrm{d}z.$$

When $\alpha > \frac{1}{4}$, the dominant contribution comes once more from the contour $\Gamma_C$ for which we get

$$\mathcal{K}_{pp}(r) \stackrel{\text{def}}{=} \frac{\gamma^2 B}{2\alpha} \int_0^1 u^{1 - 1/(2\alpha)} \exp(-2\gamma Bur) \, \mathrm{d}u. \tag{86}$$

It now follows swiftly from the estimates on $m$:

**Proposition G.1.** *Suppose $\alpha > \frac{1}{4}$. There is a positive function $C(r)$ so that*

$$\frac{1}{C(r)}\mathcal{K}_{pp}(r) \leq \mathcal{K}(r) \leq C(r)\mathcal{K}_{pp}(r),$$

*and $C(r)$ is bounded independent of $d$ by a function of $M$ for all $r\gamma B < d^{2\alpha}M$. Moreover for any $\epsilon > 0$ there is an $M$ sufficiently large so that for $r\gamma B \in [M, d^{2\alpha}/M]$, $C(r) < 1 + \epsilon$.*

*Proof.* By reflection symmetry, the real part of contour integral for $\mathcal{K}(r)$ vanishes. Also, for a given contour $\Gamma_A$, we will define

$$\mathcal{K}_A(r) = \frac{-\gamma^2 B}{\pi} \operatorname{Im} \oint_{\Gamma_A} z(1 - m(z))d(1 - 2\gamma Bz + \gamma^2 B(B+1)z^2)^r \, dz,$$

and we will estimate each piece of $\Gamma = \Gamma_R + \Gamma_C + \Gamma_L$ separately.

We begin with the contributions from $\Gamma_R$. Using Proposition E.6, we have that on $\Gamma_R$

$$\left| \frac{-\gamma^2 B}{2\pi i} \oint_{\Gamma_R} z \left( (1 - m(z))d - \sum_{j=1}^{V} \frac{j^{-2\alpha}}{j^{-2\alpha} - z} \right) (1 - 2\gamma Bz + \gamma^2 B(B+1)z^2)^r \, dz \right|$$
$$\lesssim d^{-\min\{1, 4\alpha - 1\}} \gamma^2 B \exp(-\delta \gamma Br).$$

Following the same steps as in the proof of Proposition F.2, we can extend the $\Gamma_R$ contour by a straight line $\Gamma_V$ to enclose some residues, which leads to

$$\frac{-\gamma^2 B}{2\pi i} \oint_{\Gamma_R + \Gamma_V} z \left( \sum_{j=1}^{v} \frac{j^{-2\alpha}}{j^{-2\alpha} - z} \right) (1 - 2\gamma Bz + \gamma^2 B(B+1)z^2)^r \, dz$$
$$= \gamma^2 B \sum_{j=1}^{j_0} j^{-4\alpha}(1 - 2\gamma Bj^{-2\alpha} + \gamma^2 B(B+1)j^{-4\alpha})^r,$$

with $j_0 \asymp u_1^{-1/(2\alpha)}$. Moreover, the contribution of the $\Gamma_V$ contour can be estimated (for some $\delta > 0$ which can be made small by increasing $u_1$) by

$$O(u_1^{2 - 1/(2\alpha)} \exp(-2\gamma Bu_1 r(1 - \delta))).$$

Meanwhile making a Riemann sum approximation (and changing variables by $j^{-2\alpha} = u$)

$$\sum_{j=1}^{j_0} j^{-4\alpha}(1 - 2\gamma Bj^{-2\alpha} + \gamma^2 B(B+1)j^{-4\alpha})^r \asymp \frac{1}{2\alpha} \int_{u_1}^{1} u^{1 - 1/(2\alpha)} \exp(-2\gamma Bru) \, du$$

for $2\gamma Br < \frac{1}{u_1}$. Hence by taking $u_1$ sufficiently small, we conclude that for $2\gamma Br < \frac{1}{u_1}$,

$$\mathcal{K}_R(r) \asymp \frac{\gamma^2 B}{2\alpha} \int_{u_1}^{1} u^{1 - 1/(2\alpha)} \exp(-2\gamma Bru) \, du,$$

and also that $|\mathcal{K}_R(r)| \lesssim e^{-2\gamma Bru_1(1-\delta)}$ for larger $(2\gamma Br)$.

The contributions from $\Gamma_C$ give, in a similar way for $2\gamma Br > \frac{1}{u_1}$ and $2\gamma Br < \frac{1}{u_1}$

$$\mathcal{K}_C(r) \asymp \gamma^2 B \int_{u_0}^{u_1} u^{1 - 1/(2\alpha)} \exp(-2\gamma Bru) \, du$$

Moreover for any $\epsilon > 0$ there is an $M > 0$ sufficiently large that when $(2\gamma Br)$ is in $[M, d^{2\alpha}/M]$,

$$1 - \epsilon < \frac{\gamma^2 B}{2\alpha \mathcal{K}_C(r)} \int_{u_0}^{u_1} u^{1 - 1/(2\alpha)} \exp(-2\gamma Bru) \, du < 1 + \epsilon,$$

by first choosing $\epsilon > 0$, then choosing the contour as in Proposition E.5 sufficiently far, and then possibly shrinking $[u_0, u_1]$. For larger $r$, it further satisfies an estimate that for some $\delta > 0$, which can be made smaller by increasing $u_0$, $\mathcal{K}_L(r) \lesssim e^{-2\gamma Bru_0(1-\delta)}$.

Finally, the contributions from $\Gamma_L$, we have after changing variables

$$\mathcal{K}_L(r) = \frac{-\gamma^2 B}{\pi} d^{1 - 4\alpha} \operatorname{Im} \oint_{\Gamma_L d^{2\alpha}} z(1 - m(zd^{2\alpha}))(1 - 2\gamma Bzd^{-2\alpha} + \gamma^2 B(B+1)z^2 d^{-4\alpha})^r \, dz.$$

This can be compared to the same expression with $m(zd^{2\alpha}) \to f(z)$ and replacing $(1 - 2\gamma Bzd^{-2\alpha} + \gamma^2 B(B+1)z^2 d^{-4\alpha})^r \to \exp(-2\gamma Br(\operatorname{Re} z)d^{-2\alpha})$. This gives for $2\gamma Br \lesssim d^{2\alpha}$

$$\mathcal{K}_L(r) \asymp \gamma^2 Bd^{1 - 4\alpha} \int_0^{u_0 d^{2\alpha}} u\mathfrak{f}(u) \exp(-2\gamma Brud^{-2\alpha}) \, du,$$

where $\mathfrak{f}(u) = \frac{-1}{\pi} \lim_{\epsilon \to 0} f(u + i\epsilon)$ is the spectral density corresponding to $f$. Hence it follows that for $2\gamma Br \lesssim d^{2\alpha}$,

$$\mathcal{K}_L(r) \asymp \gamma^2 B \int_{d^{-2\alpha}}^{u_0} u^{1-1/(2\alpha)} \exp(-2\gamma Bru) \, du.$$

$\square$

**Remark G.1.** *In contrast, when $\alpha < \frac{1}{4}$, the dominant contribution to $\mathcal{K}$ is from $\mathcal{K}_L$, so that*

$$\mathcal{K}(r) \asymp \gamma^2 Bd^{1-4\alpha} \int_0^\infty u\mathfrak{f}(u) \exp(-2\gamma Brud^{-2\alpha}) \, du,$$

*and moreover the density $\mathfrak{f}(u) \lesssim u^{-1/(2\alpha)}$ so that the integral is convergent (which is implicit in Proposition E.5)*

We conclude with noting that for ther norm of $\mathcal{K}$ we can directly evaluate it using a contour integral. Summing the contour integral expression

$$\|\mathcal{K}\| = \sum_{r=0}^\infty \mathcal{K}(r) = \frac{-\gamma^2 Bd}{2\pi i} \oint_\Gamma \frac{z(1 - m(z))}{2\gamma Bz - \gamma^2 B(B+1)z^2} \, dz = \frac{-\gamma d}{4\pi i} \oint_\Gamma \frac{(1 - m(z))}{1 - \frac{1}{2}\gamma(B+1)z} \, dz.$$

We additionally can more generally evaluate a partial norm

$$\sum_{s=r}^\infty \mathcal{K}(s) = \frac{-\gamma d}{4\pi i} \oint_\Gamma \frac{(1 - m(z))}{1 - \frac{1}{2}\gamma(B+1)z}(1 - 2\gamma Bz + \gamma^2 B(B+1)z^2)^r \, dz.$$

Combining this with Proposition E.6, this leads directly to tight estimates for the kernel norm.

**Corollary G.1.** *When $2\alpha > 1$ and $\gamma(B+1) < 2$,*

$$\|\mathcal{K}\| = \frac{\gamma}{2} \sum_{j=1}^\infty \frac{j^{-2\alpha}}{1 - \frac{1}{2}j^{-2\alpha}\gamma(B+1)}(1 + o(1))$$

*When $2\alpha < 1$ (and recalling that we take $\gamma$ on the order $d^{2\alpha-1}$ in this case),*

$$\|\mathcal{K}\| = \frac{\gamma}{2} \frac{v^{1-2\alpha}}{1 - 2\alpha}(1 + o(1)).$$

*Furthermore, for any $\epsilon > 0$ there is an $M > 0$ so that if $\gamma Br > M$ then*

$$\frac{1}{\|\mathcal{K}\|} \sum_{s=r}^\infty \mathcal{K}(s) < \epsilon.$$

*Proof.* For the first case with $2\alpha > 1$, Proposition E.6 gives on $\Gamma_R$

$$1 - m = \frac{1}{d} \sum_{j=1}^v \frac{j^{-2\alpha}}{j^{-2\alpha} - z} + o(1/d)$$

Using this, and completing the $\Gamma_R$ contour via a vertical line, we get a residue contribution which matches the claim, up to some number of terms $j_0$ (which can be made as large as desired). Proposition E.5 and E.3 can be used to control the parts of the contour near 0 and in the middle.

For the second case $2\alpha < 1$, since $\gamma$ is small, we may deform the contour to be at a fixed distance from $[0, 1]$, and then once more we can use Proposition E.6 which gives in 1 step that

$$\|\mathcal{K}\| = \frac{\gamma}{2} \sum_{j=1}^v \frac{j^{-2\alpha}}{1 - \frac{1}{2}j^{-2\alpha}\gamma(B+1)} + O(\gamma d^{1-4\alpha}).$$

For the final statement, under the conditions given on $r$

$$|(1 - 2\gamma Bz + \gamma^2 B(B+1)z^2)^r| < \epsilon$$

uniformly over the contours, and the estimate follows directly.

$\square$

From here, we can derive the "sub-exponential" property of $\mathcal{K}$.

**Proposition G.2.** *Suppose $\alpha > \frac{1}{4}$. For any $\epsilon > 0$, there is an $M$ sufficiently large so that for $\gamma Br \in [M, d^{2\alpha}/M]$*

$$\sum_{s=0}^{r} \mathcal{K}(s)\mathcal{K}(r-s) \leq (2+\epsilon)\|\mathcal{K}\|\mathcal{K}(r)$$

*Proof.* We note that in the range of $r$ given, we can conclude that for any $\delta, \epsilon > 0$, by increasing $M$

$$\sum_{r\delta}^{\infty} \mathcal{K}(s) < \epsilon\|\mathcal{K}\|,$$

furthermore that for $s > r/2$

$$(1-\epsilon)\mathcal{K}_{pp}(s) < \mathcal{K}(s) < (1+\epsilon)\mathcal{K}_{pp}(s),$$

and that finally for $s > r/2$

$$(1-\epsilon)\mathcal{K}_{pp}(s) < \frac{\gamma^2 B}{2\alpha}\Gamma(2 - (1/(2\alpha)))(2\gamma Bs)^{-2+(1/(2\alpha))} < (1+\epsilon)\mathcal{K}_{pp}(s),$$

where the final estimate follows by estimating

$$\int_0^1 u^{1-1/(2\alpha)} \exp(-2\gamma Bur)\,\mathrm{d}u \asymp \int_0^\infty u^{1-1/(2\alpha)} \exp(-2\gamma Bur)\,\mathrm{d}u,$$

once $\gamma Br > M$, and moreover their ratio tends to 1 as $M \to \infty$.

With these estimates in place, we can break the estimate up as

$$\sum_{s=0}^{r} \mathcal{K}(s)\mathcal{K}(r-s) < 2\sum_{s=0}^{r\delta} \mathcal{K}(s)\mathcal{K}(r-s) + \sum_{s=r\delta}^{r(1-\delta)} \mathcal{K}(s)\mathcal{K}(r-s).$$

The final sum is bounded by

$$\sum_{r\delta}^{r(1-\delta)} \mathcal{K}(s)\mathcal{K}(r-s) \lesssim \epsilon(1+\epsilon)\|\mathcal{K}\|\mathcal{K}_{pp}(r/2).$$

Meanwhile, for the first sum,

$$2\sum_{s=0}^{r\delta} \mathcal{K}(s)\mathcal{K}(r-s) \leq 2(1+O(\epsilon))(1+O(\delta))\|\mathcal{K}\|\mathcal{K}_{pp}(r),$$

where we have used that $\mathcal{K}_{pp}(r(1-\delta)) \leq \frac{1+\epsilon}{1-\epsilon}(1+O(\delta))\mathcal{K}_{pp}(r)$. $\qquad\square$

## H Asymptotics of forcing function and kernel function

With this, we now analyze the asymptotics of each of these terms individually. These asymptotics often rely on a result about how close a Riemann sum is to its integral. We state below the main result of this nature that we used:

**Proposition H.1** (Trapezoidal Rule, [15])**.** *If $f$ is continuous, then for each integer $n > 0$, the integral of $f$ on $[a, b]$ is approximated by*

$$T_n(f) \stackrel{\text{def}}{=} \frac{b-a}{2n}\big(f(x_0) + 2f(x_1) + \ldots + 2f(x_{n-1}) + f(x_n)\big)$$

*where $x_i = a + i(b-a)/n$, $0 \leq i \leq n$. Define the error in the trapezoid rule*

$$E_n^T(f) \stackrel{\text{def}}{=} \left| T_n(f) - \int_a^b f(t)\,\mathrm{d}t \right|.$$

*If $f$ has an integrable first derivative as an improper integral, thens*

$$E_n^T(f) \leq \frac{b-a}{n}\int_a^b |f'(t)|\,\mathrm{d}t.$$

## H.1 Pure point forcing term, $\mathcal{F}_{pp}(r)$

In this section, we prove an asymptotic for the pure point forcing term, see (85),

$$\mathcal{F}_{pp}(r) = \frac{1}{2\alpha} \int_0^1 u^{(2\beta-1)/(2\alpha)} \exp(-2\gamma Bru) \; \mathrm{d}u.$$

**Proposition H.2** (Pure point forcing term). *Suppose $2\alpha + 2\beta > 1$. For any $\epsilon > 0$, there is an $M > 0$ so that for $\gamma Br \geq M$,*

$$|\mathcal{F}_{pp}(r) - g(r)| \leq \epsilon \times g(r)$$

*where*

$$g(r) \stackrel{def}{=} (2\alpha)^{-1}(2\gamma B)^{1/(2\alpha)-\beta/\alpha-1} \times \Gamma\big(\tfrac{\beta}{\alpha} - \tfrac{1}{2\alpha} + 1\big) \times r^{-(1+\beta/\alpha)+1/(2\alpha)}.$$

*Furthermore, for any $\tilde{M} > 0$, there exists some constants $C, \tilde{C}, c > 0$ independent of $d$ so that*

$$c \leq \mathcal{F}_{pp}(r) \leq C \quad \text{if } \gamma Br < \tilde{M},$$

*and if $r > \tilde{M}d^{2\alpha}$,*

$$\mathcal{F}_{pp}(r) \leq \tilde{C} \times \mathcal{F}_0(r).$$

*Proof.* First, a simple computation shows that

$$g(r) = (2\alpha)^{-1}(2\gamma Br)^{-(1+\beta/\alpha)+1/(2\alpha)} \int_0^\infty w^{(2\beta-1)/(2\alpha)} \exp(-w) \; \mathrm{d}w.$$

Let $\rho = -(1 + \beta/\alpha) + 1/(2\alpha)$. A simple computation, using the change of variables $w = 2\gamma Bru$, yields

$$\mathcal{F}_{pp}(r) = (2\alpha)^{-1}(2\gamma Br)^\rho \times \int_0^{2\gamma Br} w^{(2\beta-1)/(2\alpha)} \exp(-w) \; \mathrm{d}v.$$

Then we have that

$$|\mathcal{F}_{pp}(r) - g(r)| \leq (2\alpha)^{-1}(2\gamma Br)^\rho \left( \int_{2\gamma Br}^\infty w^{(2\beta-1)/(2\alpha)} \exp(-w) \; \mathrm{d}w \right)$$

$$\leq (2\alpha)^{-1}(2\gamma Br)^\rho \int_{2M}^\infty w^{(2\beta-1)/(2\alpha)} \exp(-w) \; \mathrm{d}w.$$

Since $\int_0^\infty w^{(2\beta-1)/(2\alpha)} \exp(-w) \; \mathrm{d}w$, there exists a $M$ large so that

$$\int_{2M}^\infty w^{(2\beta-1)/(2\alpha)} \exp(-w) \; \mathrm{d}w < \epsilon.$$

Thus the first result is shown.

If $\gamma Br < \tilde{M}$, then

$$\mathcal{F}_{pp}(r) \leq (2\alpha)^{-1} \int_0^1 u^{(2\beta-1)/(2\alpha)} \; \mathrm{d}u \leq C.$$

Moreover, we have that $\exp(-2\gamma Bru) \geq \exp(-2\tilde{M})$. Therefore, we get that

$$\mathcal{F}_{pp}(r) \geq \frac{\exp(-2\tilde{M})}{2\alpha} \int_0^1 u^{(2\beta-1)/(2\alpha)} \; \mathrm{d}u = c.$$

Now suppose $\gamma Br > \tilde{M}d^{2\alpha}$. By the previous part, we know that $\mathcal{F}_{pp}(r) \leq (1 + \epsilon)g(r)$. Moreover, we see that $g$ is decreasing. As a result, we see that up to constants

$$g(r) \leq g(\tilde{M}d^{2\alpha}) = C \times d^{-2\alpha-2\beta+1} \leq \tilde{C} \times \mathcal{F}_0(r).$$

for some constants $C, \tilde{C} > 0$. Hence the result is shown.

$\square$

## H.2 Model misspecification, point mass at $0$, $F_0(r)$

Recall, from Proposition F.1, the forcing function point mass at $0$, satisfies

$$\mathcal{F}_0(r) = \sum_{j=1}^{v} \frac{j^{-2\alpha-2\beta}}{1 + j^{-2\alpha}d^{2\alpha}\kappa(v/d)}\left(1 + \mathcal{O}(d^{-1})\right) \text{ where } \kappa(v/d) \text{ solves } 1 = \int_0^{v/d} \frac{\kappa}{\kappa + u^{2\alpha}}\, du.$$

(87)

In this section, we provide an asymptotic for $\mathcal{F}_0(r)$ (see Proposition H.3) which represents the limiting value the loss obtains as $r \to \infty$. Unlike the pure point process above, this asymptotic depends on whether $2\beta > 1$.

We begin by showing that the $\kappa$ defined implicitly in (87) is uniquely determined and dimensionless.

**Lemma H.1.** *Suppose $v$ and $d$ are admissible such that the ratio $\frac{v}{d} > 1$. Then the equation*

$$1 = \int_0^{v/d} \frac{\kappa}{\kappa + u^{2\alpha}}\, du$$

*has a unique solution $\kappa$ such that $0 < \kappa < \infty$.*

*Proof.* Let $w \overset{\text{def}}{=} \kappa$ and $F(w) \overset{\text{def}}{=} \int_0^{v/d} \frac{1}{w+u^{2\alpha}}\, du$ and set $G(w) = wF(w)$. To solve the fixed point equation, we want to find $G(w) = 1$. First, it is clear that $\lim_{w \to 0} G(w) = 0$. Second, we see that as $\lim_{w \to \infty} G(w) = \frac{v}{d} > 1$. As $G(w)$ is continuous, it follows that there exists a solution $\kappa$ to $G(\kappa) = 1$.

To show that $\kappa$ is unique, amounts to showing that $G(w)$ is strictly increasing for $w \geq 0$. First, we see that

$$G(w) = \int_0^{v/d} \frac{w + u^{2\alpha} - u^{2\alpha}}{w + u^{2\alpha}}\, du = \int_0^{v/d} 1 - \frac{u^{2\alpha}}{w + u^{2\alpha}}\, du.$$

We note that $w \mapsto \frac{u^{2\alpha}}{w+u^{2\alpha}}$ is strictly decreasing in $w$. So $w \mapsto 1 - \frac{u^{2\alpha}}{w+u^{2\alpha}}$ is strictly increasing in $w$. Hence $G(v)$ is strictly increasing and there is a unique solution to $G(\kappa) = 1$. $\qquad\square$

Now we give an asymptotic for $\mathcal{F}_0$.

**Proposition H.3** (Asymptotic for $\mathcal{F}_0$). *Suppose $v$ and $d$ are admissible such that the ratio $v/d > 1$ and suppose $2\alpha + 2\beta > 1$. Let $0 < \kappa(v/d) < \infty$ be the unique solution to*

$$1 = \int_0^{v/d} \frac{\kappa}{\kappa + u^{2\alpha}}\, du.$$

*Then as $d \to \infty$*

$$\mathcal{F}_0(r) \sim \begin{cases} \frac{d^{-2\alpha}}{\kappa}\left(\sum_{j=1}^{v} j^{-2\beta}\right), & \text{if } 2\beta > 1 \\ d^{1-2(\alpha+\beta)} \int_0^{v/d} \frac{u^{-2\beta}}{\kappa + u^{2\alpha}}\, du, & \text{if } 2\beta < 1. \end{cases}$$

*Proof.* We consider 2 cases. Let $\kappa = \kappa(v/d)$.

*Case 1: Suppose $2\beta > 1$:* Let $\tilde{C} \overset{\text{def}}{=} \sum_{j=1}^{v} j^{-2\beta}$, which is finite as $2\beta > 1$. Consider the following

$$\mathcal{E}_1(r) \overset{\text{def}}{=} \frac{\left|\sum_{j=1}^{v} \frac{j^{-2(\alpha+\beta)}}{j^{-2\alpha}\kappa d^{2\alpha}+1} - \frac{d^{-2\alpha}}{\kappa}\sum_{j=1}^{v} j^{-2\beta}\right|}{\frac{d^{-2\alpha}\tilde{C}}{\kappa}} = \frac{\sum_{j=1}^{v} \frac{j^{-2\beta}}{j^{-2\alpha}\kappa d^{2\alpha}+1}}{\tilde{C}}.$$

To handle the large $j$ values, we see that there exists a $j_0$ large so that

$$\frac{\sum_{j=j_0}^{v} \frac{j^{-2\beta}}{j^{-2\alpha}\kappa d^{2\alpha}+1}}{\tilde{C}} \leq \frac{1}{\tilde{C}} \sum_{j \geq j_0} j^{-2\beta} < \epsilon,$$

where we used that $j^{-2\alpha}\kappa d^{2\alpha} + 1 > 1$. For the small $j$, we use that $d$ can be large. Hence,

$$\sum_{j=1}^{j_0} \frac{j^{-2\beta}}{j^{-2\alpha}\kappa d^{2\alpha} + 1} \leq \sum_{j=1}^{j_0} \frac{j^{-2\beta}}{j_0^{-2\alpha}\kappa d^{2\alpha} + 1} \leq \frac{j_0}{j_0^{-2\alpha}\kappa d^{2\alpha} + 1}.$$

For sufficiently large $d$, we can make the right-hand-side small. Therefore, $\mathcal{E}_1(r)$ is small for sufficiently large $d$ and hence, the result holds.

*Case 2: Suppose $2\beta < 1$:* To show this case, we define the following errors

$$\mathcal{E}_{21}(r) \overset{\text{def}}{=} \frac{\left| \sum_{j=1}^{v} \frac{j^{-2(\alpha+\beta)}}{j^{-2\alpha}\kappa d^{2\alpha}+1} - d^{1-2(\alpha+\beta)} \int_{1/d}^{v/d} \frac{u^{-2\beta}}{\kappa+u^{2\alpha}}\,\mathrm{d}u \right|}{d^{1-2(\alpha+\beta)} \int_0^{v/d} \frac{u^{-2\beta}}{\kappa+u^{2\alpha}}\,\mathrm{d}u}$$

$$\mathcal{E}_{22}(r) \overset{\text{def}}{=} \frac{\left| d^{1-2(\alpha+\beta)} \int_{1/d}^{v/d} \frac{u^{-2\beta}}{\kappa+u^{2\alpha}}\,\mathrm{d}u - d^{1-2(\alpha+\beta)} \int_0^{v/d} \frac{u^{-2\beta}}{\kappa+u^{2\alpha}}\,\mathrm{d}u \right|}{d^{1-2(\alpha+\beta)} \int_0^{v/d} \frac{u^{-2\beta}}{\kappa+u^{2\alpha}}\,\mathrm{d}u}.$$

It is clear, for sufficiently large $d$, $\mathcal{E}_{22}(r)$ is small.

For the first error term, we use a Riemann sum approximation, that is,

$$\sum_{j=1}^{v} \frac{j^{-2(\alpha+\beta)}}{j^{-2\alpha}\kappa d^{2\alpha}+1} = d^{1-2(\alpha+\beta)} \times \frac{1}{d} \sum_{j=1}^{v} \frac{(j/d)^{-2(\alpha+\beta)}}{(j/d)^{-2\alpha}\kappa+1}.$$

Letting $a = 1/d$, $b = v/d$, $n = v-1$, $x_j = 1/d + j/d$, and $f(x) = \frac{x^{-2(\alpha+\beta)}}{x^{-2\alpha}\kappa+1}$, we can approximate the summation with an integral. Using Prop. H.1,

$$\mathcal{E}_{21}(r) \leq \frac{\frac{1}{d} \times \int_{1/d}^{v/d} |f'(x)|\,\mathrm{d}x}{\int_0^{v/d} \frac{u^{-2\beta}}{\kappa+u^{2\alpha}}\,\mathrm{d}u}.$$

One can check that $\frac{\int_{1/d}^{v/d} |f'(x)|\,\mathrm{d}x}{\int_0^{v/d} \frac{u^{-2\beta}}{\kappa+u^{2\alpha}}\,\mathrm{d}u} < C$ where $C$ is independent of $d$. For sufficiently large $d$,

$$\mathcal{E}_{21}(r) \leq \frac{\frac{1}{d} \times \int_{1/d}^{v/d} |f'(x)|\,\mathrm{d}x}{\int_0^{v/d} \frac{u^{-2\beta}}{\kappa+u^{2\alpha}}\,\mathrm{d}u} < C \times \frac{1}{d} < \epsilon.$$

Hence, Case 2 is shown. $\qquad\square$

## H.3 Absolutely continuous forcing function, $\mathcal{F}_{ac}(r)$

We now turn to the absolutely continuous forcing function, defined as

$$\mathcal{F}_{ac}(r) = \frac{c_\beta}{2\alpha} \int_{d^{-2\alpha}}^{1} u^{-1/(2\alpha)} d^{-1} \exp(-2\gamma Bru)\,\mathrm{d}u,$$

where $c_\beta = \sum_{j=1}^{v} j^{-2\beta}$ if $2\beta > 1$ and 0 otherwise. From this, we derive a simple asymptotic formula.

**Proposition H.4.** *There exists a constant $C(\alpha, \beta) > 0$ such that*

$$\mathcal{F}_{ac}(r) \leq \begin{cases} C \times \mathcal{F}_0(r), & \text{if } 2\beta > 1,\, 2\alpha < 1 \\ 0, & \text{if } 2\beta < 1. \end{cases} \tag{88}$$

*Suppose now $2\alpha > 1$ and $2\beta > 1$. For any $\epsilon > 0$, there is an $M > 0$ so that for $\gamma Br \in [M, d^{2\alpha}/M]$,*

$$|\mathcal{F}_{ac}(r) - g(r)| \leq \epsilon \times g(r)$$

$$\text{where} \quad g(r) \overset{\text{def}}{=} \Big( \sum_{j=1}^{v} j^{-2\beta} \Big)(2\gamma B)^{-1+1/(2\alpha)}(2\alpha)^{-1}\Gamma\Big(1 - \frac{1}{2\alpha}\Big) \times r^{-1+1/(2\alpha)} \times d^{-1}. \tag{89}$$

*Furthermore, for any $\tilde{M} > 0$, these exists some constants $C, c > 0$ independent of $d$ so that*

$$\mathcal{F}_{ac}(r) \leq \begin{cases} C \times d^{-1}, & \text{if } \gamma Br \leq \tilde{M} \\ c \times \mathcal{F}_0(r), & \text{if } \gamma Br \geq \tilde{M} d^{2\alpha}. \end{cases}$$

*Proof.* We proceed by cases. The case $2\beta < 1$ is immediate as $c_\beta$ is only non-zero for $2\beta > 1$.

**Case:** $2\beta > 1$ **and** $2\alpha < 1$**.** In this case, we just bound directly bound $\mathcal{F}_{ac}(r)$. Dropping the exponential, we get

$$\mathcal{F}_{ac}(r) \le \frac{c_\beta}{2\alpha} \int_{d^{-2\alpha}}^{1} u^{-1/(2\alpha)} d^{-1} \, \mathrm{d}u = \frac{c_\beta}{2\alpha} d^{-1} \left( \frac{d^{1-2\alpha} - d^{1/2-\alpha}}{\frac{1}{2\alpha} - 1} \right) \le \frac{c_\beta}{2\alpha(\frac{1}{2\alpha} - 1)} \times d^{-2\alpha}.$$

We know that $\mathcal{F}_0(r) \asymp d^{-2\alpha + \max\{0, 1-2\beta\}}$ and thus the result is shown.

Next we show (89).

**Case:** $2\beta > 1$ **and** $2\alpha > 1$**.** First, we make the following observation. The integral is

$$\frac{c_\beta}{2\alpha} \int_0^\infty u^{-1/(2\alpha)} d^{-1} \exp(-2\gamma Bru) \, \mathrm{d}u = g(r).$$

Define $C = \frac{c_\beta}{2\alpha}$. Let us consider the following

$$\mathcal{E} \stackrel{\text{def}}{=} \left| C \int_{d^{-2\alpha}}^{d^{-\alpha}} e^{-2\gamma Bur} u^{-1/(2\alpha)} d^{-1} \, \mathrm{d}u - C \int_0^\infty e^{-2\gamma Bur} u^{-1/(2\alpha)} d^{-1} \, \mathrm{d}u \right|.$$

First, we see

$$\mathcal{E} \le \underbrace{\left| C \int_0^{d^{-2\alpha}} e^{-2\gamma Bur} u^{-1/(2\alpha)} d^{-1} \, \mathrm{d}u \right|}_{\mathcal{E}_1} + \underbrace{\left| C \int_1^\infty e^{-2\gamma Bur} u^{-1/(2\alpha)} d^{-1} \, \mathrm{d}u \right|}_{\mathcal{E}_2}.$$

*Case:* $\mathcal{E}_1$. Suppose $\gamma Br \le 1/Md^{2\alpha}$. Here we can just use directly the $u$ and disregard the exponential:

$$\int_0^{d^{-2\alpha}} e^{-2\gamma Bur} u^{-1/(2\alpha)} d^{-1} \, \mathrm{d}u \le \int_0^{d^{-2\alpha}} u^{-1/(2\alpha)} d^{-1} \, \mathrm{d}u = \tilde{c} \times d^{-2\alpha}.$$

for some constant $\tilde{c}$. Now we have that

$$\frac{d^{-2\alpha}}{d^{-1}(\gamma Br)^{-1+1/(2\alpha)}} = d^{-2\alpha+1}(\gamma Br)^{1-1/(2\alpha)} \le d^{-2\alpha+1}(1/M)^{1-1/(2\alpha)} d^{2\alpha-1} = M^{-1+1/(2\alpha)}$$

By choosing $M$ large, this can be made small.

*Case:* $\mathcal{E}_2$. Suppose $\gamma Br \ge M$. Let us consider

$$\mathcal{E}_2 \le C \int_1^\infty d^{-1} e^{-2\gamma Bur} \, \mathrm{d}u \le d^{-1}(\gamma Br)^{-1} \exp(-2\gamma Br).$$

It follows that

$$\frac{d^{-1}(\gamma Br)^{-1} \exp(-2\gamma Br)}{d^{-1}(\gamma Br)^{-1+1/(2\alpha)}} = (\gamma Br)^{-1/(2\alpha)} \exp(-2\gamma Brd^{-\alpha})$$

$$\le \exp(-2M)(M)^{-1/(2\alpha)} = \exp(-2M)M^{-1/(2\alpha)}.$$

Therefore, by choosing $M$ large, we have that this can be small. This proves (89).

To finish the proof, let us first suppose that $\gamma Br \le \tilde{M}$. Then we have that

$$\mathcal{F}_{ac}(r) \le \frac{c_\beta}{2\alpha} \int_{d^{-2\alpha}}^{1} u^{-1/(2\alpha)} d^{-1} \, \mathrm{d}u \lesssim d^{-1}.$$

When $\gamma Br \ge \tilde{M} d^{2\alpha}$, we have that

$$\mathcal{F}_{ac}(r) \lesssim \int_{d^{-2\alpha}}^{1} \exp(-2\gamma Bru) \, \mathrm{d}u \le d^{-2\alpha} \exp(-2\gamma B\tilde{M} d^{2\alpha} d^{-2\alpha}) \lesssim \mathcal{F}_0(r).$$

$\square$

## H.4  Kernel function asymptotic.

We recall the term $\mathcal{K}_{pp}$ defined as

$$\mathcal{K}_{pp}(r) = \frac{\gamma^2 B}{2\alpha} \int_0^1 u^{1-1/(2\alpha)} \exp(-2\gamma Bur)\,\mathrm{d}u$$

We now give an asymptotic for such a function.

**Proposition H.5** ($\mathcal{K}_{pp}$ asymptotic). *Suppose $\alpha > 1/4$. For any $\epsilon > 0$, there is an $M > 0$ so that for $\gamma Br \geq M$,*

$$|\mathcal{K}_{pp}(r) - g(r)| \leq \epsilon \times g(r) \tag{90}$$

*where*

$$g(r) \stackrel{def}{=} (2\alpha)^{-1}\gamma^2 B(2\gamma B)^{-2+1/(2\alpha)} \times \Gamma\left(2 - \tfrac{1}{2\alpha}\right) \times r^{-2+1/(2\alpha)}.$$

*Moreover, for any $\tilde{M} > 0$, there exists constants $c, C, \hat{C} > 0$, such that when $2\alpha > 1$,*

$$c \leq \mathcal{K}_{pp}(r) \leq C, \quad \text{if } \gamma Br \leq \tilde{M}$$

*and when $2\alpha < 1$,*

$$\mathcal{K}_{pp}(r) \leq \hat{C} \times d^{2\alpha-1}, \quad \text{if } \gamma Br \leq \tilde{M}.$$

*Furthermore, for any $\tilde{M} > 0$, there exist a constant $\tilde{C} > 0$, such that*

$$\mathcal{K}_{pp}(r) \leq \tilde{C} \times \mathcal{F}_0(r), \quad \text{if } \gamma Br \geq \tilde{M}d^{2\alpha}.$$

*Proof.* The first part of the argument, (90), follows immediately from the proof of $\mathcal{F}_{pp}$, Prop. H.2.

If $2\alpha > 1$, then we always have $\gamma^2 B$ is constant order. Therefore, using the same argument as $\mathcal{F}_{pp}$ (see Prop. H.2), we get that there exists constant $c, C > 0$ such that $c \leq \mathcal{K}_{pp}(r) \leq C$ for $\gamma Br \leq \tilde{M}$.

If $2\alpha < 1$, then $\gamma \asymp d^{2\alpha-1}$. Therefore, for $\gamma Br \leq \tilde{M}$, we have that

$$\frac{\gamma^2 B}{2\alpha} \int_0^1 u^{1-1/(2\alpha)} \exp(-2\gamma Bur)\,\mathrm{d}u \asymp d^{2\alpha-1}\frac{\gamma B}{2\alpha} \int_0^1 u^{1-1/(2\alpha)} \exp(-2\gamma Bur)\,\mathrm{d}u \leq d^{2\alpha-1}C.$$

The later inequality follows using the same bounding argument as $\mathcal{F}_{pp}(r)$.

As $\gamma^2 B \leq C$, then the same argument in $\mathcal{F}_{pp}(r)$ shows for $\gamma Br \geq \tilde{M}d^{2\alpha}$ that $\mathcal{K}_{pp}(r) \leq \tilde{C}\mathcal{F}_0(r)$. □

We now turn to the last quantity that appears in the Volterra equation.

**Proposition H.6** (Forcing function norm). *Provided $2\beta > 1$, we have that*

$$\sum_{s=0}^{Md^{2\alpha}/(\gamma B)} \mathcal{F}(s) \lesssim \frac{1}{\gamma B}, \tag{91}$$

*for some constant $M > 0$.*

*Next, suppose $2\beta < 1$. Then there exists an $\tilde{M}, M > 0$ such that*

$$\mathcal{K}(r) \times \sum_{s=\tilde{M}/(\gamma B)}^{Md^{2\alpha}/(\gamma B)} \mathcal{F}(s) \leq \mathcal{F}(r) \quad \text{for all } \tilde{M} \leq \gamma Br \leq Md^{2\alpha}, \tag{92}$$

*and it follows for all $\gamma Br \leq Md^{2\alpha}$,*

$$\mathcal{K}(r) \times \sum_{s=0}^{Md^{2\alpha}/(\gamma B)} \mathcal{F}(s) \lesssim \mathcal{F}(r) + \frac{1}{\gamma B} \times \mathcal{K}(r). \tag{93}$$

*Proof.* Suppose $2\beta > 1$. First note that in this region $\sum_{s=0}^{Md^{2\alpha}/(\gamma B)} \mathcal{F}_0(r) \lesssim \frac{1}{\gamma B}$ for any fixed $M > 0$, Proposition H.3.

Next, let us consider the pure point part of $\mathcal{F}$, i.e., $\mathcal{F}_{pp}$. Choose $M$ and $\tilde{M}$ so that $\mathcal{F}_{pp}$ is behaving like the asymptotic in Proposition H.2, i.e., for $\gamma Br \geq \tilde{M}$,

$$\mathcal{F}_{pp}(r) \asymp (\gamma Br)^{-(1+\beta/\alpha)+1/(2\alpha)}.$$

and for $\gamma Br \leq \tilde{M}$, $\mathcal{F}_{pp}(r) \leq C$ for some constant $C > 0$ independent of $d$. It follows then that

$$\sum_{r=0}^{\tilde{M}/(\gamma B)} \mathcal{F}_{pp}(r) \lesssim \frac{1}{\gamma B}.$$

To handle the rest of the sum, we see that

$$
\begin{aligned}
\sum_{r=\tilde{M}/(\gamma B)}^{Md^{2\alpha}/(\gamma B)} \mathcal{F}_{pp}(r) &\asymp \sum_{r=\tilde{M}/(\gamma B)}^{Md^{2\alpha}/(\gamma B)} (\gamma Br)^{-(1+\beta/\alpha)+1/(2\alpha)} \asymp \frac{1}{\gamma B} \sum_{r=\tilde{M}}^{Md^{2\alpha}} r^{-(1+\beta/\alpha)+1/(2\alpha)} \\
&= \frac{(d^{2\alpha})^{-\beta/\alpha+1/(2\alpha)}}{\gamma B} \times \frac{1}{d^{2\alpha}} \sum_{r=\tilde{M}}^{Md^{2\alpha}} \left(\frac{r}{d^{2\alpha}}\right)^{-(1+\beta/\alpha)+1/(2\alpha)} \\
&\leq \frac{(d^{2\alpha})^{-\beta/\alpha+1/(2\alpha)} M}{\gamma B} \int_{\tilde{M}/d^{2\alpha}}^{M+\tilde{M}/d^{2\alpha}} x^{-(1+\beta/\alpha)+1/(2\alpha)} \, dx \\
&= \frac{(d^{2\alpha})^{-\beta/\alpha+1/(2\alpha)} M2\alpha}{\gamma B} \times \left. \frac{x^{-\beta/\alpha+1/(2\alpha)}}{1-2\beta} \right|_{\tilde{M}/d^{2\alpha}}^{M+\tilde{M}/d^{2\alpha}} \\
&\lesssim \frac{1}{\gamma B}.
\end{aligned}
$$

Here we use that the Riemann sum approximation with $a = \frac{\tilde{M}}{d^{2\alpha}}$, $b = M + \frac{\tilde{M}}{d^{2\alpha}}$, $n = Md^{2\alpha}$ and $f(x) = x^{-(1+\beta/\alpha)+1/(2\alpha)}$.

Using a similar argument for $\mathcal{F}_{ac}(r)$ (Proposition H.4) when $2\alpha > 1$ (otherwise we do not need to worry about $\mathcal{F}_{ac}$), we have that

$$
\begin{aligned}
\sum_{r=\tilde{M}/(\gamma B)}^{Md^{2\alpha}/(\gamma B)} \mathcal{F}_{ac}(r) &\asymp \frac{d^{-1}}{\gamma B} \sum_{r=\tilde{M}}^{Md^{2\alpha}} s^{-1+1/(2\alpha)} = \frac{M}{\gamma B} d^{-2\alpha} \sum_{\tilde{M}}^{Md^{2\alpha}} \left(\frac{s}{d^{2\alpha}}\right)^{-1+1/(2\alpha)} \\
&\lesssim \frac{M}{\gamma B} \int_0^M x^{-1+1/(2\alpha)} \, dx \asymp \frac{1}{\gamma B}.
\end{aligned}
$$

When $r \leq \tilde{M}/(\gamma B)$, we have that $\mathcal{F}_{ac}(r) \lesssim d^{-1}$. Hence, $\sum_{r=0}^{\tilde{M}/(\gamma B)} \mathcal{F}_{ac}(r) \lesssim \frac{1}{\gamma B}$.

The first result, (91), then follows from Corollary F.1.

Consider $2\beta < 1$ and $2\alpha < 1$. We do not need to worry about $\mathcal{F}_{ac}$ in this region because it does not exist in this region. Choose $M$ and $\tilde{M}$ so that both $\mathcal{K}_{pp}$ and $\mathcal{F}_{pp}$ are in their asymptotic regions and, using Proposition G.1, $\mathcal{K}(r) \asymp \mathcal{K}_{pp}(r)$. Using a similar argument as above, we estimate the summation of $\mathcal{F}_{pp}$ as an integral. For any $r \in [\tilde{M}/(\gamma B), Md^{2\alpha}/(\gamma B)]$

$$
\begin{aligned}
\sum_{s=\tilde{M}/(\gamma B)}^{r} \mathcal{F}_{pp}(s) &\lesssim \frac{(d^{2\alpha})^{-\beta/\alpha+1/(2\alpha)} M2\alpha}{\gamma B} \times \left. \frac{x^{-\beta/\alpha+1/(2\alpha)}}{1-2\beta} \right|_{\tilde{M}/d^{2\alpha}}^{r\gamma B/d^{2\alpha}+\tilde{M}/d^{2\alpha}} \\
&\lesssim \frac{1}{\gamma B} (\gamma Br)^{-\beta/\alpha+1/(2\alpha)}.
\end{aligned}
$$

Using the asymptotic for $\mathcal{K}_{pp}$ (Proposition H.5),

$$\mathcal{K}(r) \times \sum_{s=\tilde{M}/(\gamma B)}^{r} \mathcal{F}_{pp}(s) \lesssim \gamma \times (\gamma Br)^{-\beta/\alpha+1/(2\alpha)} \times (\gamma Br)^{-2+1/(2\alpha)}.$$

We will show that this is less than $\mathcal{F}_{pp}(r)$. Using the asymptotic for $\mathcal{F}_{pp}(r)$ (Proposition H.2), let us suppose

$$\gamma \times (\gamma Br)^{-\beta/\alpha+1/(2\alpha)} \times (\gamma Br)^{-2+1/(2\alpha)} \leq (\gamma Br)^{-1-\beta/\alpha+1/(2\alpha)}$$

$$\Leftrightarrow \quad \gamma \leq (\gamma Br)^{1-1/(2\alpha)}.$$

In this region, the learning rate is $\gamma \asymp d^{2\alpha-1}$. Thus, we see that

$$d^{2\alpha-1} \leq (\gamma Br)^{(2\alpha-1)/(2\alpha)}$$

$$\Leftrightarrow \quad d^{(2\alpha-1)\frac{2\alpha}{2\alpha-1}} \geq (\gamma Br)^{\frac{2\alpha-1}{2\alpha} \cdot \frac{2\alpha}{2\alpha-1}}$$

$$\Leftrightarrow \quad d^{2\alpha} \geq (\gamma Br).$$

This is true and so we have that

$$\mathcal{K}(r) \times \sum_{s=\tilde{M}/\gamma B}^{r} \mathcal{F}_{pp}(r) \lesssim \mathcal{F}_{pp}(r), \qquad \text{for all } r \in [\tilde{M}/(\gamma B), Md^{2\alpha}/(\gamma B)].$$

For $\mathcal{F}_0$, with $r \in [\tilde{M}/(\gamma B), Md^{2\alpha}/(\gamma B)]$

$$\sum_{s=\tilde{M}/(\gamma B)}^{r} \mathcal{F}_0 \lesssim (\gamma Br) \times d^{1-2\beta-2\alpha} \times \frac{1}{\gamma B}$$

Therefore, we get that

$$\mathcal{K}(r) \times \sum_{s=\tilde{M}/(\gamma B)}^{r} \mathcal{F}_0(r) \lesssim (\gamma Br) \times d^{1-2\beta-2\alpha} \times \gamma \times (\gamma Br)^{-2+1/(2\alpha)}$$

$$\lesssim d^{-2\beta}(\gamma Br)^{-1+1/(2\alpha)}.$$

We will show that this is less than $\mathcal{F}_{pp}$. For this, we see

$$d^{-2\beta}(\gamma Br)^{-1+1/(2\alpha)} \lesssim (\gamma rB)^{-\beta/\alpha-1+1/(2\alpha)}$$

$$\Leftrightarrow \quad d^{-2\beta} \lesssim (\gamma Br)^{-\beta/\alpha}$$

$$\Leftrightarrow \quad (\gamma Br)^{\beta/\alpha} \lesssim d^{2\beta}$$

$$\Leftrightarrow \quad (\gamma Br) \lesssim d^{2\alpha}.$$

Hence, we have that

$$\mathcal{K}(r) \times \sum_{s=\tilde{M}/(\gamma B)}^{r} \mathcal{F}_0 \leq \mathcal{F}_{pp}(r), \quad \text{for all } r \in [\tilde{M}/(\gamma B), Md^{2\alpha}/(\gamma B)].$$

Since there is no $\mathcal{F}_{ac}$ in this region, we immediately get from Corollary F.1

$$\mathcal{K}(r) \times \sum_{s=\tilde{M}/(\gamma B)}^{r} \mathcal{F}(r) \lesssim \mathcal{F}_{pp}(r), \quad \text{for all } r \in [\tilde{M}/(\gamma B), Md^{2\alpha}/(\gamma B)].$$

For $s \in [0, \tilde{M}/(\gamma B)]$, we have that $\mathcal{F}(s) \lesssim C$. Thus we immediately get that

$$\mathcal{K}(r) \times \sum_{s=0}^{\tilde{M}/(\gamma B)} \mathcal{F}(s) \lesssim \mathcal{K}(r) \times \frac{1}{\gamma B},$$

for all $r$. This proves the result for $2\beta < 1$ and $2\alpha < 1$.

Consider $2\beta < 1$ and $2\alpha > 1$. As in the previous case, we do not need to consider $\mathcal{F}_{ac}$ as it does not exist here. The proof will be similar to the previous case. Choose $M$ and $\tilde{M}$ so that both $\mathcal{K}_{pp}$ and $\mathcal{F}_{pp}$ are in their asymptotic regions and, using Proposition G.1, $\mathcal{K}(r) \asymp \mathcal{K}_{pp}(r)$.

First, by the same argument as in $2\beta < 1$ and $2\alpha > 1$, we immediately have for $s \in [0, \tilde{M}/(\gamma B)]$, we have that $\mathcal{F}(s) \lesssim C$,

$$\mathcal{K}(r) \times \sum_{s=0}^{\tilde{M}/(\gamma B)} \mathcal{F}(s) \lesssim \mathcal{K}(r) \times \frac{1}{\gamma B},$$

for all $r$.

As before, we have for any $r \in [\tilde{M}/(\gamma B), Md^{2\alpha}/(\gamma B)]$

$$\mathcal{K}(r) \times \sum_{s=\tilde{M}/(\gamma B)}^{r} \mathcal{F}_{pp}(s) \lesssim \gamma \times (\gamma B r)^{-\beta/\alpha + 1/(2\alpha)} \times (\gamma B r)^{-2+1/(2\alpha)}.$$

Note here that $\gamma$ is constant. We will show that this is less than $\mathcal{F}_{pp}(r)$. Using the asymptotic for $\mathcal{F}_{pp}(r)$ (Proposition H.2), let us suppose

$$(\gamma B r)^{-\beta/\alpha + 1/(2\alpha)} \times (\gamma B r)^{-2+1/(2\alpha)} \leq (\gamma B r)^{-1-\beta/\alpha+1/(2\alpha)}$$
$$\Leftrightarrow \quad 0 \leq (\gamma B r)^{1-1/(2\alpha)}.$$

Hence, we have that

$$\mathcal{K}(r) \times \sum_{s=\tilde{M}/\gamma B}^{r} \mathcal{F}_{pp}(r) \lesssim \mathcal{F}_{pp}(r), \qquad \text{for all } r \in [\tilde{M}/(\gamma B), Md^{2\alpha}/(\gamma B)].$$

For $\mathcal{F}_0$, with $r \in [\tilde{M}/(\gamma B), Md^{2\alpha}/(\gamma B)]$

$$\sum_{s=\tilde{M}/(\gamma B)}^{r} \mathcal{F}_0 \lesssim (\gamma B r) \times d^{1-2\beta-2\alpha} \times \frac{1}{\gamma B}.$$

Therefore, we get that

$$\mathcal{K}(r) \times \sum_{s=\tilde{M}/(\gamma B)}^{r} \mathcal{F}_0(r) \lesssim (\gamma B r) \times d^{1-2\beta-2\alpha} \times (\gamma B r)^{-2+1/(2\alpha)}$$
$$\lesssim d^{-2\alpha+1-2\beta}(\gamma B r)^{-1+1/(2\alpha)}.$$

We will show that this is less than $\mathcal{F}_{pp}$. For this, we see

$$d^{-2\alpha+1-2\beta}(\gamma B r)^{-1+1/(2\alpha)} \lesssim (\gamma r B)^{-\beta/\alpha-1+1/(2\alpha)}$$
$$\Leftrightarrow \quad d^{-2\alpha+1-2\beta} \lesssim (\gamma B r)^{-\beta/\alpha}$$
$$\Leftrightarrow \quad (\gamma B r)^{\beta/\alpha} \lesssim d^{2\beta+2\alpha-1}.$$

Now we see that $(\gamma B r)^{\beta/\alpha} \lesssim d^{2\beta} \lesssim d^{2\beta+2\alpha-1}$. Hence, we have that

$$\mathcal{K}(r) \times \sum_{s=\tilde{M}/(\gamma B)}^{r} \mathcal{F}_0 \leq \mathcal{F}_{pp}(r), \quad \text{for all } r \in [\tilde{M}/(\gamma B), Md^{2\alpha}/(\gamma B)].$$

The result is thus shown in this case. □

# I Optimizing over batch and learning rate

The previous sections use batch size $B = 1$ and the maximal learning rate allowed. In this section, we consider optimizing compute-optimal curves with respect to batch size and learning rate, i.e., find $d^\star, \gamma^\star, B^\star$ such that

$$(d^\star, \gamma^\star, B^\star) \in \arg\min_{d,\gamma,B} \in \arg\min \mathcal{P}(\tfrac{f}{dB}, d, \gamma) \quad \text{s.t. } \gamma B < 1 \text{ and } \|\mathcal{K}_{pp}\| < 1. \tag{94}$$

## I.1 Optimal batch size

We see that batch essentially scales out of the problem and therefore the batch has no effect on the compute-optimal curves. To see this, we observe from Table 5 that

$$\mathcal{F}_{pp}(r) \asymp (\gamma Br)^{-(1+\beta/\alpha)+1/(2\alpha)} \quad \Rightarrow \quad \mathcal{F}_{pp}(\tfrac{\mathfrak{f}}{dB}) \asymp (\tfrac{\gamma\mathfrak{f}}{d})^{-(1+\beta/\alpha)+1/(2\alpha)}$$

$$\mathcal{F}_{ac}(r) \asymp (\gamma Br)^{-1+1/(2\alpha)} \times d^{-1} \quad \Rightarrow \quad \mathcal{F}_{ac}(\tfrac{\mathfrak{f}}{dB}) \asymp (\tfrac{\gamma\mathfrak{f}}{d})^{-1+1/(2\alpha)} \times d^{-1} \tag{95}$$

$$\tfrac{1}{\gamma B}\mathcal{K}_{pp}(r) \asymp \gamma(\gamma Br)^{-2+1/(2\alpha)} \quad \Rightarrow \quad \tfrac{1}{\gamma B}\mathcal{K}_{pp}(\tfrac{\mathfrak{f}}{dB}) \asymp \gamma(\tfrac{\gamma\mathfrak{f}}{d})^{-2+1/(2\alpha)}.$$

It immediately follows that the batch size has no effect on the compute-optimal curves. Therefore any batch size (e.g., $B = 1$) that satisfies the necessary and sufficient condition for convergence (Prop. C.2), that is, $\gamma(B+1) < 2$, will yield the same compute-optimal curves.

This is not necessarily true for the learning rate as we will see in the next section.

## I.2 Optimal learning rate

Without loss of generality, we let the batch size $B = 1$. From the expressions in (95), we see that $\mathcal{F}_{pp}$ and $\mathcal{F}_{ac}$ are monotonically decreasing in learning rate $\gamma$. Moreover in Phase III, $\tfrac{1}{\gamma B}\mathcal{K}_{pp}$, is also monotonically decreasing in the learning rate. Therefore, in Phases I, II, and III, the optimal learning rate choice is to choose $\gamma$ maximally. In the cases of Phase Ia, II, and III, this would mean $\gamma$ constant (see Prop C.2) and in Phase Ib, Ic, $\gamma \sim d^{2\alpha-1}$. It remains to understand the effect of the learning rate in Phase IV.

From Proposition C.6, we know that the loss is given by

$$\mathcal{P}(\tfrac{\mathfrak{f}}{d}, d, \gamma) \asymp \mathcal{F}_0(\tfrac{\mathfrak{f}}{d}) + \mathcal{F}_{pp}(\tfrac{\mathfrak{f}}{d}) + \tfrac{1}{\gamma B}\mathcal{K}_{pp}(\tfrac{\mathfrak{f}}{d}) \asymp d^{-2\alpha} + (\tfrac{\gamma\mathfrak{f}}{d})^{\rho} + \gamma(\tfrac{\gamma\mathfrak{f}}{d})^{-2+1/(2\alpha)},$$

$$\text{where } \rho \overset{\text{def}}{=} \tfrac{1}{2\alpha} - \tfrac{\beta}{\alpha} - 1.$$

By taking derivatives, we see that

$$\gamma^\star \asymp (\tfrac{\mathfrak{f}}{d^\star})^{\alpha/\beta-1} \quad \text{and} \quad d^\star(\mathfrak{f}) \asymp \mathfrak{f}^{\rho/(\rho-2\beta)}.$$

We need to check that $\gamma^\star$ is feasible, i.e., $\gamma^\star < 1$ (which it is) and $\gamma^\star < d^{2\alpha-1}$. For the later, a simple check shows that

$$\gamma^\star \asymp d^{\frac{4\alpha^2-4\alpha\beta}{2\alpha+2\beta-1}} < d^{2\alpha-1}$$

when $\alpha > 1/4$ and $2\beta > 1$, i.e., precisely Phase IV. The compute-optimal curve in Phase IV with optimal stepsize is the following.

**Proposition I.1** (Phase IV, optimal $\gamma$, compute-optimal curve). *Suppose $1/4 < \alpha < 1/2$ and $2\beta > 1$. Then*

$$\gamma^\star \asymp \mathfrak{f}^{\frac{4\alpha(\alpha-\beta)}{4\alpha\beta+2\alpha+2\beta-1}}, \quad d^\star(\mathfrak{f}) \asymp \mathfrak{f}^{\frac{2\alpha+2\beta-1}{4\alpha\beta+2\alpha+2\beta-1}}, \quad \text{and} \quad \mathcal{P}^\star(\mathfrak{f}) \asymp \mathfrak{f}^{\frac{-2\alpha(2\alpha+2\beta-1)}{4\alpha\beta+2\alpha+2\beta-1}}.$$

*The trade off occurring where $\tfrac{1}{\gamma B}\mathcal{K}_{pp} = \mathcal{F}_0$.*

We note that there is only one Phase IV (and no sub-phases).

# J Experimental Results

To measure the exponents of the scaling law and parameter count, we follow approach[12] 1 and 2 from [24]. We explain the method below using $(\alpha, \beta) = (0.5, 0.7)$ as an example. The theoretical prediction of the scaling law and parameter count exponents for this example are $\eta = 0.5$ and $\xi = 0.5$, resp. (see Table 2). We then repeat this procedure for total of 32 pairs of $(\alpha, \beta)$ in the phase diagram; see Fig. 15 and Fig. 16. The theoretical predictions of these two exponents are shown in the heatmaps Fig. 8.

First, we run SGD for parameter counts

$$d \in [200, 300, 400, 600, 800, 1200, 1600, 2400, 3200, 4800, 6400, 9600, 12800].$$

The SGD learning curves for $(\alpha, \beta) = (0.5, 0.7)$ with parameters $d \in [800, 1600, 3200, 6400, 12800]$ are shown in Fig. 9a.

---

[12]We did not use approach 3 in [24], which is more subtle than the other two; see [7].

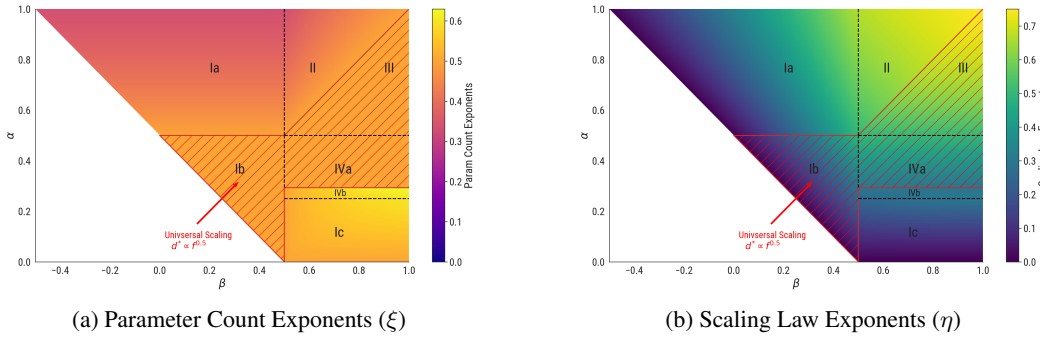

(a) Parameter Count Exponents ($\xi$)  (b) Scaling Law Exponents ($\eta$)

Figure 8: Theoretical predictions of parameter count and scaling law exponents.

## J.1 Measuring the Scaling Law Exponent

We follow Sec 3.1 in [24]. First, we choose an IsoFLOP window $[\mathfrak{f}_{\min}, \mathfrak{f}_{\max}]$ and construct $\mathfrak{f}_j$'s using a geometric spacing between $\mathfrak{f}_1 = \mathfrak{f}_{\min}$ and $\mathfrak{f}_n = \mathfrak{f}_{\max}$. For each IsoFLOP slice, $\mathfrak{f}_j$, (e.g., $\mathfrak{f}_j = 2e7$ is the vertical line in Fig. 9a), we find the minimum loss across all $d$. We denote this minimum value by $\mathscr{P}^\star(\mathfrak{f}_j)$ and the associated optimal parameter by $d^\star(\mathfrak{f}_j)$. As an example, in Fig. 9a, $\mathscr{P}^\star(\mathfrak{f}_j) = 1.6e - 3$ and the associated optimal parameter $d^\star(\mathfrak{f}_j) = 6400$.

We obtain the compute-optional frontier (highlighted in red in Fig. 9b) by plotting

$$[(\mathfrak{f}_j, \mathscr{P}^\star(\mathfrak{f}_j)]_{1 \le j \le n}, \tag{96}$$

and the optimal parameter count

$$[(\mathfrak{f}_j, d^\star(\mathfrak{f}_j)]_{1 \le j \le n}. \tag{97}$$

We then fit a power-law curve $\mathscr{P}^\star(\mathfrak{f}) = a \times \mathfrak{f}^{-\hat{\eta}}$ to predict the relationship between the compute $\mathfrak{f}$ and optimal loss $\mathscr{P}^\star$. This is shown as the dashed line in Fig. 9b. For $(\alpha, \beta) = (0.5, 0.7)$, this gives

$$\mathscr{P}^\star(\mathfrak{f}) = 22.61 \times \mathfrak{f}^{-0.515},$$

whereas our theoretical result predicts

$$\mathcal{P}^\star(\mathfrak{f}) \asymp \mathfrak{f}^{-0.5}.$$

## J.2 Measuring Parameter Count Exponent: Approach 0

One benefit of our theoretical framework is that the solution of the Volterra equation (eq. 10) is deterministic. As such, precise numerical evaluation can determine the instantaneous slope of the compute-optimal curves using a new approach that is not necessarily feasible when dealing with noisy

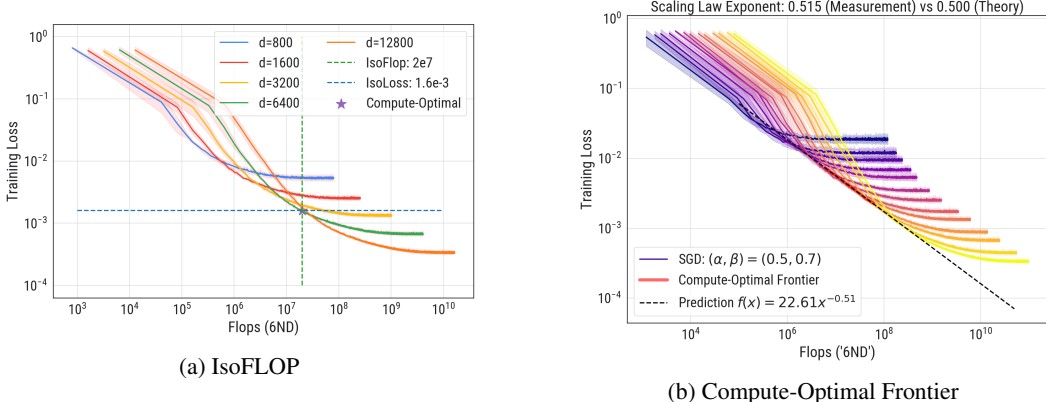

(a) IsoFLOP  (b) Compute-Optimal Frontier

Figure 9: Measuring the scaling law exponent for $(\alpha, \beta) = (0.5, 0.7)$.

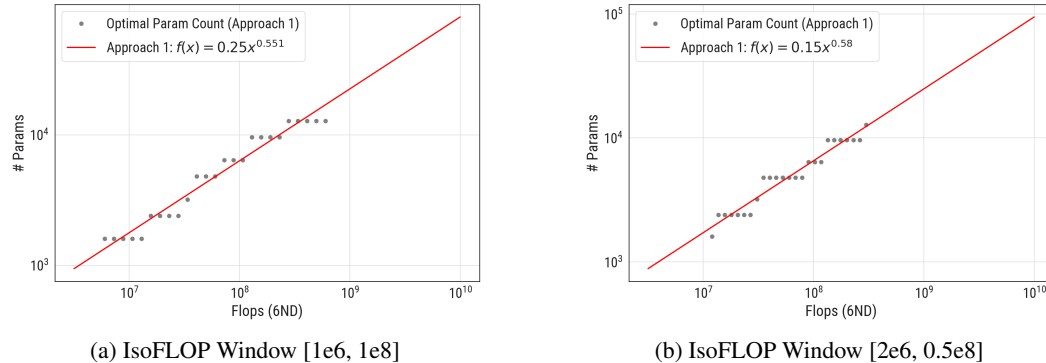

(a) IsoFLOP Window [1e6, 1e8]                    (b) IsoFLOP Window [2e6, 0.5e8]

Figure 10: 2 different IsoFLOP windows for measuring the parameter count exponent with Approach 1 for $(\alpha, \beta) = (0.5, 0.7)$.

SGD curves. Specifically, we search for the unique tangent line that intersects the loss-versus-flops curves for two adjacent values of $d$, i.e. we numerically solve the following system for $f_1$ and $f_2$:

$$P_1'(f_1) = P_2'(f_2) = \frac{P_2(f_2) - P_1(f_1)}{f_2 - f_1}, \tag{98}$$

where $P_1$ is the loss curve for $d = d_1$ and $P_2$ is the loss curve for $d = d_2$. When $d_1$ and $d_2$ are close, we obtain an accurate estimate of the parameter count exponent by measuring the discrete logarithmic derivative, $(\log(d_2) - \log(d_1))/(\log(f_2^*) - \log(f_1^*))$.

### J.3 Measuring Parameter Count Exponent: Approach 1

To predict the optimal parameter count exponent, we fit the function $d^\star = a \times f^b$, $a, b$ constants, to the measurements in (97) (see e.g., Fig. 10a). For the example $(\alpha, \beta) = (0.5, 0.7)$ (Fig. 10a), this approach gives

$$d^\star = .25 \times f^{0.551}. \tag{99}$$

Note that the fit of the exponent is very sensitive to the choice of IsoFLOP window. When we change the window from $[1e6, 1e8]$ to $[2e6, 0.5e8]$, the parameter count exponent changes from $0.51$ to $0.58$, as shown in Fig.10b. The theoretical prediction of this exponent is $0.5$.

### J.4 Measuring Parameter Count Exponent: Approach 2

For each IsoFLOP slice, $f_j$, we obtain a set of training loss values depending on $d$, $\{\mathscr{P}(f_j, d_i)\}_{1 \leq i \leq m}$. In our running example, $d_1 = 200$ and $d_m = 12800$ (see Fig. 9a). We then fit a parabola (quadratic

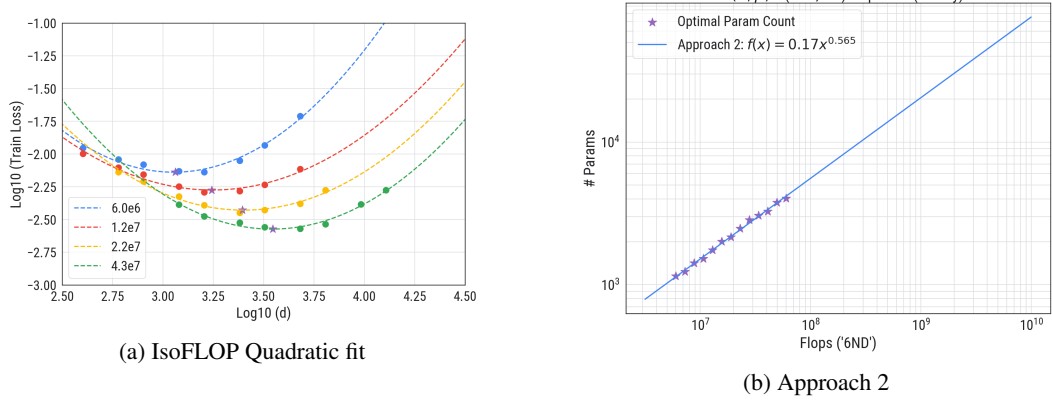

(a) IsoFLOP Quadratic fit

(b) Approach 2

Figure 11: Measuring parameter count exponent with Approach 2 for $(\alpha, \beta) = (0.5, 0.7)$.

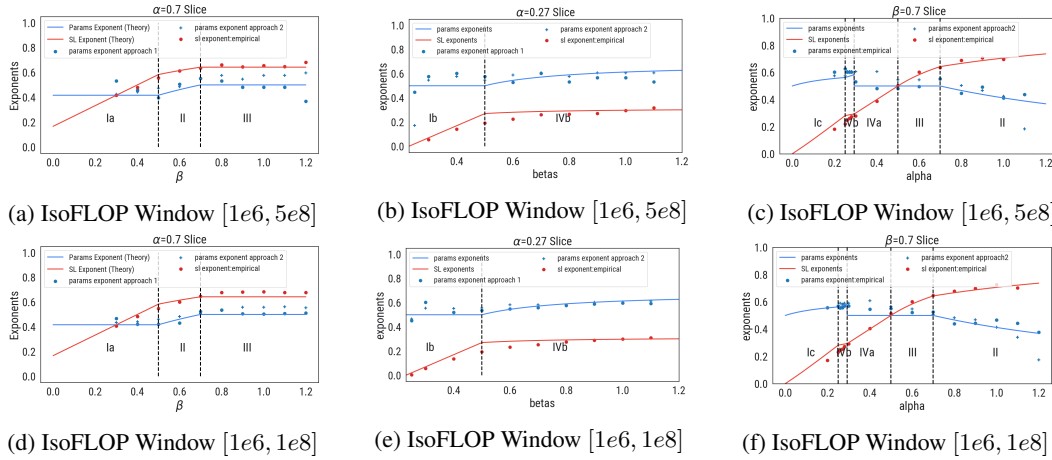

(a) IsoFLOP Window $[1e6, 5e8]$     (b) IsoFLOP Window $[1e6, 5e8]$     (c) IsoFLOP Window $[1e6, 5e8]$

(d) IsoFLOP Window $[1e6, 1e8]$     (e) IsoFLOP Window $[1e6, 1e8]$     (f) IsoFLOP Window $[1e6, 1e8]$

Figure 12: The empirical measurements of the scaling law exponent and parameter count exponent are sensitive to the choice of the IsoFLOP windows. **Top:** larger IsoFLOP window $[1e6, 5e8]$ **Bottom:** smaller IsoFLOP window $[1e6, 1e8]$.

function) to $\{(\log \mathscr{P}(\mathfrak{f}_j, d_i), \log d_i)\}_{1 \le i \le m}$, i.e., we find $(a, b, c)$ such that

$$\log \mathscr{P}(\mathfrak{f}_j, d_i) = a \log^2 d_i + b \log d_i + c.$$

This is shown in Fig. 11a. After solving for $(a, b, c)$, we find the $d^\star(f_j)$ that minimizes $a^2 \log^2 d + b \log d + c$. Repeating this procedure for all $\mathfrak{f}_j$'s gives a set of pairs $\{(\mathfrak{f}_j, d_j^\star)\}_{1 \le j \le n}$. In the final step, we power-law fit this set. For the example $(\alpha, \beta) = (0.5, 0.7)$ (see Fig. 11b), this gives

$$d^\star = 0.17 \times \mathfrak{f}^{0.565}.$$

### J.5 Exponents comparison: Theory vs Measurement

We compare the empirical measurements of the exponents against their theoretical predictions in Fig. 12. We chose three slices across the phase diagram

1. $\alpha = 0.7$ Slice (Fig. 12a), in which $(\alpha, \beta)$ goes from Phase Ia, II and III.
2. $\alpha = 0.27$ Slice (Fig. 12b), in which $(\alpha, \beta)$ goes from Phase Ib to Phase IVb.
3. $\beta = 0.7$ Slice (Fig. 12c),, in which $(\alpha, \beta)$ goes from Phase Ic, IVb, IVa, III and to II.

For the scaling law exponents, the empirical measurement agrees with the theoretical prediction quite well. For the parameter count, the agreement is good but not as good as that of the scaling law exponents. Noticeably, there is disagreement between Approach 1 and Approach 2. Such disagreement is not surprising, as empirical measurements are sensitive to the choice of the IsoFLOP windows and we use the *same* IsoFLOP window $[1e6, 5e8]$ for all $(\alpha, \beta)$. This is clearly suboptimal. We briefly discuss this in the next subsection.

### J.6 Instantaneous slope

In this section, we demonstrate that there can be strong finite-size $d$ effects in the measurements of the scaling law and parameter count exponents. We measure the instantaneous slope as a function of parameter count $d$ for the Volterra equation (10). See Fig. 13. To do so, we generate the Volterra solutions for a geometrically spaced sequence of $d$'s with ratio $1.05$. We then apply Approach 1 with a very dense IsoFLOP window (100 IsoFLOP Slices between $[2e4, 2e7]$). We then slide a smaller IsoFLOP window (20 IsoFLOP slices) from left to right to generate a sequence of scaling law (parameter count) exponent, as shown in the middle (right) plot in Fig. 13. These exponents varying slowly when the window slides from a small flops regime to a large flops regime. For example, the scaling law exponent $\eta$ changes from $\eta = 0.440$ to $\eta = 0.413$ from left to right, while the parameter count exponent changes from $\xi = 0.450 \rightarrow 0.575$. Using the global window (100 IsoFLOP) to measure these exponents, we have $\eta = 0.42$ and $\xi = 0.52$ which are very close to their average over all small windows: $\eta = 0.418$ and $\xi = 0.526$.

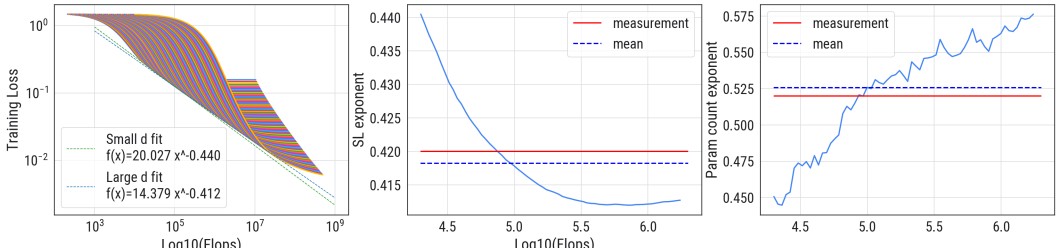

Figure 13: **Instantaneous Slope. (Left)** Volterra equation (10) dynamics for a highly dense grid of $d$. We also plot the compute-optimal front obtained from using the left small window (small flops regime) and the right window (larger flops regime). **(Middle)** Shows the evolving measurements of scaling exponents when the flops increases. **(Right)** Shows the evolving measurements of parameter count exponents when the flops increases.

### J.7 Negative $\beta$.

In Fig. 14, we show that our theoretical results work well for $\beta < 0$.

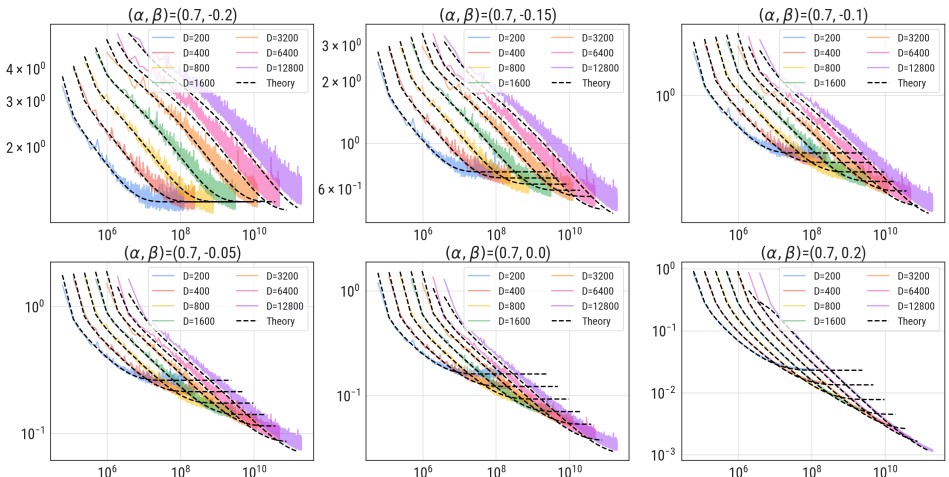

Figure 14: **Negative $\beta$.** Sweeping $\beta = -0.2$ to $\beta = 0.2$. We see good agreement between Volterra (theory) dynamics and SGD dynamics.

### J.8 Additional plots for different phases

We summarize the measurement of the scaling law exponents and optimal parameter count exponents in Fig. 15 and Fig. 16, resp.

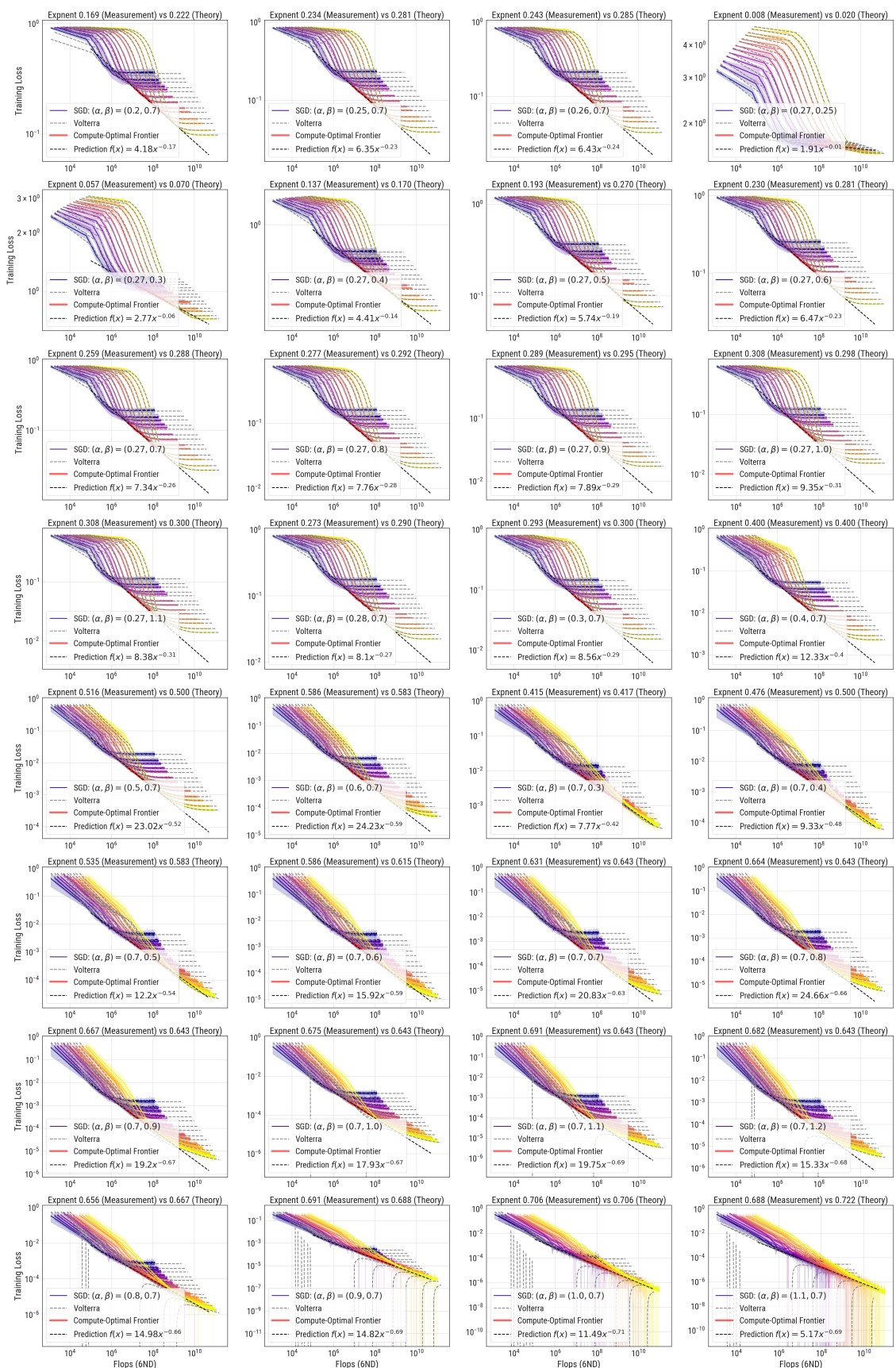

Figure 15: Theory vs. empirical scaling law across different phases.

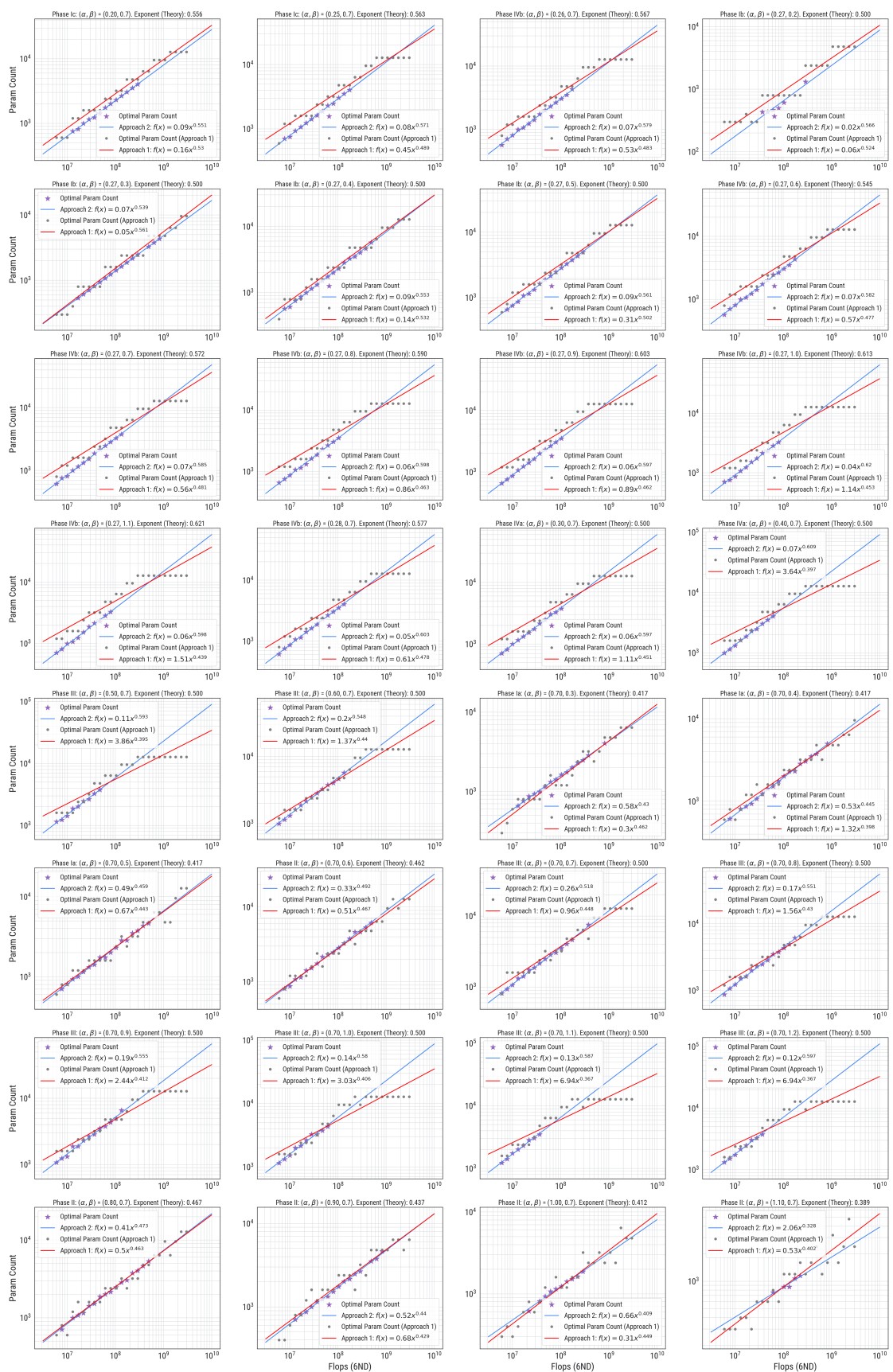

Figure 16: Theory vs. empirical optimal parameter count across different phases.

