# OpenReview forum: "4+3 Phases of Compute-Optimal Neural Scaling Laws"
_NeurIPS.cc/2024/Conference — NeurIPS 2024 spotlight_

### Official Review · Reviewer_9ixz · 2024-07-11

**Soundness:** 4
**Presentation:** 3
**Contribution:** 3
**Rating:** 7
**Confidence:** 3

**Summary:**

The authors consider a simple scaling model and derive scaling laws for one pass SGD in an asymptotic limit. From the scaling laws they identify several phases and subphases where certain components of the loss dominate and the compute-optimal model parameter count is affected. The loss components are related to the model capacity, SGD noise, and feature embedding. Detailed calculations are performed to completely characterize the behaviors in different regimes and are accompanied by simulations to give supporting evidence. Interestingly large regions of the phase space result in a "universal scaling" independent of the problem parameters.

**Strengths:**

Understanding scaling behavior of learning models is of great theoretical and practical importance. This work provides a very thorough and comprehensive analysis of the simplest, non-trivial model. Despite the simplicity of the model a rich variety of behavior is observed which requires effort to analyze and catalogue. The observations of universality and effects of finite size are interesting and potentially relevant to more realistic settings.

**Weaknesses:**

1. It is unclear how novel the mathematical techniques are and how general purpose they are, some discussion would be helpful.
2. It is unclear how related the simple model is to observations about more realistic models so some more commentary could be useful.

**Questions:**

1. Is there potential for a more general explanation for universality since the same scaling appears to hold in the Chinchilla scaling law?
2. Can any of the techniques be extended to analyze more complex models?
3. Are there any lessons for practical LLM scaling?

**Limitations:**

Yes.

---

> ### Author Rebuttal · Authors · 2024-08-06
>
> We thank the reviewer for their insightful comments and suggestions which were very helpful. We address these below.
>
> 1. ***Novelty of Techniques.***
>    * *Analysis of learning rates via Volterra equations in concert with random matrix theory has appeared before (say \[10\]/\[11\] in the paper).*  Volterra analysis is not widespread, and it remains to be seen how general purpose they are.  For precise risk curve analysis of stochastic algorithms, they seem useful, including momentum methods \[1\].  A more general point of view is needed for analysis of non-quadratic problems, but there is an equivalent system of ODEs reminiscent of mean field theory (this is equivalent to the Volterra description in the least-squares case) which has been used as well \[2\].  Generalizing the type of results in this paper to a non-linear setting would require understanding how these systems of ODEs evolve without appealing to convolution-Volterra-theory; these look quite similar for the linear and nonlinear cases, so I would be cautiously optimistic that the risk curve analysis could be adapted to nonlinear PLRF
>
>
>      \[1\] Paquette et al. Dynamics of stochastic momentum methods. 2021
>      \[2\] Collins-Woodfin et al.  Hitting the high-dimensional notes: An ODE for SGD learning dynamics on GLMs and multi-index models. 2023\.
>
>
>    * Random matrix theory analyses of generalization errors in random features models are now pretty well-developed, and this paper certainly fits within that tradition. The majority of the technical work and mathematical novelty in this paper is the analysis of the PLRF resolvent, which is a pure random matrix theory question.  After that, there’s a fair bit of asymptotic analysis which is needed, which will probably be common to all analyses of power-law scaling law problems.
>
>
>
> 2. ***Regarding applications to realistic models.*** We’ve also been asking ourselves this question.  Probably the quantitative predictions of this model will be impossible to fit to anything like Chinchilla; one would need to have a way to estimate both $\\alpha$ and $\\beta$, which is approaching science fiction.
>
>    Now on the other hand, it is possible (as in Chinchilla) to measure the resulting scaling laws.  One may attempt to vary parameters that could influence  $(\\alpha,\\beta)$, and see if the scaling laws respond in the way one would expect from this paper.  One might also look for a phase plane of LLM parameters in which the optimal scaling $d^\* \= \\sqrt{f}$ with $f = $flops changes; we saw that this is possible in some parts of the $(\\alpha, \\beta)$ plane.
>
>    Another possible complexity is that adaptive and preconditioned methods are all but necessary for training LLMs, and so some effort should be made to establish to what extent the phase diagram we described is affected by different optimizers.
>
> **Responses to Questions:**
>
> 1. ***(Deeper pattern to where $d^\* \= \\sqrt{f}$).***  This is a great question and perhaps one of the biggest mysteries of the phase diagram, given that it appears in 3 distinct regimes.  I can share some possible speculations, but I don’t really know.
>    * One possible answer is that real-life is closest to $\\alpha=\\beta$ (or perhaps autoregressive problems are always somewhere close to $\\alpha=\\beta$), and along this line it just so happens $d^\*= \\sqrt{f}$.
>    * Potentially, regimes where $d^\* \= \\sqrt{f}$ are not optimal are a reflection of the algorithm ‘failing’ in some sense, and one should be looking to improve it. Conversely, existing work suggests that Phase Ia might be algorithmically unimprovable – see the discussion above.
> 2. ***(Extensions to more complex models).***  Of course the proof will be \`\`in the pudding,’’ but we would guess that the answer is yes – nonlinear power-law-random-features, anisotropic weight matrices, models with some level of feature learning in mean-field scaling.   Existing work on random features regression and 2-layer networks strongly suggests that the extension to the nonlinear case should be solvable.  The case of anisotropic weight matrices should be quite similar mathematically to this work (it mostly comes to mind as a way to better understand the effect of using preconditioned algorithms on scaling laws).
> 3. ***(Practical lessons for LLM scaling).***  We have a few possible lessons:
>    * We give some further evidence that $d^{\\star} \= f^{½}$ is justified.
>    * The ‘functional form’ of the scaling laws of the risk curves which is empirically fit to scaling laws should be updated to be $Risk(n,d) \= c \+ a\_1 n^{-b\_1} \+ a\_2 d^{-b\_2} \+ a\_3 d^{-b\_2} n^{-b\_3}.$.  The last term looks like $F\_{ac}$, and is usually dropped.  But it is needed even in this simple setting to get the correct scaling law behavior (in our setting $b\_2$ is always $1$).
>    * Finite-dimensional effects in size of parameters (d) exist and potentially have a huge impact on the compute optimal exponents. We see even in our simple model (see Fig. 4). If one is doing empirical scaling laws, then one can easily see measured exponents that change by something like $0.05$ as one increases the FLOP-budget by a factor of $10^9$.  In LLM scaling, this $0.05$ is the same order of magnitude as the observed scaling law itself, and it is not (per se) related to a breakdown of the scaling laws.
>    * Speculatively: with some hypotheses about which phase one is in, one may be able to determine what is the dominant feature of the loss curve ($F\_{ac}$ dominated, $F\_{pp}$ dominated, $K\_{pp}$ dominated).  This could allow one to determine if in future training runs, how to increase parameter counts or compute.  (Roughly $F\_{ac}$ is expensive from a compute point of view, and it is cost efficient to increase scale – or reserve compute for other parts of the model when one enters this regime).   Of course, this heuristic is just a hypothesis, based on the work here and needs to be tested.

---

> > ### Comment · Reviewer_9ixz · 2024-08-13
> >
> > Thank you for the detailed responses. I will maintain my score.

---

### Official Review · Reviewer_rEuG · 2024-07-12

**Soundness:** 2
**Presentation:** 1
**Contribution:** 2
**Rating:** 6
**Confidence:** 2

**Summary:**

This submission studies the Power Law Random Feature (PLRF) that depends on three parameters: data complexity, target complexity, and model parameter count. They derive a deterministic closed expression for the dynamics of SGD using a clever mapping to Volterra equations. They are able to determine the compute-optimal frontier for different learning scenarios. The theoretical claims are supported by extensive numerical simulations.

**Strengths:**

The main strength of this paper stands clearly in being both theoretically and numerically exhaustive. The numerical illustrations and details backing up the theoretical claims are explored in detail both in the main and appendices. Moreover, the questions addressed are of great interest to the theoretical machine learning community.

**Weaknesses:**

The strong weakness of this submission is the presentation. It is challenging for a non-expert reader to navigate the related literature with only thirtheen references and eleven lines devoted to the related works.

**Questions:**

As mentioned above, a large part of my concerns resides in the presentation of the results and the framing of the present submission in terms of the related literature. See below for some examples of needed citations/explanations.


- Deterministic equivalent at page 2, what is it and how is it used in machine learning theory? Cite reference for this, e.g., [1].
- What is a "neural scaling model". This appears in the first line of the abstract.
- When introducing PLRF there is no mention to what classical RF looks like [2]. Many works could be cited that use also Random Matrix Theory tools to connect at the previously described deterministic equivalent, e.g. [3] among many others.
-  [4,5] are two papers that drew phase diagrams for the training dynamics of SGD. Although they do not seem related to the present submission, they deal with the optimal tuning of SGD from a theoretical point of view.

The setting presented in the manuscript has many limitations-- which is completely acceptable for a primarily theoretical work. However, the technical challenges that the authors would face if they were to lift them are never discussed. For example, square loss, lack of non-linearity in the PLRF, deterministic teacher vector $b$, need for $v>d$, etc. All these assumptions are reasonable, but must be compared to related works on the subject and why it is difficult to lift them.

### Minor points
- Explain in deeper detail footnote number 6. Why is this the case?
- The mapping to Volterra equations is nice and I believe it would deserve more space in the main body.
- Different subfigures might have different aspect ratios (see e.g. Figures 2 and 3).
### References

- [1] Random Matrix Methods for Machine Learning. Coulliet & Liao 2022, Cambridge University Press.
- [2] Random Features for Large-Scale Kernel Machines. Rahimi & Recht. NeurIPS 2007.
- [3] The generalization error of random features regression: Precise asymptotics and double descent curve. Mei & Montanari. Communications on Pure and Applied Mathematics 2021.
- [4] Online Learning and Information Exponents: On The Importance of Batch size, and Time/Complexity Tradeoffs. Arnaboldi et al. ICML 2024.
- [5] Dissecting the effects of SGD noise in distinct regimes of deep learning. Sclocchi et al. ICML 2023.

**Limitations:**

The suggestions for improving the description of the theoretical limitations are given above.

---

> ### Author Rebuttal · Authors · 2024-08-06
>
> **Responses to Weaknesses and questions:**  Thank you for your comments.  We will definitely add a more thorough evaluation of related works.  Great catch about Footnote number 6 – it’s actually not true – we’ve removed it.  We’d be very happy to hear any other comments about the content of the paper – time allowing, we will add some discussion in the appendix about batch size, which was a not-fully-explored thread of this paper.
>
> Regarding the questions (square loss, lack of non-linearity in PLRF, deterministic teacher vector $b$, need for $v \> d$). \`\`All these assumptions are reasonable, but must be compared to related works on the subject and why it is difficult to lift them.’’
>
> 1. Extending to square loss and adding a nonlinearity are both very interesting directions of future research, and it would certainly be reasonable to do them after considering the square loss – linear case\!
> 2. The deterministic teacher vector could be replaced by a random one with neither an increase in complexity, but also no change in phenomenology (if it has the same behavior as $b$).  Perhaps we can add: the goal here isn’t really theory-building, in the sense of covering as wide a class of kernel regression problems as possible.  The goal is to map out as much phenomenology as possible.  So generalizations of the problem-setup which are not expected to change this phenomenology were not prioritized.
> 3. The case of $v\>d$ is for simplicity and also because we broadly believe it aligns with what one sees in neural scaling laws.  Indeed you can consider another phase when ($2\\alpha \+ 2\\beta \< 1$) with $v \< d$, but we suspect it is another ‘Phase I’.  The same comment goes for negative $\\beta$.
>
> We agree with the reviewer that more references and a more thorough discussion of background and related work would improve the accessibility of this paper. For that reason we have included 32 references and proposed changes to the related work and background sections (see the general response). In light of these changes, we hope the reviewer will reconsider their score.

---

> > ### Comment · Reviewer_rEuG · 2024-08-10
> > **Thank you for your rebuttal**
> >
> > I thank the authors for their rebuttal. I carefully read it along with the other reviewers’ comments and I would like to increase my score to 6. As I mentioned in my first review, the outstanding weakness of this paper was the presentation, and I believe the proposed changes will greatly improve the quality of the submission.

---

> ### Comment · Area_Chair_vGNS · 2024-08-10
>
> Hi reviewer, in the response you wrote that you would like to increase their score from a 4$\rightarrow$6. However seems the system not actually change their score. If you want to increase your score, may you implement this to the openreview system?
>
> Thanks.
>
> Your AC

---

### Official Review · Reviewer_wrFd · 2024-07-12

**Soundness:** 4
**Presentation:** 2
**Contribution:** 3
**Rating:** 6
**Confidence:** 4

**Summary:**

The paper studies a linear random feature model trained on power law data under online SGD. By using a Volterra approach and leveraging deterministic equivalence, they characterize the loss in the high dimensional limit. From this, they extract scaling laws for this model, that determine compute optimality. The show that depending on the spectral decay of the covariates and the task vector, SGD noise as well as finite width effects can effect the scaling of the loss curves with time. A consequence of this is that the optimal allocation of compute can change. By systematically studying finite width and time effects, the authors characterize the scaling laws one should expect in this model.

**Strengths:**

The paper investigates an important problem. The reviewer has checked much of the math in the (extensive) appendix. The derivations are technically sound. The conclusions of this paper certainly add meaningfully to the present machine learning literature and our understanding of the role of the optimizer on neural scaling laws. The experiments are beautifully done and add meaningfully to the presentation of the paper. I commend the authors on the care put into the experiments and encourage them to make the code public.

**Weaknesses:**

There are two primary weakness:

1. The paucity of citations in this work is stunning

Many authors have studied random feature models before and concurrently with the Maloney paper. The set of random feature model papers that deserve a citation are many, and I leave it to the authors to do a thorough review. Certainly Mei and Montanari https://arxiv.org/abs/1908.05355, as well as Bach's paper that also derives the same scalings as Maloney: https://epubs.siam.org/doi/abs/10.1137/23M1558781
Before all of this, Hastie et al in their original paper studied random projections on structured data as well:
https://arxiv.org/abs/1903.08560
Although not explicitly calculated, the results of Mel and Pennington give as a special case the generalization error formula in Maloney et al  and was published much earlier. They just didn't calculate the scaling law exponents.
https://openreview.net/forum?id=JfaWawZ8BmX
Similar models were studied and exactly solved by Loureiro and colleagues in several papers, for example
https://arxiv.org/abs/2102.08127

Moreover, although I have not read all of the Bordelon paper due to lack of familiarity with DMFT, I can see that they explicitly treat the case of SGD noise. Given the substantial overlap in model and problem studied I think it is worth clarifying sharply what this paper puts forward beyond the initial studies of that one.

I think this work is important. Putting it in the context of works that came before is also important. The authors are hurting the reception of their paper in the broader community by citing so sparsely.

2. Although the readability of this paper in the main text is reasonable, the accessibility of the appendix is quite poor.

The first section deriving the Volterra equation is written quite clearly and accessibly. From there forward, it becomes increasingly impossible to read.

By hopping around the theorems and propositions one can eventually recover your bounds on the forcing function and the kernel function. I strongly recommend restructuring the order of your theorems and propositions. I also strongly encourage an "overview of proofs" to start the appendix off so that a reader can navigate the dense math more carefully.

The contour presented in Figure 7 comes out of nowhere. Even after reading most of this paper several times over I still have no idea how the authors arrived at it. I've managed to reproduce most of the results and I don't need this type of analytic contour argument at all.


If these two concerns can be addressed, I will be happy to raise my score. Specifically, I would like the authors to list all of the relevant papers that they will cite in the revised draft, and I would strongly encourage them to follow through.

**Questions:**

1. It is not obvious without going deep into the appendix why these terms are called $K_{pp}$, $F_{ac}$, etc. It would be much better if you explained this sooner rather than later.

2. If I am not mistaken, the limit that isn't high-dimensional, $2 \alpha < 1$, would never be encountered for a kernel trained on natural data.  Although there do indeed seem to be interesting phenomena below that line, I am wondering why the authors have decided to so carefully study this? Is is purely out of a desire to characterize the entire $\alpha, \beta$ plane?

3. Further, $\beta > 1$ corresponds to tasks that are ``in the Hilbert space'' of the original $v$ dimensional space. Again, this is never the case for natural data, where the spectral decay is much much slower. If anything this paper seems to tell us that SGD has very little effect on the scaling laws on realistic data.  Please let me know if I am incorrect in this characterization. Otherwise, given the discussion of scaling laws, there will likely be some practitioners that read this paper who are confused about how to interpret $\alpha$ and $\beta$. I think the authors would do well to state clearly the relationship between the regions in the $\alpha$ and $\beta$ plane and the values one would expect in real datasets.

4. Why are you using the term `SGD frustrated'? In the statistical physics literature this is taken to mean something else entirely.

5. As a smaller comment, the notation for the exponents is quite different from other works. In linear regression its standard to report things in terms of the "source and capacity" exponents, where $\alpha$ is the decay exponent of the spectrum. Of course the authors don't need to change their notation but a table comparing notations with other works would be very useful and go a long way.

**Limitations:**

Given that this is a toy model of neural scaling laws, I do not expect major societal implications. However, the principled understanding of such scaling properties may well have important impacts on the future of the field.

---

> ### Author Rebuttal · Authors · 2024-08-06
>
> We thank the reviewer for their insightful comments and suggestions. We address below questions and concerns raised by the reviewer. Because there was a lot of depth in the questions raised, we needed additional characters to respond adequately so below is the first part of the response. *There will be an additional “Official Comment” with the remaining response.*
>
> **Responses to Weaknesses:**
>
> 1. *(Related Work)* We are very happy to add discussion regarding random feature generalization (especially the relations to generalization error bounds which were developed for source/capacity conditions) and SGD complexity on power-law problems, which are absolutely related to this work. **See response to the ‘General Reviewers’ and the attached pdf** for comparison of our notation to source/capacity conditions.
>
> 2. *(Accessibility)*
>    * We will add an introduction to the appendix to guide how the proofs are done.
>    * The contour is \`what works.’  We do not believe the contour is absolutely necessary to proving the result; it is a nice technical tool, but not essential.   Indeed – we heuristically derived all the arguments long before we found the contour in the process of formulating the proof.
>
>      The origin of the contour is solving the fixed point equation for $m$.  The ‘kink’ in the contour occurs at real-part $x \\approx d^{-\\alpha}$. This corresponds to a point of transition in the spectrum of the PLRF matrix: for smaller scales, the self-consistent spectrum forms a spectral bulk.  For larger scales, the self-consistent spectra forms outliers, roughly located at $j^{-2\\alpha}$ for integer $j$. In the self-consistent spectra, these outliers actually have densities of width approximately $O(1/sqrt(d))$.
>
>      Now the contour is basically chosen as the closest contour to this axis that makes this transition invisible, which means the imaginary part of the contour is just a bit larger than the inter-eigenvalue spacing, and it means we see a clean power-law everywhere along the contour.
>
>      One *could* instead not put the kink, but then would have an absolutely continuous part and an \`\`approximately’’ pure point part of the spectrum, which is more annoying than anything (mostly because the m function undergoes some relatively high-frequency changes near each outlier, and all this high-frequency excitement turns out to be completely invisible and unimportant to the larger picture).  The method we use to approximate the m is fundamentally perturbative, so matching all the high-frequency changes in m is super-hard; so it’s easier to change the contour and then have a good, smooth ansatz for m.
>
> 3. *(Comparison with Bordelon et al. \[1\])*
>    We highlight below the main differences between our work and Bordelon et. al \[1\]:
> * Bordelon et. al assumes the functional form for the loss is $n^{-\\tau} \+ d^{-\\sigma}$ (see Appendix N, Eq. (153)). Then they use DMFT to find the exponents $\\tau$ and $\\sigma$. *We prove that the functional form for the loss of power-law random features is*
>
>   $$ P(n,d) \\asymp \\underbrace{n^{-\sigma\_1}}\_{F\_{pp}} \+ \\underbrace{n^{-\sigma\_2}}\_{K\_{pp}} \+ \\underbrace{d^{-\tau\_1}}\_{F\_0} \+ \\underbrace{ d^{-\tau\_2} n^{-\sigma\_3}}\_{F\_{ac}} $$
>
>   Note the cross-term $\\underbrace{ d^{-\tau\_2} n^{-\sigma\_3}}\_{F\_{ac}}$ which is missing from the functional form for Bordelon and plays an important role in Phase II and III for the compute-optimal curves. Additionally, Bordelon et. al. do not consider phase III, and  the impact of the SGD algorithm through the term $K\_{pp}$.
>
> * Our work holds for any $\\alpha$ and $\\beta$ and  Bordelon et. al. only work for the trace class setting, i.e., $2 \\alpha \> 1$. Moreover Bordelon et. al. compute-optimal result agrees with our compute-optimal exponents when $2\\beta \< 1$. This is consistent since the loss curves in Phase Ia only depend on $F\_{pp}$ and $F\_0$. The cross term $F\_{ac}$ and the impact of SGD $K\_{pp}$ do not appear until Phase II-IV.
>
>   Comparison of $(a,b)$ and $(\\alpha, \\beta)$ between the two papers
>
>
>
>   ----------------------------------
>
>         Bordelon, et. al. |   This paper
>   ----------------------------------
>
>
>             b             |    $2\alpha$
>
>             a             |    $2\beta + 2\alpha$
>
>
>   We will add a comparison to the text (see also Table 1 in attached pdf of revision).
>
> \[1\] B. Bordelon, A. Atanasov, and C. Pehlevan. *A Dynamical Model of Neural Scaling Laws.* arXiv preprint arXiv:2402.01092, 2024\.

---

> > ### Author Response · Authors · 2024-08-07
> > **(Response to questions)**
> >
> > **Response to Reviewer Questions**
> >
> > 1. **(Moving up definition of $K\_{pp}$ and $F\_{ac}$):** We agree with the reviewer and we will move the explanations of $K\_{pp}$ and $F\_{ac}$ earlier in the text.  Actually, we didn’t explain the names because the notation was grandfathered in from earlier parts of the project (which featured such abominations as $K\_{ac}$  and $F\_{bulk}$ and which all turned out to be irrelevant), and this notation turned out to not be well-aligned with what the terms mean.   We plan to change the names.
> >
> > 2. **(Below the line $2 \\alpha \< 1$ and $\\beta \> 1$):**  We do not view this work as motivated by kernel regression, which perhaps explains the philosophical divide. The random features model here with $d$ parameters is a toy model of non-linear optimization problems with $d$ parameters, e.g. LLM training problems of growing parameter count.  We can imagine approximating these optimization problems by kernel problems, which would lead to a **sequence** of kernels, one for each $d$. To make a nice limit theory these kernels would need to converge weakly – but they certainly don’t have to converge to a trace-class operator.  So any eigenvalue decay/growth rate could be realized this way.  For example, if the NTKs performed some type of ‘whitening’ operation on some underlying data, one would expect long stretches of eigenvalues with very slow (or almost no) decay.
> >
> >    **(Which $\\alpha, \\beta$ are important:)** By the same token, even with a fixed target, if you have a sequence of kernels you could have basically anything for the target decay rate.  As for $\\beta$ in various phases being more “real” than others – I’m curious to know if you have any in-depth study.  My guess is that $\\beta=\\alpha$ is actually super common, especially in autoregressive tasks or in classification tasks which are well–approximated (spectrally) to the nearest-neighbor function. The exponents we see in LLMs scalings \[1\] are really small so this would suggest that there is some practical merit to considering what happens below the high-dimensional line.
> >
> >    For LLMs scaling, which is the motivation of this work, estimating $\\alpha, \\beta$ is hard.
> >
> > \[1\] Hoffmann et. al. Training Compute-Optimal Large Language Models. 2022
> >
> > 3. **(SGD frustrated)**: We are not married to this terminology and can change it. In Phase III, the optimization trajectory is slowed primarily due to the rate of SGD noise production, which is sufficiently fast that the underlying problem is solved faster than SGD can solve for its own mistakes.  In other words, geometrically the gradients produced by SGD are sufficiently randomly orientated that they themselves are the slowdown to the algorithm (and this overwhelms the difficulty of finding the signal).  So this looks a little like frustration, even if the usage is not the same as in spin systems. But to avoid confusion we will adopt other terminology, perhaps ‘SGD limited’.
> >
> > 4. **(source/capacity):** We have provided a table with the comparisons in the attached pdf to “All Reviewers”.

---

> ### Comment · Reviewer_wrFd · 2024-08-10
> **Response to Reviewer Comments**
>
> I thank the reviewers for compiling a more proper bibliography for this quality of work.
>
> I also thank them for explaining in detail the differences between this work and the prior works, especially of Bordelon et al. I looked at the supplementary note, comparing notations and *especially* Table 3 comparing this work with others. I think this is a very nice table, that will be important and useful for the community, and I encourage them to either start the appendix with this table or to even consider putting it in the main text. I definitely hope that the related work section on page two can be expanded beyond just five sentences to highlight some these facts and point the reader to the table. It makes it much more clear what this paper contributes over prior work.
>
> Lastly, given that the contour is indeed just "what works" I strongly encourage the authors to state that clearly in the appendix. I found the appendix relatively readable until this contour came out of nowhere. Even just an idea of what motivated the authors to consider it would go a long way.
>
> As promised, I have raised my score.

---

> > ### Comment · Reviewer_wrFd · 2024-08-10
> > **Thoughts on Studies of the High Dimensional Line**
> >
> > The authors raise an interesting question in point 2. To my knowledge, extensive study of $\alpha$ and $\beta$ has not been performed but it certainly can be done so in principle. In Fig 12 of the Bordelon paper I see that they measure $\alpha$ for image datasets for the NTK of a resnet and find throughout the course of training that the effective $\alpha$ remains above the high dimensional line. The other paper I am familiar with is one of Steinhardt and Wei:
> >
> > https://arxiv.org/pdf/2203.06176
> >
> > see especially Fig 5 and Table 3.
> >
> > There, for a variety of architectures on basic vision tasks they find that both the initial and final kernels have spectral decay that keeps the kernel trace class. I have a strong belief that this will remain true for virtually all vision datasets, even at much higher resolutions.
> >
> > For text data I do not have strong intuition, but would believe as with images that $\alpha$ remains above $1/2$ and only $\beta$ is small. Our differing intuitions are indeed interesting and I would be very excited to see an empirical paper study this!

---

### Official Review · Reviewer_ahLt · 2024-07-25

**Soundness:** 3
**Presentation:** 3
**Contribution:** 3
**Rating:** 7
**Confidence:** 3

**Summary:**

This submission studies the generalization error dynamics of one-pass SGD in a sketched linear regression setting, where the data and target signal are distributed according to certain power laws, and SGD optimizes a linear model on top of a Gaussian random projection. Using random matrix theoretical tools, the authors precisely computed the asymptotic generalization error of the SGD iterates in high dimensions; this reveals various scaling laws and allows the authors to characterize the compute (flop) optimal frontier.

**Strengths:**

This is an emergency review, so I shall simply state that this is probably the most interesting submission in my batch and should be accepted.

**Weaknesses:**

My main concern is that the authors do not adequately discuss the overlap with prior results.

1. Generalization error of the random features model under various source and capacity conditions has been extensively studied. For instance, (Rudi and Rosasco 2016) derived scaling laws of random features regression with respect to the model width and number of training data, and this result is later extended to SGD in (Carratino et al. 2018), taking into account the scaling of iteration number. The authors should explain the similarity and differences in the findings to highlight the advantage of a precise analysis.
* (Rudi and Rosasco 2016) *Generalization Properties of Learning with Random Features.*
* (Carratino et al. 2018) *Learning with SGD and Random Features.*

2. If we do not take optimization into account, then neural scaling laws in the form of $(\text{data}^{-\beta} + \text{param}^{-\gamma})$ have been established in many prior works on nonparametric regression (Schmidt-Hieber 2017) (Suzuki 2018) -- these are rigorous versions of the hand-wavy arguments in (Bahri et al. 2021) which the authors cited.
If we interpret the number of training data to be the same as the iteration number, can we obtain similar plots on the compute-optimal front as reported in Figure 2(a)(b)?
* (Schmidt-Hieber 2017) *Nonparametric regression using deep neural networks with ReLU activation function.*
* (Suzuki 2018) *Adaptivity of deep ReLU network for learning in Besov and mixed smooth Besov spaces: optimal rate and curse of dimensionality.*

Some additional questions:

1. In light of empirical findings that training small models for a longer time can be beneficial, can the authors comment on the possibility of extending this analysis to the multiple-pass setting? In the kernel regression literature, it is known that multi-pass SGD has statistical superiority (Pillaud-Vivien et al. 2018).
* (Pillaud-Vivien et al. 2018) *Statistical Optimality of Stochastic Gradient Descent on Hard Learning Problems through Multiple Passes.*

2. Does the sample size scaling match the minimax optimal rate in (Caponnetto and de Vito 2007) in certain regimes?
* (Caponnetto and de Vito 2007) *Optimal rates for regularized least-squares algorithm*.

3. What are the technical challenges to show the asymptotic equivalence between the SGD dynamics and the deterministic equivalent description that the authors analyze?

4. Can the authors comment on the restriction to proportional $v,d$? Why is there a sharp transition at $2\alpha=1$ that decides the scaling of $d$?

**Questions:**

See Weaknesses.

**Limitations:**

See Weaknesses.

---

> ### Author Rebuttal · Authors · 2024-08-06
>
> We thank the reviewer for their comments which were very helpful. We address these below. Because there was a lot of depth in the questions raised, we needed additional characters to respond adequately so below is the first part of the response. *There will be an additional “Official Comment” with the remaining response.*
>
> **Response to weakness:** Yes we agree, the comparison to existing work should be expanded.  We will include a more detailed discussion of existing work and how it compares.  We will also include a section detailing how these SGD-generalization bounds compare to optimized non-algorithmic bounds (especially to optimal ridge regression).  The suggestions for references are super helpful – thank you – and we have made an attempt to integrate them.   We’ll outline the results of our attempt to answer your question in the points below.  We’re happy to have feedback if you believe the works considered are still inappropriate.
>
> Regarding the specific suggestions, thanks, many are new to us.  We will formulate our response partly based upon the concurrent article \[1\], which features a sharp analysis of ridge regression on a problem with the capacity/source conditions and which improves over Rudi-Rosasco.  Note the source/capacity conditions are not meaningful for $2\\alpha \< 1$, so we restrict the discussion to phases Ia/II/III ($2\\alpha \> 1$).
>
> Directly addressing the points you have made below:
>
> 1. ***Ridge regression comparison.*** The bounds for ridge regression agree with SGD in phases Ia/II, which is to say that if one takes ridge regression with r-samples, computes the generalization error with the ridge estimator with a O(1) ridge parameter; then the SGD risk curve we compute agrees with the generalization error and the ridge error agree up to constants.  (Using Corollary 4.1 of \[1\]).
>
> To match notation with the source/capacity literature, we'll use our $\alpha$ parameter (so the capacity is $2\alpha$) and we'll use the source parameter $r = (\\alpha+\\beta-0.5)/(2\\alpha)$; see also Table 1 in the pdf attached to OpenReview.   We note that phases Ia/II/III correspond to previously observed hardness regimes ($r \\in (0,0.5), (0.5,1.0), (1.0, \\infty)$)
>
>    A few comments about this: from \[1\], one recovers that the ridge-generalization-error equals the SGD-generalization-error in Phase 1a/II, at all points in the risk curve.  In Phase III, ridge regularization with O(1) ridge has better generalization error than SGD.  The comparison to the Rudi-Rosasco bounds can be seen in (48) of \[1\].  They match the generalization errors in Phase Ia, but are not optimal in Phase II/Phase III.
>
>    The Carratino et al. bounds (Corollary 1 of that paper) match the minimax-optimal rates attained by kernel ridge regression, and they effectively show that there are parameter choices for one-pass SGD which match those rates.  We discuss below minimax optimal rates.
>
> 2. ***Regarding non-parametric regression***: thanks for these references.  Bahri et al. and Maloney et al are indeed not mathematics papers.  We still believe they are quite valuable scientifically to this paper.  We’re happy to clarify how we view their contributions.
> * Regarding your main question Figure 2, it is **not possible to get this figure by identifying steps with samples.**  It’s super important to plot the x-axis of the curve in FLOPs, or there is no compute-optimal frontier (i.e. each iteration has a cost which is d-dependent).  So without choosing an algorithm which attains the regression bounds in the papers you list and assigning them a computational cost, the problem is not well-posed.
>
>
>   Now if we are talking about ridge regression with $O(1)$ ridge, then we could assign a computational cost of ($n \\times d$), for example using conjugate gradient (CG) (up to log factors).  One could also do this by considering vanishing ridge parameter, but then (since the condition numbers of these matrices grow like $d^{2\\alpha}$), the computational cost of computing using CG will increase to $n \\times d \\times (\\min\\{ d^{\\alpha}, \\sqrt{1/ridge}\\})$.
>
>
>   Let’s suppose we continue with ridge parameter $O(1)$, which is morally comparable to one-pass SGD, in that they use the same amount of compute – it’s also comparable in the sense that a non-optimized algorithm like gradient descent on the ell2-regularized-least-square objective will be similar in complexity (and vanilla SGD is surely a non-optimized algorithm).
>
>   In Phase Ia/II, the optimal ridge estimates in \[1\] (which agree with Rudi-Rosasco in Phase 1a but not in II) would indeed yield the same curves, treating computational complexity as $n \times d$ (with $n$ the number of samples and $d$ the dimension).  In Phase Ia, you get the same loss curves as SGD with O(1) ridge regularization, and so you get the same compute-optimal frontiers for CG+O(1)-ridge as SGD.  Incidentally, here you gain nothing by increasing the amount of ridge regularization.
>
>
>   In Phase III, CG+O(1)-ridge performs better than SGD for small sample counts.  However, if you run SGD to the stopping criteria for which it is compute optimal, it performs the same as CG+O(1)-ridge *with the same choices of parameters/sample counts*.  Now, on the other hand, it could be (and should be) that using CG+O(1)-ridge there is a different compute-optimal frontier curve.
>
>
>   We’re very happy to add the details (and some of the computations) of the comparison of SGD and ridge regression in an appendix, showing how these curves agree/do not agree.  We will also improve the discussion surrounding how compute plays a role, and link it to kernel regression literature.
>
>
> \[1\] Defillips, Loureiro, Misiakiewicz. Dimension-free deterministic equivalents for random feature regression.  arXiv: May, 2024

---

> > ### Author Response · Authors · 2024-08-07
> > **(Response to reviewer questions)**
> >
> > **Response to Reviewer Questions:**
> >
> > 1. ***(Extension to multipass.)*** Yes this is a super interesting question. We believe lots of the technology is in place, but there will also need to be a substantial mathematical effort, as the theory needs to be pushed.
> >
> >
> > Regarding the Pillaud-Vivien et al. reference, they identify two phases: “easy”, “hard”.  Hard is further divided into “improved” and “multipass optimal”.  The phase boundary of easy/hard is $\beta \= 0$ (so all of phases Ia/II/III are easy).  And if my reading of Pillaud-Vivien et al is correct, it should be anticipated that multipass SGD does nothing across the whole phase plane.  On the other hand (if I understand correctly) \`\`easy’’ is made in comparison to the Rosasco-Rudi upper bound (showing SGD attains it), and this upper bound is not optimal for our problem.  We have some simulations that suggest multipass SGD improves the sample complexity in Phases II/III.
> >
> >    The paper \[2\] shows that it is possible to derive risk curves in the form of convolutional Volterra equations, much like we have here.   \[2\] only proves them below the high-dimensional line ($2\\alpha \< 1$), but we expect that they generalize to the case $2\\alpha \> 1$ up to errors that should not affect the scaling laws (this is the case for the one-pass case).
> >
> >    The random matrix theory requires an extension as well, as one needs to study Gram matrices built atop samples of the random vectors we use in the paper under review.  This is another application of the Marchenko-Pastur map and so it requires another analysis.  It could introduce additional technical complexities and/or qualitative phenomena which need to be handled, but the path is certainly clear.
> >
> > 2. ***(Sample size complexity of Caponnetto and de Vito)***  No, the rates do not match minimax optimal rates (over problems classes with ‘source/capacity’ conditions).  The minimax optimal rates are $n^{-\\frac{2\\alpha+2\\beta-1}{2\\alpha+2\\beta}}$ for $r \\leq 1$  (which can be attained with small stepsize one-pass SGD by Dieuleveut and Bach; this is also used as a baseline by Pillaud-Vivien et al.).  In comparison the rates here are given by (Phases Ia/II) are $n^{-\\frac{2\\alpha+2\\beta-1}{2\\alpha}}$ in the $F\_{pp}$-dominated part of the loss curve.  This is no contradiction – we are studying a single problem with the source/capacity class, for which the performance can be better.  We further use a stepsize which is much larger than what is used in Dieuleveut-Bach. *See Table 2 & 3 in the attached pdf “All Reviewers” comment.*
> >
> > 3. ***(What are the technical challenges)*** There are three components of the proof, broadly.
> >    * The first is establishing a convolutional Volterra representation of the expected risk of the Gaussian streaming SGD.  This uses Gaussianity and is a relatively simple computation, but to our knowledge is new; previous Volterra equation analyses of SGD were done for non-Gaussian problems but required $2\\alpha \< 1$.  This is also relatively simple, and we do not really view this as a major contribution.
> >    * The second component is an approximation of the solution of the convolutional Volterra equation in which we keep 1 convolutional power of the bias and variance terms – similar approximations have been developed in the branching process literature and are expected in power-law risk curves.  This part is probably not new, but the precise formulation we needed was not readily available, so we proved the relevant approximations (pointing to related work).
> >    * The third part is the analysis of the self-consistent equation for the PLRF resolvent.  This part, to our knowledge, is absolutely new (if the referees know otherwise, please let us know).  This is also the vast majority of the technical challenges, in part because the PLRF spectrum exhibits a transition at $d^{-\\alpha}$ (from absolutely continuous to pure point) and in part because we need to know the spectrum at all scales to get the whole loss curve.  This part represents the largest component of the technical challenges.
> >
> > 4. ***(Proportional v/d).***  For the regime $2\\alpha \> 1$, it’s not important for $v/d$ to be linear, and indeed $v=\\infty$ works fine, consistent with all the kernel literature.  For $2\\alpha \< 1$, $v=\\infty$ is actually meaningless, and in fact there is a transition that occurs when $v \\gg d^{1/(2\\alpha)}$ (the whole problem setup becomes trivial as the target becomes orthogonal to the span of the model).  So for simplicity we fixed a scaling of how $v$ grows with $d$.
> >
> > With a more sophisticated analysis, one could change the problem setup, releasing $v$ to the meta-optimizer and then optimizing over $d$ with respect to $v$.  It would be very interesting if it turns out that $d^\* \\ll v$.
> >
> > \[2\] C. Paquette, E. Paquette, B. Adlam, J. Pennington.  Homogenization of SGD in high-dimensions: exact dynamics and generalization properties. arXiv 2022

---

> > > ### Comment · Reviewer_ahLt · 2024-08-10
> > >
> > > Thank you for the detailed response, which I find illuminating.
> > > One minor follow up: in question 3, what I intend to ask is what are the technical challenges to rigorously justify the remark on the deterministic equivalent (footnote 7).

---

> > > > ### Author Response · Authors · 2024-08-10
> > > > **Response question 3**
> > > >
> > > > >What are the technical challenges to show the asymptotic equivalence between the SGD dynamics and the deterministic equivalent description that the authors analyze?
> > > >
> > > > Sorry, we misunderstood.
> > > >
> > > > The text of footnote 7:
> > > > > There is good numerical evidence that the deterministic equivalent captures all interesting features of the
> > > > PLRF. There is a vast random matrix theory literature on making precise comparisons between resolvents and
> > > > their deterministic equivalents. It seems a custom analysis will be needed for this problem, given the relatively
> > > > high precision required, and we do not attempt to resolve this mathematically here.
> > > >
> > > > The difficult part is proving the random matrix deterministic equivalent (for $\hat{K}$ in (eq 8)) holds on the contour we use (Figure 7).  This contour is at `local' scale, for real part $x \ll d^{-\alpha}$.  So to prove the true random matrix resolvent matches the deterministic equivalent on this contour, we would need to prove the equivalent of a local law, like in [1].  Part of the work is done in this paper, and lots of the estimates one would need are here (good approximations for the 'm' function and the 'stability operator').
> > > >
> > > > But then one needs to do all the concentration inequalities, and one needs to make sure the errors are relative (since an additive $O(d^{-1/2})$ or $O(d^{-1})$ error term would kills you).
> > > >
> > > > In my opinion, local laws are pretty serious random matrix theory, and the total mathematical complexity required to do the local law would be well suited to a standalone RMT journal paper.
> > > >
> > > > [1] Knowles, Antti, and Jun Yin. "Anisotropic local laws for random matrices." Probability Theory and Related Fields 169 (2017): 257-352.

---

### Official Review · Reviewer_HQ3F · 2024-07-26

**Soundness:** 3
**Presentation:** 3
**Contribution:** 3
**Rating:** 7
**Confidence:** 3

**Summary:**

This paper studied a solvable neural scaling law model (the power-law random features, PLRF) that involve three parameters (data complexity: $\alpha$, target complexity: $\beta$ and the number of parameters: $d$). The PLRF model here is trained by applying the stochastic gradient descent (SGD) algorithm to the mean-squared loss. For different choices of the parameters $\alpha$ and $\beta$, the optimal number of parameters $d$ is solved by minimizing the expected loss. A corresponding phase diagram is drawn with respect to $(\alpha,\beta)$. An extensive set of numerical experiments is also conducted to justify the main theoretical results.

**Strengths:**

1. This is a technically solid paper with rigorous mathematical proof. For each of the transition boundaries in the phase diagram, an intuitive explanation is provided to help the readers understand the key idea behind.
2. An extensive set of numerical experiments are included to justify the theoretical results.

**Weaknesses:**

One possible drawback of the current version of this paper is that the list of references is incomplete. It seems to the reviewer that the discussion on related work is incomplete and ignores many recent and concurrent work on the theoretical aspects of neural scaling law. See for instance [1,2].

**Questions:**

Overall, the reviewer enjoyed reading both the proof and the experiments presented in the paper a lot. One possible broad question that the reviewer is interested in is how the theoretical results presented here can be linked to previous work on the learning of timescales in two-layer neural networks [3] (In general, the key question is also to address which features get learned at which stage).

**Limitations:**

It seems that the authors didn't perform experiments on large-scale language models (LLMs). One possible way to make the work more impactful is to do some empirical investigation on LLMs to see if certain parts of the theoretical results still hold when the model complexity drastically increases.

References:

[1] Jain, A., Montanari, A., & Sasoglu, E. (2024). Scaling laws for learning with real and surrogate data. arXiv preprint arXiv:2402.04376.

[2] Lin, L., Wu, J., Kakade, S. M., Bartlett, P. L., & Lee, J. D. (2024). Scaling Laws in Linear Regression: Compute, Parameters, and Data. arXiv preprint arXiv:2406.08466.

[3] Berthier, R., Montanari, A., & Zhou, K. (2023). Learning time-scales in two-layers neural networks. arXiv preprint arXiv:2303.00055.

---

> ### Author Rebuttal · Authors · 2024-08-06
>
> We thank the reviewer for their report, for their comments on directions of improvement, and for the questions.
>
> **Response to weakness:** Yes we agree, the comparison to existing work should be expanded.  We will include a more detailed discussion of existing work and how it compares (see comments to all reviewers).  The suggestions for references are helpful – thank you – and we have made an attempt to integrate them (see the proposed related works section).  We’re happy to have feedback if you believe the works considered are still inappropriate. We provided a comparison with the concurrent work \[2\] (see Table attached in pdf) which was released after NeurIPS submission.
>
> **Response to question (2-layer Neural Network):** With regards to the question, the reference \[3\] is very interesting.  It is the first time this author is reading it, but it’s very close to my interests and I will dig back into it after NeurIPS.
>
> In \[3\], if one takes the single-index model to have a suitable decay rate of its Hermite expansion (which I believe is power-law decay of its Hermite coefficients), the gradient flow loss curve would have power-law decay.  That seems like it should open the door to an analysis of one-pass SGD that exhibits similar phenomenology to what we’ve done.   I’m not bold enough to speculate how similar it would be to what we’ve done, because I don’t have a good picture of what the SGD-noise looks like in this setup.  If the SGD noise behaves similarly to the quadratic problem we study, then it could fit naturally as a line through our phase diagram parameterized by the exponent $\\beta$ (which would come from the decay rate of the target function).
>
> That would be a beautiful theorem\!
>
> \[2\] Lin, L., Wu, J., Kakade, S. M., Bartlett, P.L., & Lee, J.D. (2024) Scaling laws for learning with real and surrogate data. arXiv preprint arXiv:2402.04376.
>
> \[3\] Berthier, R., Montanari, A., & Zhou, K. (2023) Learning time-scales in two-layers neural networks. arXiv preprint arXiv:2303.00055.
>
> **Response to limitation:** A deeper empirical investigation of LLMs would also interest this author, who hopes it can be done.  This is certainly a limitation; also estimating the alpha (and worse) estimating beta are serious obstructions in using the results of this paper quantitatively to do optimal model-parameter decision making.  However, as a matter of better understanding the training and optimization of LLMS, we agree this is an important direction of investigation.

---

> > ### Comment · Reviewer_HQ3F · 2024-08-09
> > **Response to authors' rebuttal**
> >
> > Dear authors,
> >
> > Thank you so much for your response, which have addressed my questions. Just as pointed out by the other reviewers, the theoretical results are pretty interesting, but please include a separate section to compare the result and proof techniques presented in your paper with the missing references listed in your global response above. Also, it would be a good idea to include what you wrote in your rebuttal as part of the future work as well.
> >
> > Best regards,
> >
> > Reviewer HQ3F

---

### Author Rebuttal · Authors · 2024-08-06

We thank the reviewers for all the constructive comments and suggestions for comparison.  All the reviewers requested additional discussion of related work.

We will do the following.

1. Expand our discussion of related work and background in the main text and add a section in the Appendix. A draft of some of these additions is visible here on OpenReview.  *See the “Official Comment” below.*
2. Add an appendix with quantitative comparisons of the risk curves we derive and related risk curves from the random features literature *(Table 2/3 in attached pdf)*, including a table bridging the notation *(Table 1 attached in pdf)*.

We have responded to each of you with a discussion of the points that you have raised.

**Novelty/Comparison to existing work.**

* We emphasize that the motivation of the work is *not* kernel/random features regression, but rather explaining *compute scaling laws.* This is why we consider the full $\\alpha, \\beta$-plane.
* When this work is compared to sample-complexity bounds, it is important to also remember that the estimator has an algorithmic cost. This algorithmic compute cost is what we are capturing in this work. Some works on sample-complexity are not explicit about their algorithm and so some additional work is needed to produce compute scaling laws.  *In response to Reviewer ahLt*, we have done this comparison for ridge regression. We plan to add an appendix containing these comparisons. *See attached pdf*
* While it may not be true in general, for this work the compute scaling laws happen to be corollaries of precise risk curves as a function of samples.  Many of our risk curves are new; they cannot be derived from existing estimates on SGD.  We discuss these in the response to *Reviewer ahLt (in re Caponetto et al.),* the improved works-cited (see *new Related Work section in 'Official Comment to All Reviewers' below*), and **included** a table of comparisons in the PDF.
* From a mathematical point of view, this paper also contains new random matrix theory: we analyze the (long-established) self-consistent equations for the resolvent of the power-law-random-features covariance matrix.  This has to be done for spectral parameters (the argument of the resolvent) all throughout the spectrum; in contrast, ridge regression requires negative spectral parameters, and this leads to different challenges. *(See also discussion to Reviewer 9ixz)*

*We will add an additional "Official Comment" with the new proposed related work section.*

---

> ### Author Response · Authors · 2024-08-07
> **(Proposed related work)**
>
> *Related work section to be added to our article.*
>
> **Random features and random matrices**
>
> This paper uses random matrix theory to analyze a random features problem, which in statistical language would be the generalization error of the one-pass SGD estimator.  Random matrix theory has played an increasingly large role in machine learning (see for \[11\] for a modern introduction).
>
> The input we need for our random matrix analysis is for sample covariance matrices with power-law population covariance (i.e. linear random features). Of course, the analysis of sample covariance matrices precedes their usage in machine learning (see e.g. \[5\]), but to our knowledge, a detailed study of all parts of the spectrum of sample covariance matrices with power-law population covariances has not appeared before. The narrower study of ridge regression has been extensively investigated (see for e.g.\[4,9\]), and the concurrent work \[13\] provides a complete analysis of the ridge regression problem when $2\\alpha \> 1$. However, while (roughly speaking) ridge regression requires analysis of the resolvent $(A-z\\text{Id})^{-1}$ statistics for negative spectral parameter $z$ (which might be very close to $0$), the analysis in this work requires resolvent statistics for essentially all $z$.
>
> There is a larger theory of nonlinear random features regression, mostly in the case of isotropic random features.  Including nonlinearities in this model is a natural future direction; for isotropic random features with \\emph{proportional dimension asymptotics} this has been explored in works such as \[21\] and for some classes of anisotropic random features in \[22,31,32\] (we mention that lots of the complexity of the analysis of power-law random features arises from the analysis of the self-consistent equations \-- indeed the self-consistent equations we use date to \[28\], but the analysis of these equations may still be novel).  This strongly motivates non-proportional scalings (which would be inevitable in power-law random features with nonlinearities); in the isotropic case, the state of the art is \[18\].
>
> **Random features regression, \`\`source/capacity'' conditions, and SGD.**
>
> A large body of kernel regression and random features literature is formulated for \`\`source/capacity'' conditions, which are power-law type assumptions that contain the problem setup here, when $2 \\alpha \> 1$ (the low-dimensional regime). For convenience, we record the parameters
> $$
> \\alpha\_{\\text{source}} \= 2\\alpha
> \\quad\\text{and}\\quad
> r \= \\frac{2\\alpha \+ 2\\beta \- 1}{4\\alpha}.
> $$
> Here we have taken $r$ as the limit of those $r$'s for which the source/capacity conditions hold (see Table in attached pdf).  We note that in this language $r$ is often interpreted as 'hardness' (lower is harder), and that $r \\in (0,0.5)$, $r \\in (0.5,1.0)$ and $r \\in (1.0,\\infty)$ correspond to $3$ regimes of difficulty which have appeared previously (see the citations below); they are also precisely the 3 phases Ia, II, and III.
>
> The authors of \[26\] establish generalization bounds for random feature regression with power-law structures in $2\\alpha \> 1$ case.  These bounds were sharpened and extended in \[13\] (see also the earlier \[7\] which shows kernel ridge regression is \`minimax optimal' under various \`source-capacity conditions'); we give a comparison to these bounds in Appendix (..), but we note that the problem setup we have is not captured by \`minimax optimality' (in particular minimax optimality is worst-case behavior over a problem class, and our problem setup is not worst-case for the traditional source/capacity conditions)
>
> We note that this paper is fundamentally about computation, but the novel mathematical contributions could also be recast in terms of generalization bounds of one-pass SGD, some of which are new.  The work of \[8\] compares SGD to kernel ridge regression, showing that one-pass SGD can attain the same bounds as kernel ridge regression and hence is another minimax optimal method (again under \`source-capacity' conditions).  See also \[15\] which considers similar statements for SGD with iterate averaging and \[25\] for similar statements for multipass SGD; see also \[14, 27\] which also prove the single-batch versions of these.  These bounds attain the minimax-optimal rate, which are worse than the rates attained in this paper (see Appendix (..) for a comparison).

---

> > ### Author Response · Authors · 2024-08-07
> > **(Proposed related work, continued)**
> >
> > **Dynamical deterministic equivalents, Volterra equations and ODEs.**
> >
> > Using the deterministic equivalents for random matrix resolvents \[17\], we in turn derive deterministic equivalents for the risk curves of SGD.
> >
> > The method of analysis of the risk curves in this paper is by formulation of a convolutional Volterra equation \[24\].  This can be equivalently formulated as a system of coupled difference equations for weights of the SGD residual in the observed data covariance, which generalizes beyond the least-squares context \[10\]; in isotropic instances, this simplifies to a finite-dimensional family of ODES \[1\].  This can also be generalized to momentum SGD methods \[23\] and large batch SGD methods \[19\].  Convolution-Volterra are convenient tools, as they are well-studied parts of renewal theory \[2\] and branching process theory \[3\].
> >
> > Another method of analysis is dynamical mean field theory.  The closest existing work to this one in scientific motivations is \[6\], which uses this technique.  This formally can be considered as a type of Gaussian process approximation, but for a finite family of observables (\`\`order parameters'').  In instances of one-pass SGD (including in anisotropic cases), this is rigorously shown to hold in \[16\].  The analysis of the resulting self-consistent equations is nontrivial, and \[6\] does some of this analysis under simplifying assumptions on the structure of the solutions of these equations.
> >
> > Besides these works, there is a large theory around generalization error of SGD.  The work of \[29\] gives a direct analysis of risks of SGD under \`\`source/capacity'' type assumptions which formally capture the $F\_{pp}$ parts of the Phase Ia/II loss curves.  The risk bounds of \[30\] give non-asymptotic estimates which again reproduce tight estimates for the $F\_{pp}$ parts of the loss (note that to apply these bounds to this case, substantial random matrix theory needs to be worked out first); see also \[20\] where some of this is done.

---

> ### Author Response · Authors · 2024-08-07
> **(Proposed related work, references)**
>
> **References**
>
> \[1\] Luca Arnaboldi, Ludovic Stephan, Florent Krzakala, and Bruno Loureiro. From high-
> dimensional & mean-field dynamics to dimensionless odes: A unifying approach to sgd
> in two-layers networks. In The Thirty Sixth Annual Conference on Learning Theory,
> pages 1199–1227. PMLR, 2023\.
>
> \[2\] Søren Asmussen. Applied probability and queues. Springer, 2003\.
>
> \[3\] Krishna B Athreya, Peter E Ney. Branching processes. Courier Corporation,
> 2004\.
>
> \[4\] Francis Bach. High-dimensional analysis of double descent for linear regression with random
> projections. SIAM Journal on Mathematics of Data Science, 6(1):26–50, 2024\.
>
> \[5\] Zhidong Bai and Jack W Silverstein. Spectral analysis of large dimensional random matrices,
> volume 20\. Springer, 2010\.
>
> \[6\] Blake Bordelon, Alexander Atanasov, and Cengiz Pehlevan. A dynamical model of neural
> scaling laws. arXiv preprint arXiv:2402.01092, 2024\.
>
> \[7\] Andrea Caponnetto and Ernesto De Vito. Optimal rates for the regularized least-squares
> algorithm. Foundations of Computational Mathematics, 7:331–368, 2007\.
>
> \[8\] Luigi Carratino, Alessandro Rudi, and Lorenzo Rosasco. Learning with sgd and random
> features. Advances in neural information processing systems, 31, 2018\.
>
> \[9\] Chen Cheng and Andrea Montanari. Dimension free ridge regression. Arxiv 2022\.
>
> \[10\] Elizabeth Collins-Woodfin, Courtney Paquette, Elliot Paquette, and Inbar Seroussi. Hitting
> the high-dimensional notes: An ode for sgd learning dynamics on glms and multi-index models. Arxiv 2023\.
>
> \[11\] Romain Couillet and Zhenyu Liao. Random matrix methods for machine learning. Cambridge
> University Press, 2022\.
>
> \[12\] Hugo Cui, Bruno Loureiro, Florent Krzakala, and Lenka Zdeborov´a. Generalization error
> rates in kernel regression: The crossover from the noiseless to noisy regime. Advances in
> Neural Information Processing Systems, 2021\.
>
> \[13\] Leonardo Defilippis, Bruno Loureiro, and Theodor Misiakiewicz. Dimension-free deterministic equivalents for random feature regression. arXiv preprint arXiv:2405.15699, 2024\.
>
> \[14\] Ofer Dekel, Ran Gilad-Bachrach, Ohad Shamir, and Lin Xiao. Optimal distributed online
> prediction using mini-batches. Journal of Machine Learning Research, 13(1), 2012\.
>
> \[15\] Aymeric Dieuleveut and Francis Bach. Nonparametric stochastic approximation with large
> step-sizes. The Annals of Statistics, 44(4):1363 – 1399, 2016.
>
> \[16\] Cedric Gerbelot, Emanuele Troiani, Francesca Mignacco, Florent Krzakala, and Lenka Zdeborova. Rigorous dynamical mean-field theory for stochastic gradient descent methods.
> SIAM Journal on Mathematics of Data Science, 6(2):400–427, 2024\.
>
> \[17\] Walid Hachem, Philippe Loubaton, and Jamal Najim. Deterministic equivalents for certain
> functionals of large random matrices. Ann. Appl. Probab., 17(3):875–930, 2007\.
>
> \[18\] Hong Hu, Yue M. Lu, and Theodor Misiakiewicz. Asymptotics of random feature regression
> beyond the linear scaling regime, 2024. arXiv
>
> \[19\] Kiwon Lee, Andrew Cheng, Elliot Paquette, and Courtney Paquette. Trajectory of mini-
> batch momentum: batch size saturation and convergence in high dimensions. Advances in
> Neural Information Processing Systems, 35:36944–36957, 2022\.
>
> \[20\] Licong Lin, Jingfeng Wu, Sham M. Kakade, Peter L. Bartlett, and Jason D. Lee. Scaling
> laws in linear regression: Compute, parameters, and data. arXiv: 2024
>
> \[21\] Song Mei and Andrea Montanari. The generalization error of random features regression:
> Precise asymptotics and the double descent curve. Communications on Pure and Applied
> Mathematics, 75(4):667–766, 2022\.
>
> \[22\] Gabriel Mel and Jeffrey Pennington. Anisotropic random feature regression in high dimen-
> sions. In International Conference on Learning Representations, 2021\.
>
> \[23\] Courtney Paquette and Elliot Paquette. Dynamics of stochastic momentum methods on
> large-scale, quadratic models. Advances in Neural Information Processing Systems, 34:
> 9229–9240, 2021\.
>
> \[24\] Courtney Paquette, Kiwon Lee, Fabian Pedregosa, and Elliot Paquette. Sgd in the large:
> Average-case analysis, asymptotics, and stepsize criticality. In Conference on Learning
> Theory, pages 3548–3626. PMLR, 2021\.
>
> \[25\] Loucas Pillaud-Vivien, Alessandro Rudi, and Francis Bach. Statistical optimality of stochas-
> tic gradient descent on hard learning problems through multiple passes. Advances in Neural
> Information Processing Systems, 31, 2018\.

---

> > ### Author Response · Authors · 2024-08-07
> > **(Proposed related work, final)**
> >
> > \[26\] Alessandro Rudi and Lorenzo Rosasco. Generalization properties of learning with random
> > features. Advances in neural information processing systems, 30, 2017\.
> >
> > \[27\] Ohad Shamir and Tong Zhang. Stochastic gradient descent for non-smooth optimization:
> > Convergence results and optimal averaging schemes. In: International conference on machine learning, pages 71–79. PMLR, 2013\.
> >
> > \[28\] Jack W Silverstein and Zhi Dong Bai. On the empirical distribution of eigenvalues of a class
> > of large dimensional random matrices. Journal of Multivariate analysis, 54(2):175–192,
> > 1995\.
> >
> > \[29\] Aditya Vardhan Varre, Loucas Pillaud-Vivien, and Nicolas Flammarion. Last iterate convergence of sgd for least-squares in the interpolation regime. Advances in Neural Information
> > Processing Systems, 34:21581–21591, 2021\.
> >
> > \[30\] Difan Zou, Jingfeng Wu, Vladimir Braverman, Quanquan Gu, and Sham M Kakade. Benign
> > overfitting of constant-stepsize sgd for linear regression. Journal of Machine Learning
> > Research, 24(326):1–58, 2023\.
> >
> > \[31\] d'Ascoli, Stéphane, et al. "On the interplay between data structure and loss function in classification problems." *Advances in Neural Information Processing Systems* 34 (2021): 8506-8517.
> >
> > \[32\] Loureiro, Bruno, et al. "Learning curves of generic features maps for realistic datasets with a teacher-student model." *Advances in Neural Information Processing Systems* 34 (2021): 18137-18151.

---

### Comment · Area_Chair_vGNS · 2024-08-08

Dear reviewers,

could you have a look at the authors response and comment on them if you have done so, yet.

thanks in advance

your area chair

---

### Decision · Program_Chairs · 2024-09-25

**Decision:**

Accept (spotlight)

**Comment:**

This paper will become an influential paper of understanding the scaling law research. I would recommend a solid accept.